# Summary Statistic Privacy in Data Sharing

## Abstract

Data sharing between different parties has become increasingly common across industry and academia. An important class of privacy concerns that arises in data sharing scenarios regards the underlying distribution of data. For example, the total traffic volume of data from a networking company can reveal the scale of its business, which may be considered a trade secret. Unfortunately, existing privacy frameworks (e.g., differential privacy, anonymization) do not adequately address such concerns. In this paper, we propose *summary statistic privacy*, a framework for analyzing and protecting these summary statistic privacy concerns. We propose a class of quantization mechanisms that can be tailored to various data distributions and statistical secrets, and analyze their privacy-distortion tradeoffs under our framework. We prove corresponding lower bounds on the privacy-utility tradeoff, which match the tradeoffs of the quantization mechanism under certain regimes, up to small constant factors. Finally, we demonstrate that the proposed quantization mechanisms achieve better privacy-distortion tradeoffs than alternative privacy mechanisms on real-world datasets.

## 1 Introduction

Data sharing between organizations is an important driver for many use cases, including data-driven product development (Lee & Whang, 2000), industry-wide coordination efforts (e.g., cybersecurity (Choucri et al., 2016), law enforcement (Jacobs & Blitsa, 2008)), and the creation of benchmarks for evaluating scientific progress (Deng et al., 2009; Reiss et al., 2011; Luo et al., 2021). For example, network traces shared from customers to networking vendors enable vendors to debug and improve products (Yin et al., 2022; cai). Medical data shared between hospitals (Esteban et al., 2017; Warren et al., 2019) enables them to develop new machine-learning-based diagnosis algorithms collaboratively (Chaibub Neto et al., 2019). Data shared by researchers allow their research to be reproducible by others (Deng et al., 2009; Lin et al., 2020). In recent years, data sharing has grown into its own sub-industry (e.g., data marketplaces on platforms such as Databricks and Snowflake). Shared data can take many forms, including processed or scrubbed raw data (Reiss et al., 2012; Google, 2018; Commission, 2018; Warren et al., 2019), aggregate analytics, and/or synthetic data (Liu & Wu, 2022).

However, *summary statistics* of the shared data may leak sensitive information (Suri & Evans, 2021; Suri et al., 2023). For example, *property inference* attacks allow an attacker to infer properties about the individuals in the training dataset of a released machine learning model (Ateniese et al., 2015; Ganju et al., 2018; Zhang et al., 2021; Mahloujifar et al., 2022; Chaudhari et al., 2022). A video content provider that shares video session data may wish to hide the total or mean traffic volume, which could be used to infer the company's total revenue (Manousis et al., 2021). A cloud provider that shares cluster performance traces may not want to reveal the proportions of different server types that the cloud provider owns, which are regarded as business secrets (Lin et al., 2020). Note that this information (total/mean traffic volume, proportions of data types) cannot be inferred from any single record, but is inherent to the overall data distribution (or the aggregate dataset).

Unfortunately, existing privacy metrics and privacy-preserving data sharing algorithms do not adequately address these *summary statistic privacy concerns*. They either focus on protecting the privacy of individual records in a database (e.g., differential privacy (Dwork et al., 2006), anonymization (Reiss et al., 2012), sub-

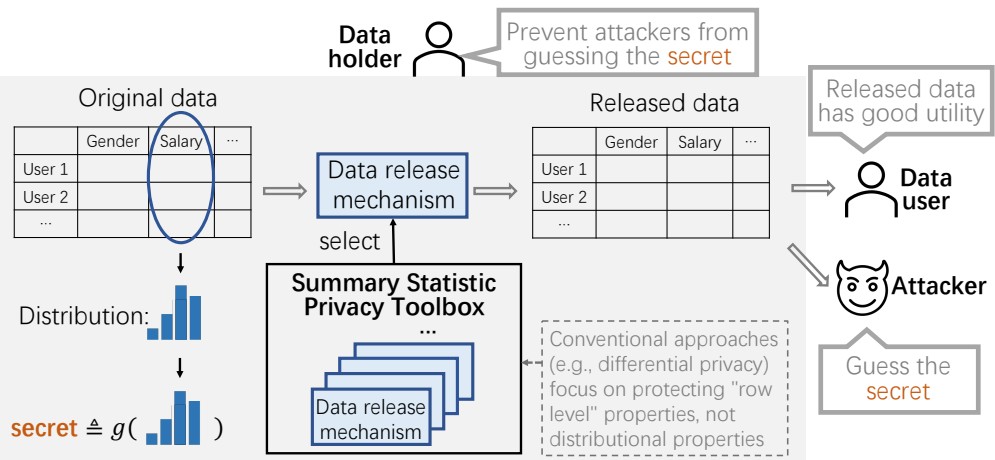

Figure 1: Problem overview. The data holder produces released data and wants to hide *statistical secrets* of the original data. The data user requires that the utility of the released data be good. The attacker (could be the data user) also observes the released data, and wants to guess the *secrets* of the original data. Note that we focus on secrets about the *underlying distribution* (e.g., mean, quantile, standard deviation, of a specific data *column*). As a comparison, many of existing frameworks (e.g., differential privacy (Dwork et al., 2006), anonymization (Reiss et al., 2012), sub-sampling (Reiss et al., 2012)) protect information from *individual samples (rows)*. Our end goal is to provide a *summary statistic privacy toolbox* for data holders to use. The summary statistic privacy toolbox contains data release mechanisms for a set of pre-defined secrets and data distributions. Data holders can choose the mechanism according to the secret that they want to hide and the closest data distributions.

sampling (Reiss et al., 2012)), or are designed for algorithms that release low-dimension statistical queries of the dataset instead of the entire dataset (Zhang et al., 2022; Makhdoumi et al., 2014; Issa et al., 2019). For example, differential privacy (DP) (Dwork et al., 2006), a *de facto* privacy definition, evaluates how much individual samples influence the final output of an algorithm. Assume that a video content provider has a dataset of daily page views that they want to release, and they are concerned about the *mean* page views (as this implies the revenue). A typical DP algorithm (Wasserman & Zhou, 2010) would add noise (e.g., Laplace) to the individual page view counts. This process does not change the *mean* of the entire data on expectation. Indeed, DP mechanisms have been shown not to protect summary statistics (Ateniese et al., 2015) (in fact, they are designed to preserve them). See more discussion in §2.2.

Hence, a privacy framework is needed for *defining, analyzing, and protecting summary statistic privacy concerns* in data sharing settings. Early work in this space has aimed to obfuscate only between two possible data distributions (Suri & Evans, 2021; Suri et al., 2023), or has been implicitly designed for the release of low-dimensional query release (Zhang et al., 2022). In this paper, we aim to design a general summary statistic privacy framework that can apply to general data release settings. At a high level, the proposed framework works as follows (detailed formulation in §3). A data holder first chooses one or more secrets, which are mathematically defined as functions of the data holder's data distribution. For example, a video analytics company might choose the mean daily observed traffic as a secret quantity. Then, the data holder obfuscates their data according to some *mechanism* and releases the output (Fig. 1). Our framework quantifies the *privacy* of this mechanism by analyzing the probability that a worst-case attacker can infer the data holder's true secret after observing the output. To capture the utility of released data, we define the *distortion* of a mechanism as the worst-case distance (where the distance metric can be chosen by the data holder or data user) between the original and released data distributions. Our goal is to design data release mechanisms that control tradeoffs between privacy and distortion.

## 1.1 Contributions

Our contributions are as follows.

- **Formulation (§3):** We formalize the notion of summary statistic privacy and propose privacy and distortion metrics tailored to data sharing applications. Intuitively, we define privacy as a worst-case adversary's probability of guessing a secret function of the underlying data distribution. We define distortion as the worst-case distributional distance[1] between the original data distribution and the released, perturbed data distribution. Precise definitions are in §3.
- **Mechanism design (§5):** We propose a class of mechanisms that achieve summary statistic privacy called *quantization mechanisms*, which intuitively quantize a data distribution's parameters[2] into bins. We present a *sawtooth technique* for theoretically analyzing the quantization mechanism's privacy tradeoff under various types of secret functions and data distributions (§5.3). Intuitively, the sawtooth technique exploits the geometry of the distribution parameter(s) to divide the parametric space into two regions: one in which privacy risk is small and analytically tractable, and another in which privacy risk can be high, but which occurs with low probability. The method is named after the boundary of the tractable region, which has a sawtooth shape. We use the sawtooth technique to analyze the quantization mechanism under various secret functions and data distributions (summary in Table 1). For most of these case studies, we provide concrete upper bounds characterizing the exact privacy-distortion tradeoff under a family of priors over the true data distribution parameters. For the remaining case studies, we provide a dynamic programming algorithm that efficiently numerically instantiates the quantization mechanism.
- **Lower bounds (§4):** We derive general lower bounds on distortion given a privacy budget for any mechanism. These bounds depend on both the secret function and the data distribution. We then instantiate the lower bounds for each of our case studies to show that for the case studies we analyze theoretically in Table 1, our proposed quantization mechanism achieves a privacy-distortion tradeoff within a small constant factor of optimal (usually 3) in the regime where quantization bins are small relative to the overall support set of the distribution parameters.
- **Empirical evaluation (§7):** We give empirical results showing how to use summary statistic privacy to release a real dataset, and how to evaluate the corresponding summary statistic privacy metric. We show that the proposed quantization mechanism achieves better privacy-distortion tradeoffs than other natural privacy mechanisms.

This paper is only a first step in the study of summary statistic privacy. Our formulation has many limitations and leaves many questions unanswered (§9). Still, we hope it will draw attention to what we believe to be an important privacy concern and research question.

## 2 Motivation and Related Work

In this section, we discuss motivating scenarios where summary statistic privacy is a concern (§2.1), and why existing privacy frameworks are not able to capture and protect summary statistic privacy (§2.2).

### 2.1 Motivating Scenarios

Whether sharing data models (e.g., classifiers (Ateniese et al., 2015; Ganju et al., 2018; Mahloujifar et al., 2022; Chaudhari et al., 2022), generative models (Zhou et al., 2021)) or datasets (e.g., cluster traces (Wilkes, 2020; Cortez et al., 2017; Luo et al., 2021), video session data (Jiang et al., 2016; Manousis et al., 2021), network flow datasets (Zeng, 2017)), data sharing can leak *sensitive global properties of the data distribution*. Examples include:

**S1. Business strategies** can be leaked from data. As mentioned before, cluster trace datasets (Wilkes, 2020; Cortez et al., 2017; Luo et al., 2021) are very useful in the systems community. However, cluster traces can reveal strategic enterprise choices, such as the fraction of server types in use (Lin et al., 2020). Such information reflects the company's business strategy and should be kept secret from competitors and vendors. Note that simply removing the server type from the dataset is not a good option, as server type is an

---

[1] In this work, we consider Wasserstein-1 distance and total variation distance (§3), though our formulation can accommodate other distance metrics.

[2] We assume data distributions are drawn from a parametric family; more details in §3.

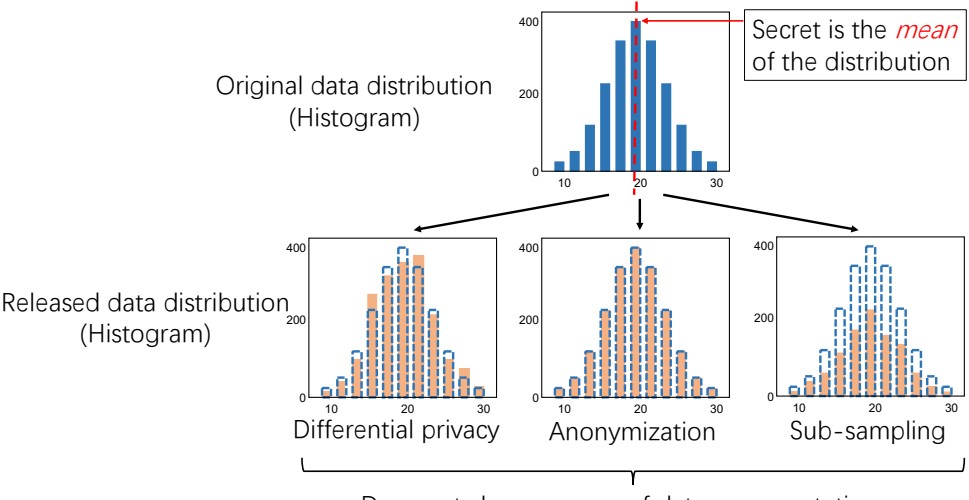

Figure 2: An illustrative example of why some of the privacy frameworks are not suitable for summary statistic privacy. Assume that we want to protect the *mean* of the data. A typical differential privacy algorithm (Wasserman & Zhou, 2010) would add zero-mean noise (e.g., Laplace noise) to the bins. Anonymization (Wilkes, 2020) removes sensitive features (e.g., name of users) from data but leaves other features the same. Sub-sampling (Reiss et al., 2012) down-sample the dataset. All of these mechanisms do not change the expected mean of the data, and thus an attacker can still guess the mean with a small (expected) error. See §2.2 for the discussion of other privacy mechanisms.

important feature for the downstream applications of the dataset (e.g., for predicting future CPU/memory usage).

**S2. Business scales** can be leaked from data. For example, networking datasets that contain traffic measurements or raw records are another common type of data (e.g., Meta flow trace dataset (Zeng, 2017), Wikipedia Web Traffic Dataset (Google, 2018), video session data used in Manousis et al. (2021)). While being useful, the total (or mean) traffic volume in these datasets (e.g., number of transferred bytes in a network, number of page views of websites, viewership values of video delivery systems) can reveal the scale of the business such as the number of users and the revenue of the company. Indeed, due to these concerns, it is a common practice to hide the actual traffic volumes of sensitive proprietary datasets even in research papers (e.g., removing the actual traffic values in Manousis et al. (2021)).

**S3. System capabilities** can also be revealed. For instance, the cluster trace datasets mentioned before (Wilkes, 2020; Cortez et al., 2017; Luo et al., 2021) contain CPU and memory usage of servers. It is likely that the maximum value of memory usage is close to the memory size of the system. Such system capabilities could be used by adversaries to launch attacks (e.g., denial-of-service attacks). Due to these concerns, some companies use customized techniques to obfuscate system capabilities before data release (e.g., normalizing system usage (Wilkes, 2020)).

**S4. Company sentiment or performance** [Example 1 from Mahloujifar et al. (2022)] A company releases a spam classifier trained on company emails. However, using property inference, an attacker is able to infer the aggregate sentiment of those emails (positive/negative). If the fraction of negative emails is high, it suggests that company morale is low, which is sensitive.

## 2.2 Existing Privacy Frameworks are Insufficient for Summary Statistic Privacy

Most existing privacy frameworks or mechanisms are not suitable for summary statistic privacy because they either focus on protecting individual records in the data (e.g., differential privacy (Dwork et al., 2006), anonymization (Wilkes, 2020), sub-sampling (Reiss et al., 2012)) (Fig. 1), or are designed for algorithms

that release low-dimension statistical queries of the dataset instead of the entire dataset (e.g., attribute privacy (Zhang et al., 2022), maximal leakage (Issa et al., 2019), privacy funnel (Makhdoumi et al., 2014)). We divide the relevant work into three categories: approaches that are based on indistinguishability over candidate distributions or inputs, industry heuristics, and information-theoretic approaches.

### 2.2.1 Indistinguishability Approaches

This class of approaches provides privacy by ensuring that pairs of input datasets or data distributions are indistinguishable. These approaches are typically motivated by differential privacy (Dwork et al., 2006).

**Differential privacy (DP)** Dwork et al. (2006) is one of the most popular privacy notions. A random mechanism $\mathcal{M}$ is $(\epsilon, \delta)$-differentially-private if for any neighboring datasets $D_0$ and $D_1$ (i.e., $D_0$ and $D_1$ differ one sample), and any set $S \subseteq range(\mathcal{M})$, we have

$$\mathbb{P}\left(\mathcal{M}\left(D_0\right) \in S\right) \leq e^\epsilon \cdot \mathbb{P}\left(\mathcal{M}\left(D_1\right) \in S\right) + \delta \ .$$

In our data sharing scenarios, we could apply DP framework by treating $\mathcal{M}$ as the data release mechanism that reads the original dataset and outputs the released dataset. However, the privacy concerns of DP and our suggested framework are completely different: we aim to hide functions of *a distribution*, while DP aims to hide whether *any given sample* contributed to the shared data. For example, we say that we want to release the data in Fig. 2 while protecting its mean. A typical differential privacy algorithm (Wasserman & Zhou, 2010) would add zero-mean noise (e.g., Laplace noise) to the bins. This process does not change the expected mean of the data, and therefore, the attack is still able to derive an unbiased estimator of the mean from the released data. Indeed, we will show through experiments in §7 that this DP mechanism is not effective in hiding statistical secrets.

There exist generalizations of DP for protecting more general random variables (besides individual samples) (Chatzikokolakis et al., 2013). However, a strong DP guarantee such that any two datasets with different secrets are indistinguishable from the released datasets implies that the released dataset has bad utility. For example, suppose that the original distributions are Gaussian distributions $\mathcal{N}\left(\mu, \sigma^2\right)$, and the secret is the mean of the distribution $\mu$. Two distributions with different secrets could have very different $\sigma^2$. To make any two distributions with different secrets (e.g., $\mathcal{N}(0, 1)$ and $\mathcal{N}(1, 100)$) indistinguishable from the released dataset, we must destroy information about the true $\sigma$. While relaxations like metric differential privacy relaxation may help (Chatzikokolakis et al., 2013), this also introduces new challenges, e.g., how to choose the metric function that maps dataset distance to a privacy parameter.

**Attribute privacy** (Zhang et al., 2022) considers a similar privacy concern as us: it tries to protect a function of a sensitive column in the dataset (named *dataset attribute privacy*) or a sensitive parameter of the underlying distribution from which the data is sampled (named *distribution attribute privacy*). Attribute privacy addresses the previously-mentioned shortcomings of vanilla DP under the *pufferfish privacy framework* (Kifer & Machanavajjhala, 2014). Roughly, an algorithm is said to satisfy dataset/distribution attribute privacy if for any two different ranges of a secret function value (e.g., the fraction of the server type $A$ is in $[0.1, 0.2)$ or $[0.2, 0.3)$), the distributions of the algorithm output do not differ too much. Attribute privacy constrains the set of candidate distributions a priori, which prevents the problem we discussed earlier, in which vanilla DP requires the addition of unbounded noise (Zhang et al., 2021).

Although their privacy concerns are highly related to ours, attribute privacy focuses on algorithms that output *a statistical query of the dataset* instead of the entire dataset. We could apply their framework to analyze full-dataset-sharing algorithms, but due to the high dimensionality of the dataset, attribute privacy needs to add substantial noise, which harms utility (§7).

**Distribution privacy** (Kawamoto & Murakami, 2019) is a closely related notion, which releases a full data distribution under DP-style indistinguishability guarantees. Roughly, for any two input distributions $\theta_0$ and $\theta_1$ from a pre-defined set of candidate distributions, a distribution private mechanism outputs a distribution $\mathcal{M}(\theta_i)$ such that for any set $S$ in the output space, we have $\mathbb{P}[\mathcal{M}(\theta_i) \in S] \leq e^\epsilon \mathbb{P}[\mathcal{M}(\theta_{1-i}) \in S] + \delta$.

This formulation is stronger than what we need; by obfuscating the whole distribution, we inherently protect the private information in question. However mechanisms that protect distribution privacy may add

more noise than what is required only to protect select secret(s). A recent work by Chen and Ohrimenko (Chen & Ohrimenko, 2022) proposes mechanisms for distribution privacy, and we observe exactly this trend experimentally in §7; the noise added by the mechanisms in Chen & Ohrimenko (2022) is larger than what we require with summary statistic privacy.

**Distribution inference** (Suri & Evans, 2021; Suri et al., 2023) is very closely related to our goals. Like our setting, the data holder is trying to protect a secret function of its data (or data distribution). To this end, it sets up a hypothesis test in which the adversary must choose whether the released model (or data) comes from one of two fixed data distributions, which are derived from an underlying public data distribution. These two distributions are assumed to be known both to the attacker and the defenders. In many practical settings, it may be difficult to establish a reasonable pair of candidate distributions; moreover, this approach is not directly aligned with the data holder's goal, which is simply to hide some secret quantities — not to render the full data distribution indistinguishable with another (the latter is closer to distribution privacy).

### 2.2.2 Industry Heuristics

Industry heuristics are algorithms that are commonly used in industrial data sharing settings. They may not provide provable privacy guarantees, and indeed, many of these heuristics have been broken in practice. Examples include **anonymization**, which removes certain attributes (e.g., name of the patients in medical data, name of jobs in cluster dataset) (Reiss et al., 2012); anonymization is widely used in the release of datasets (e.g., Wilkes (2020)). However, it does not change the distribution of attributes. Another example is **sub-sampling**, which works by sampling the original datasets at the level of individual records (Reiss et al., 2012). The intuition is that by reducing the number of samples, less information is leaked. However, sub-sampling does not change statistical properties of the distribution.

### 2.2.3 Information-Theoretic Approaches

The third category of defenses are information theoretic. These approaches have a similar goal to ours and typically rely on (or relate to) the mutual information between problem variables.

**Maximal leakage** (Issa et al., 2019) is an information-theoretic framework for quantifying the leakage of sensitive information. We denote $X$ as the random variable of the data to be shared (which may contain sensitive information), and $Y$ as the random variable of the information that is processed from $X$ and is accessible to the attacker. Having observed $Y$, the attacker's goal is to guess a secret function of $X$ denoted by $U$, and the guess is denoted by $\hat{U}$. Based on this setup, the Markov chain $U - X - Y - \hat{U}$ holds. Maximal leakage $\mathcal{L}$ from $X$ to $Y$ is defined as

$$\mathcal{L}\left(X \to Y\right) = \sup_{U-X-Y-\hat{U}} \log \frac{\mathbb{P}\left(U = \hat{U}\right)}{\max_u P_U(u)} \tag{1}$$

where the sup is taken over $U$ (i.e., considering the worst-case secret) and $\hat{U}$ (i.e., considering the strongest attacker). Intuitively, Eq. (1) evaluates the ratio (in nats) of the probabilities of guessing the secret $U$ correctly with and without observing $Y$.

To apply maximal leakage in data sharing scenario, we may regard $X$ as the original dataset, $Y$ as the released dataset, and $U$ as the secret (e.g., the fraction of a specific server type). However, this formulation is still unsuitable for the following reasons. (1) Maximal leakage only considers discrete $U$ and $\hat{U}$ under finite alphabet. Note that it is a critical assumption for making sure that $\mathbb{P}\left(U = \hat{U}\right)$ in the definition (Eq. (1)) is nonzero. However, in our problem, secrets typically have continuous support (e.g., §2.1). (2) Maximal leakage assumes that the secret to protect $U$ is unknown a priori and therefore considers the worst-case leakage among all possible secrets. However, in our problem, data holders know what secret they want to protect. Although we cannot directly use maximal leakage in our problem, its core idea can be useful for extending our framework (see §9).

**Privacy funnel** (Makhdoumi et al., 2014) is another popular information-theoretic privacy framework. As with maximal leakage, we denote $X$ as the random variable of the data that many contain sensitive

information $U$, and $Y$ as the random variable of the information that is processed from $X$ and is accessible by the attacker. The privacy funnel framework evaluates privacy leakage with the mutual information $I(U;Y)$, and the utility of $Y$ with mutual information $I(X;Y)$. To find a good privacy-preserving data processing strategy $P_{Y|X}$, the privacy funnel solves the optimization

$$\min_{P_{Y|X}:I(X;Y)\geq R} I(U;Y) \quad ,$$

where $R$ is a desired threshold on the utility of $Y$.

To apply it in data sharing problems, we could regard $X$ as the original data, $Y$ as the released data, and $U$ as the secret data holder wants to protect (e.g., the fraction of a specific server type). However, mutual information is not a good metric for either privacy or utility. On the privacy front, prior work has shown that $I(U;Y)$ can be reduced while allowing the attacker to guess $S$ correctly from $Y$ with higher probability (see Example 1 in Issa et al. (2019)). On the utility front, higher mutual information $I(X;Y)$ does not mean that the released data $Y$ is a useful representation of $X$. For example, $Y$ could be an arbitrary one-to-one transformation of $X$. In that case, $I(X;Y)$ is maximized, but the data structure could be completely destroyed. In addition, privacy funnel (Makhdoumi et al., 2014) only considers $X$ and $Y$ in discrete supports, which is too restrictive for our setting.

## 3  Summary Statistic Privacy Formulation

**Notation.** We denote random variables with uppercase English letters or upright Greek letters (e.g., $X, \upmu$), and their realizations with italicized lowercase letters (e.g., $x, \mu$). For a random variable $X$, we denote its probability density function (PDF), or, in the case of discrete random variables, its probability mass function (PMF), as $f_X$, and its distribution measure as $\omega_X$. If a random variable $X$ is drawn from a parametric family (e.g., Gaussian with specified mean and covariance); the parameters will be denoted with a subscript of $X$, i.e., the above notations become $X_\theta$, $f_{X_\theta}$, $\omega_{X_\theta}$ respectively for parameters $\theta \in \mathbb{R}^q$, where $q \geq 1$ denotes the dimension of the parameters. In addition, we denote $f_{X|Y}$ as the conditional PDF or PMF of $X$ given another random variable $Y$. We use $\mathbb{Z}, \mathbb{Z}_{>0}, \mathbb{N}, \mathbb{R}, \mathbb{R}_{>0}$, to denote the set of integers, positive integers, natural numbers, real numbers, and positive real numbers respectively.

**Original data.** Consider a data holder who possesses a dataset of $n$ samples $\mathcal{X} = \{x_1, \ldots, x_n\}$, where for each $i \in [n]$, $x_i \in \mathbb{R}^p$ is drawn i.i.d. from an underlying distribution. We assume the distribution comes from a parametric family, and the parameter vector $\theta \in \mathbb{R}^q$ of the distribution fully specifies the distribution. That is, $x_i \sim \omega_{X_\theta}$, where we further assume that $\theta$ is itself a realization of random parameter vector $\Theta$, and $\omega_\Theta$ is the probability measure for $\Theta$. We will discuss how to relax the assumption on this prior distribution of $\theta$ in §9. We assume that the data holder knows $\theta$ (and hence knows its full data distribution $\omega_{X_\theta}$); our results and mechanisms generalize to the case when the data holder only possesses the dataset $\mathcal{X}$ (see §6).

For example, suppose the original data samples come from a Gaussian distribution. We have $\theta = (\mu, \sigma)$, and $X_\theta \sim \mathcal{N}(\mu, \sigma)$. $\omega_\Theta$ (or $f_\Theta$) describes the prior distribution over $(\mu, \sigma)$. For example, if we know a priori that the mean of the Gaussian is drawn from a uniform distribution between 0 and 1, and $\sigma$ is always 1, we could have $f_\Theta(\mu, \sigma) = \mathbb{I}(\mu \in [0, 1]) \cdot \delta(\sigma)$, where $\mathbb{I}(\cdot)$ is the indicator function, and $\delta$ is the Dirac delta function. In practice, the underlying distribution can be much more complicated than a Gaussian.

In general, the data can be multi-dimensional (i.e., $p > 1$). We study one-dimensional data as a starting point (§3.2).

**Statistical secrets to protect.** We assume the data holder wants to hide $\ell \in \mathbb{Z}_{>0}$ *secrets* from the original data distribution. Since the true data distribution is fully-specificed by parameter vector $\theta$, these secrets can be expressed as a function $g(\theta) : \mathbb{R}^q \to \mathbb{R}^\ell$. In the Gaussian example $X_\theta \sim \mathcal{N}(\mu, \sigma)$, suppose the random variable $X_\theta$ represents the traffic volume experienced by an enterprise in a day. The data holder may wish to hide the mean traffic per day, in which case $g(\cdot)$ would be the mean of the distribution, i.e., $g(\mu, \sigma) = \mu$. In this example, we are hiding only one secret (the mean), so $\ell = 1$. *In general, the secret can be any (vector-valued) function that can be deterministically computed from $\theta$.* As shown in Fig. 1, the secret could be derived from one feature (e.g., the mean salary) or computed from multiple features (e.g., the

mean salary of males). The secrets could also be multi-dimensional (e.g., mean of salary, and the fraction of males). In this paper, we present general results for one-dimensional secrets (i.e., $\ell = 1$) and defer a discussion of higher-dimensional secrets to future work (see §9).

**Data release mechanism.** The data holder releases data by passing the private parameter $\theta$ through a *data release mechanism* $\mathcal{M}_g$. That is, for a given $\theta$, the data holder first draws internal randomness $z \sim \omega_Z$, and then releases another distribution parameter $\theta' = \mathcal{M}_g(\theta, z)$, where $\mathcal{M}_g$ is a deterministic function, and $\omega_Z$ is a fixed distribution from which $z$ is sampled. Note that we assume both the input and output of $\mathcal{M}_g$ are distribution parameters. It is straightforward to generalize to the case when the input and/or output are datasets of samples (see §6).

For example, in the Gaussian case discussed above, the data release mechanism can be $\mathcal{M}_g((\mu, \sigma), z) = (\mu + z, \sigma)$ where $z \sim \mathcal{N}(0, 1)$. I.e., this mechanism shifts the mean of the Gaussian by a random amount drawn from a standard Gaussian distribution and keeps the variance.

**Threat model.** We assume that the attacker knows the parametric family from which our data is drawn, but does not know the initial parameter $\theta$. The attacker is also assumed to know the data release mechanism $\mathcal{M}_g$ and output $\theta'$ but not the realization of the data holder's internal randomness $z$. The attacker guesses the initial secret $g(\theta)$ based on the released parameter $\theta'$ according to estimate $\hat{g}(\theta')$. $\hat{g}$ can be either random or deterministic, and we assume no computational bounds on the adversary. For instance, in the running Gaussian example, an attacker may choose $\hat{g}(\mu', \sigma') = \mu'$. When the data holder releases a dataset of samples instead of the parameter $\theta'$, this formulation can be used to upper bound the attacker's performance on correctly guessing the secret, since the estimation error on released distribution parameter is induced due to the finite samples in the released dataset.

### 3.1 Metrics

**Privacy metric.** The data holder wishes to prevent an attacker from guessing its secrets. We define our privacy metric privacy $\Pi_{\epsilon, \omega_\Theta}$ as the attacker's probability of guessing the secret(s) to within a tolerance $\epsilon$, taken worst-case over all attackers $\hat{g}$:

$$\Pi_{\epsilon, \omega_\Theta} \triangleq \sup_{\hat{g}} \ \mathbb{P}\left(|\hat{g}(\theta') - g(\theta)| \leq \epsilon\right) . \tag{2}$$

The probability is taken over the randomness of the original data distribution ($\theta \sim \omega_\Theta$), the data release mechanism ($z \sim \omega_Z$), and the attacker strategy ($\hat{g}$).

**Distortion metric.** The main goal of data sharing is to provide useful data; hence, we (and data holders and users) want to understand how much the released data distorts the original data. We define the *distortion* $\Delta$ of a mechanism as the worst-case distance between the original distribution and the released distribution:

$$\Delta \triangleq \sup_{\substack{\theta \in \mathrm{Supp}(\omega_\Theta), \theta', \\ z \in \mathrm{Supp}(\omega_Z): \mathcal{M}_g(\theta, z) = \theta'}} d\left(\omega_{X_\theta} \| \omega_{X_{\theta'}}\right), \tag{3}$$

where $d$ is a general distance metric defined over distributions. The choice of the distance metric depends on the data type and potentially on the applications that stakeholders care about. For example, if the data holders or users have concrete metrics that they want to preserve (e.g., the difference between the mean salaries of males and females in Fig. 1), they could use this quantity as the distance metric. Otherwise, one can use statistical distance metrics between distributions (e.g., total variation distance, Wasserstein distance). In this paper, we adopt Wasserstein-1 distance for continuous distributions and total variation (TV) distance for discrete distributions. These distances are often used for evaluating data quality (e.g., Yin et al. (2022); Lin et al. (2020)) and as the distance metric in neural network design (e.g., Arjovsky et al. (2017); Lin et al. (2018)). Note that the definition in Eq. (3) can be extended to data release mechanisms that take datasets as inputs and/or outputs.

**Objective.** To summarize, the data holder's objective is to choose a data release mechanism that minimizes distortion metric $\Delta$ subject to a constraint on privacy $\Pi_{\epsilon,\omega_\Theta}$:

$$\min_{\mathcal{M}_g} \quad \Delta$$
$$\text{subject to} \quad \Pi_{\epsilon,\omega_\Theta} \leq T. \tag{4}$$

The alternative formulation, $\min_{\mathcal{M}_g} \Pi_{\epsilon,\omega_\Theta}$ subject to $\Delta \leq T$ is analyzed in App. A.

The optimal data release mechanisms for Eq. (4) depends on the secrets, the distance metric $d$ in Eq. (3), and the characteristics of the original data. We envision a *summary statistic privacy toolbox* (Fig. 1) that encodes data release mechanisms for a list of predefined secrets, $d$, and data distributions. Data holders specify the secret function they want to protect and the desired distance metric; the toolbox then selects the data distribution parametric family that most closely reflects the holder's raw data and uses the corresponding data release mechanism to process the raw data for sharing.

### 3.2 Scope of This Work

#### 3.2.1 Simplifying Assumptions

Although our formulation supports a wide range of distribution distance metrics, secret functions, and parametric families of data distributions, we make simplifying assumptions as a starting point on this problem.

**Distortion metric.** As discussed in §3.1, we use Wasserstein-1 and TV as the distance metrics for continuous and discrete distributions respectively in the case studies (§6). We leave the discussion of other metrics to §9.

**The type and the number of secrets.** Our formulation supports general statistical secrets, as long as they are a (possibly vector-valued) function of the data distribution. In this paper, we start by assuming that the secret is one-dimensional, and discuss several natural secret functions in §6.

**The dimension and distribution of the data.** Although our formulation includes multi-dimensional data, in this paper, we consider one-dimensional distributions as a starting point.

#### 3.2.2 Research Questions

We aim to answer two questions:

Q1 What are fundamental limits on the tradeoff between privacy and distortion?

Q2 Do there exist data release mechanisms that can match or approach these fundamental limits?

In general, these questions can have different answers for different choices of distance metric in Eq. (3), different parametric families of data distributions, and different secret functions. In §4 and §5, we first present general results that do not depend on data distribution or secret function. We then present case studies for specific secrets and data distributions for building up our initial *summary statistic privacy toolbox* in §6.

## 4 General Lower Bound on Privacy-Distortion Tradeoffs

Given a privacy budget $T$, we first present a lower bound on distortion that applies *regardless of the prior distribution of data* $\omega_\Theta$ and *regardless of the secret* $g$. As discussed in §3.2, we assume that the secret is scalar (i.e., $\ell = 1$), but the data distribution can have arbitrary dimension.

**Theorem 1** (Lower bound of privacy-distortion tradeoff). *Let* $D\left(X_{\theta_1}, X_{\theta_2}\right) \triangleq \frac{1}{2}d\left(\omega_{X_{\theta_1}} \| \omega_{X_{\theta_2}}\right)$, *where* $d\left(\cdot\|\cdot\right)$ *is defined in the line after Eq. (3). Further, let* $R\left(X_{\theta_1}, X_{\theta_2}\right) \triangleq |g(\theta_1) - g(\theta_2)|$ *and*

$$\gamma \triangleq \inf_{\theta_1,\theta_2 \in Supp(\omega_\Theta)} \frac{D\left(X_{\theta_1}, X_{\theta_2}\right)}{R\left(X_{\theta_1}, X_{\theta_2}\right)}. \tag{5}$$

*For any $T \in (0, 1)$, when $\Pi_{\epsilon, \omega_\Theta} \leq T$,*

$$\Delta > \left( \lceil \frac{1}{T} \rceil - 1 \right) \cdot 2\gamma\epsilon . \tag{6}$$

The proof is shown as below. From Thm. 1 we see that the lower bound of distortion is inversely correlated with the privacy budget and positively correlated with the guess tolerance $\epsilon$. The dependent quantity $\gamma$ in Eq. (5) can be thought of as a conversion factor that bounds the translation from probability of detection to distributional distance. Note that we have not made $\gamma$ exact as its form depends on the type of the secret and prior distribution of data. We will instantiate it in the cases studies in §6.

*Proof.* Our proof proceeds by constructing an ensemble of attackers, such that at least one of them will be correct by construction. We do this by partitioning the space of possible secret values, and having each attacker output the midpoint of one of the subsets of the partition. We then use the fact that each attacker can be correct with probability at most $T$, combined with $\gamma$, which intuitively relates the distance between distributions to the distance between their secrets, to derive the claim. Recall that $\theta$ is the true private parameter vector, $\theta'$ is the released parameter vector as a result of the data release mechanism.

$$T \geq \Pi_{\epsilon, \omega_\Theta}$$
$$= \sup_{\hat{g}} \mathbb{P} \left( \hat{g}(\theta') \in [g(\theta) - \epsilon, g(\theta) + \epsilon] \right)$$
$$= \sup_{\hat{g}} \mathbb{E} \left( \mathbb{P} \left( \hat{g}(\theta') \in [g(\theta) - \epsilon, g(\theta) + \epsilon] \,\middle|\, \theta' \right) \right)$$
$$= \mathbb{E} \left( \sup_{\hat{g}} \mathbb{P} \left( \hat{g}(\theta') \in [g(\theta) - \epsilon, g(\theta) + \epsilon] \,\middle|\, \theta' \right) \right) , \tag{7}$$

where Eq. (7) is due to the following facts: (1) LHS $\leq$ RHS because $\sup_{\hat{g}} \mathbb{P} \left( \hat{g}(\theta') \in [g(\theta) - \epsilon, g(\theta) + \epsilon] \,\middle|\, \theta' \right) \geq \mathbb{P} \left( \hat{g}(\theta') \in [g(\theta) - \epsilon, g(\theta) + \epsilon] \,\middle|\, \theta' \right)$ for any $\theta'$; (2) RHS $\leq$ LHS because $\hat{g}$ can only depend on $\theta'$. Therefore, we can map any $\arg\sup_{\hat{g}}$ in the RHS to the LHS and obtain the same value, since the expectation is taken over $\theta'$. Thus, there exists $\theta'$ s.t. $\sup_{\hat{g}} \mathbb{P} \left( \hat{g}(\theta') \in [g(\theta) - \epsilon, g(\theta) + \epsilon] \,\middle|\, \theta' \right) \leq T$. Let

$$L_{\theta'} \triangleq \inf_{\theta \in \mathrm{Supp}(\omega_\Theta), z : \mathcal{M}_g(\theta, z) = \theta'} g(\theta) ,$$

$$R_{\theta'} \triangleq \sup_{\theta \in \mathrm{Supp}(\omega_\Theta), z : \mathcal{M}_g(\theta, z) = \theta'} g(\theta) .$$

We can define a sequence of attackers and a constant $N$ such that $\hat{g}_i(\theta') = L_{\theta'} + (i + 0.5) \cdot 2\epsilon$ for $i \in \{0, 1, \ldots, N-1\}$ and $L_{\theta'} + 2N\epsilon \geq R_{\theta'} > L_{\theta'} + 2(N-1)\epsilon$ (Fig. 3). From the above, we have

$$T \cdot N \geq \sum_i \mathbb{P} \left( \hat{g}_i(\theta') \in [g(\theta) - \epsilon, g(\theta) + \epsilon] \,\middle|\, \theta' \right) \geq 1,$$

Therefore, we have $N \geq \lceil \frac{1}{T} \rceil$, and

$$R_{\theta'} - L_{\theta'} > \left( \lceil \frac{1}{T} \rceil - 1 \right) \cdot 2\epsilon . \tag{8}$$

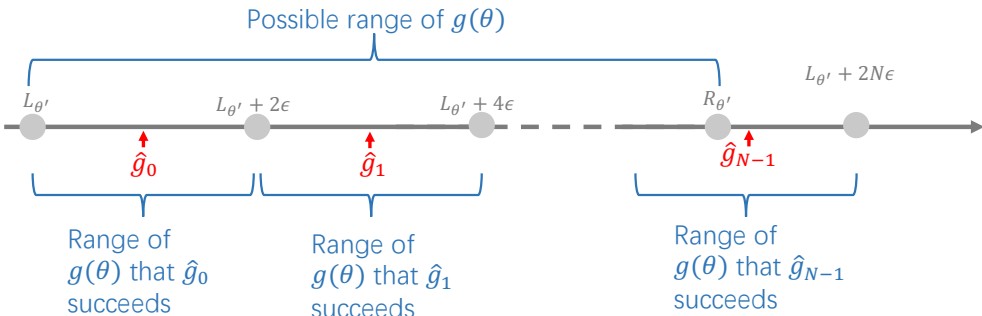

Figure 3: The construction of attackers for proof of Thm. 1. The $2\epsilon$ ranges of $\hat{g}_0, ..., \hat{g}_{N-1}$ jointly cover the entire range of possible secret $[L_{\theta'}, R_{\theta'}]$. The probability of guessing the secret correctly for any attacker is $\leq T$. Therefore, $R_{\theta'} - L_{\theta'} > \left(\lceil \frac{1}{T} \rceil - 1\right) \cdot 2\epsilon$ (Eq. (8)).

Then we have

$$\Delta \geq \sup_{\theta \in \mathrm{Supp}(\omega_\Theta), z \in \mathrm{Supp}(\omega_Z) : \mathcal{M}_g(\theta, z) = \theta'} d\left(\omega_{X_\theta} \| \omega_{X_{\theta'}}\right)$$

$$\geq \sup_{\theta_i \in \mathrm{Supp}(\omega_\Theta), z_i : \mathcal{M}_g(\theta_i, z_i) = \theta'} D\left(X_{\theta_1}, X_{\theta_2}\right) \tag{9}$$

$$> \left(\lceil \frac{1}{T} \rceil - 1\right) \cdot 2\gamma\epsilon. \tag{10}$$

where in Eq. (9), $\theta_i$ for $i \in \{1, 2\}$ denotes two arbitrary parameter vectors in the support space, and Eq. (9) comes from the triangle inequality, and Eq. (10) utilizes $R_{\theta'} - L_{\theta'} > \left(\lceil \frac{1}{T} \rceil - 1\right) \cdot 2\epsilon$ and the definition of $\gamma$. $\qquad\square$

## 5 Data Release Mechanisms

We first present in §5.1 the *quantization mechanism*, a template for data release mechanisms used in the case studies of §6. The quantization mechanism can be instantiated differently for different secret functions and data distributions. We show in §5.2 techniques for instantiating the quantization mechanism, either based on theoretical insights or numerically. Finally, we give some intuition in §5.3 about how to analyze the quantization mechanism. These insights will be used in our case studies (§6) to show that we can sometimes match the lower bounds from §4 up to small constant factors.

### 5.1 The Quantization Mechanism

At a high level, the quantization mechanisms follow two steps:

1. **Offline Phase:** Partition the space of parameters $\mathrm{Supp}(\Theta)$ into carefully-chosen bins.

2. **Online Phase:** For an observed data distribution parameter $\theta$, deterministically release the quantized parameters, according to the partition from the Offline Phase.

More precisely, we first divide the set of possible distribution parameters $\mathrm{Supp}(\Theta)$ into subsets $\mathcal{S}_i$ such that $\cup_{i \in \mathcal{I}} \mathcal{S}_i \supseteq \mathrm{Supp}(\Theta)$ and $\mathcal{S}_{i_1} \cap \mathcal{S}_{i_2} = \emptyset$ for $i_1 \neq i_2$, where $\mathcal{I}$ is the (possibly uncountable) set of indices of the subsets. For $\theta \in \mathrm{Supp}(\Theta)$, $I(\theta)$ is the index of the set that $\theta$ belongs to; in other words, we have $I(\theta) = i$, where $\theta \in \mathcal{S}_i$. The mechanism first looks up which set $\theta$ belongs to (i.e., $I(\theta)$), then *deterministically* releases a parameter $\theta^*_{I(\theta)}$ that corresponds to the set. Here, $\theta^*_i$ for $i \in \mathcal{I}$ denotes another parameter. In short, our data release mechanism has the form

$$\mathcal{M}_g(\theta, z) = \theta^*_{I(\theta)} \quad .$$

Note that the policy is fully determined by $\mathcal{S}_i$ and $\theta_i^*$. In the remainder of the paper, we will show different ways of instantiating quantization mechanism to approach the lower bound in §4.

Intuitively, quantization mechanisms will have a bounded distortion as long as $d\left(\omega_{X_\theta} \| \omega_{X_{\theta_{I(\theta)}^*}}\right)$ is bounded for all $\theta \in \text{Supp}(\Theta)$. At the same time, they obfuscate the secret as different data distributions within the same set are mapped to the same released parameter. It turns out this simple *deterministic* mechanism is sufficient to achieve the (order) optimal privacy-distortion trade-offs in many cases, as opposed to DP where randomness is required to achieve DP guarantees (Dwork et al., 2006) (examples in the case studies §6).

### 5.2 Algorithms for Instantiating the Quantization Mechanism

To implement the quantization mechanism, we need to define the quantization bins $\mathcal{S}_i$ and the released parameter per bin $\theta_i^*$. Depending on the data distribution, the secret function, and quantization mechanism parameters, the mechanism can have very different privacy-distortion tradeoffs. We present two methods for selecting quantization parameters: (1) an analytical approach, and (2) a numeric approach.

**(1) Analytical approach.** In some cases, outlined in the case studies of §6 and the appendices, we can find analytical expressions for $\mathcal{S}_i$ and $\theta_i^*$ while (near-)optimally trading off privacy for distortion. This is usually possible when the lower bound depends on the problem parameters in a particular way.

For example, for the Gaussian distribution where $\theta = (\mu, \sigma)$, when secret=standard deviation, we can work out the lower bound from Thm. 1 (details in App. G). Note that the lower bound is tight if our mechanism minimizes

$$\frac{D\left(X_{\mu_1,\sigma_1}, X_{\mu_2,\sigma_2}\right)}{R\left(X_{\mu_1,\sigma_1}, X_{\mu_2,\sigma_2}\right)} = \sqrt{\frac{1}{2\pi}} e^{-\frac{1}{2}\left(\frac{\mu_1-\mu_2}{\sigma_1-\sigma_2}\right)^2} - \left(\frac{\mu_1-\mu_2}{\sigma_1-\sigma_2}\right)\left(\frac{1}{2} - \Phi\left(\left(\frac{\mu_1-\mu_2}{\sigma_1-\sigma_2}\right)\right)\right) \tag{11}$$

where where $D\left(X_{\theta_1}, X_{\theta_2}\right)$ and $R\left(X_{\theta_1}, X_{\theta_2}\right)$ are defined in Thm. 1, and $\Phi$ denotes the CDF of the standard Gaussian distribution. That is, for any true parameters $\mu_1$ and $\sigma_1$, the mechanism should always choose to release $\mu_2$ and $\sigma_2$ such that Eq. (11) is as small as possible. The exact form of Eq. (11) is not important for now; notice instead that the problem parameters $(\sigma_i, \mu_i)$ take the same form every time they appear in this equation. We define $t(\theta_1, \theta_2) = \frac{\mu_1-\mu_2}{\sigma_1-\sigma_2}$ to be that form.[3] Next, we find the $t(\theta_1, \theta_2)$ that minimizes Eq. (11):

$$t_0 \triangleq \underset{t(\theta_1,\theta_2)}{\arg\inf} \frac{D\left(X_{\theta_1}, X_{\theta_2}\right)}{R\left(X_{\theta_1}, X_{\theta_2}\right)}$$

For instance, in our Gaussian example, we can write $t_0$ as

$$t_0 = \underset{t(\theta_1,\theta_2)}{\arg\inf} \sqrt{\frac{1}{2\pi}} e^{-\frac{1}{2}(t(\theta_1,\theta_2))^2} - (t(\theta_1,\theta_2))\left(\frac{1}{2} - \Phi\left(t(\theta_1,\theta_2)\right)\right),$$

which can be solved numerically. Finally, we can choose $\mathcal{S}_i$ and $\theta_i^*$ to be sets for which $t\left(\theta, \theta_i^*\right) = t_0$, $\forall \theta \in \mathcal{S}_i$. Using this rule, we derive the mechanism:

$$\mathcal{S}_{\mu,i} = \left\{(\mu + t_0 \cdot t, \underline{\sigma} + (i + 0.5) \cdot s + t) \,|\, t \in \left[-\frac{s}{2}, \frac{s}{2}\right)\right\} \quad,$$
$$\theta_{\mu,i}^* = (\mu, \underline{\sigma} + (i + 0.5) \cdot s) \quad,$$
$$\mathcal{I} = \{(\mu, i) \,|\, i \in \mathbb{N}, \mu \in \mathbb{R}\},$$

where $s$ is a hyper-parameter of the mechanism that divides $(\overline{\sigma} - \underline{\sigma})$, and $\overline{\sigma}, \underline{\sigma}$ are upper and lower bounds of $\sigma$.

For our Gaussian example, the resulting sets $\mathcal{S}_{\mu,i}$ for the quantization mechanism are shown in Fig. 4; the space of possible parameters is divided into infinitely many subsets $\mathcal{S}_{\mu,i}$, each consisting of a diagonal line

---

[3]Indeed, for many of the case studies in §6, $t(\theta)$ takes an analogous form; we will see the implications of this in the analysis of the upper bound in §5.3.

segment (parallel blue lines in Fig. 4). The space of possible $\sigma$ values is divided into segments of length $s$, which correspond to the horizontal bands in Fig. 4. The fact that the intervals $\mathcal{S}_{\mu,i}$ are diagonal lines arises from choosing $t(\theta_1, \theta_2) = \frac{\mu_1 - \mu_2}{\sigma_1 - \sigma_2}$; each interval corresponds to a set of points that satisfy $t(\theta_1, \theta_2) = t_0$, i.e., with slope $1/t_0$.

We will see how to use this construction to obtain upper bounds on privacy-distortion tradeoffs in §5.3.

**(2) Numeric approach.** In some cases, the above procedure may not be possible. To this end, we present a dynamic programming algorithm to numerically compute the quantization mechanism parameters. This algorithm achieves an optimal privacy-distortion tradeoff (Bellman, 1966) among the class of quantization algorithms with finite precision and continuous intervals $\mathcal{S}_i$. We use this algorithm in some of the case studies in §6. We present our dynamic programming algorithm for univariate data distributions.

We assume $\text{Supp}(\Theta) = [\underline{\theta}, \overline{\theta})$, where $\underline{\theta}, \overline{\theta}$ are lower and upper bounds of $\theta$, respectively. We consider the class of quantization mechanisms such that $\mathcal{S}_i = \left[\underline{\theta^i}, \overline{\theta^i}\right)$, i.e., each subset of parameters are in a continuous range. Furthermore, we explore mechanisms such that $\underline{\theta^i}, \overline{\theta^i}, \theta_i^* \in \{\underline{\theta}, \underline{\theta} + \kappa, \underline{\theta} + 2\kappa, \ldots, \overline{\theta}\}$, where $\kappa$ is a hyper-parameter that encodes numeric precision (and therefore divides $(\overline{\theta} - \underline{\theta})$). For example, if we want to hide the mean of a Geometric random variable with $\underline{\theta} = 0.1$ and $\overline{\theta} = 0.9$, we could consider three-decimal-place precision, i.e., $\kappa = 0.001$ and $\underline{\theta^i}, \overline{\theta^i}, \theta_i^* \in \{0.100, 0.101, 0.102, \ldots, 0.900\}$.

Since $\Delta$ (Eq. (3)) is defined as the *worst-case* distortion whereas $\Pi_{\epsilon, \omega_\Theta}$ (Eq. (2)) is defined as a *probability*, which is related to the original data distribution, optimizing $\Pi_{\epsilon, \omega_\Theta}$ given bounded $\Delta$ (Eq. (12)) is easier to solve than the final goal of optimizing $\Delta$ given bounded $\Pi_{\epsilon, \omega_\Theta}$ (Eq. (4)).

$$\min_{\mathcal{M}_g} \ \Pi_{\epsilon, \omega_\Theta} \qquad \text{subject to} \ \ \Delta \leq T. \tag{12}$$

Observing that in Eq. (4) the optimal value of $\min_{\mathcal{M}_g} \Delta$ is a monotonic decreasing function w.r.t. the threshold $T$, we can use a binary search algorithm (shown in App. B) to reduce problem Eq. (4) to problem Eq. (12). It calls an algorithm that finds the optimal quantization mechanism with numerical precision over continuous intervals under a distortion budget $T$ (i.e., solving Eq. (12)). This problem can be solved by a dynamic programming algorithm. Let $pri(t^*)$ ($t^* \in \{\underline{\theta}, \underline{\theta} + \kappa, \underline{\theta} + 2\kappa, \ldots, \overline{\theta}\}$) be the minimal privacy $\Pi_{\epsilon, \omega_\Theta}$ we can get for $\text{Supp}(\Theta) = \{X_\theta : \theta \in [\underline{\theta}, t^*)\}$ such that $\Delta \leq T$. Denote $\mathcal{D}(\theta_1, \theta_2)$ as the minimal distortion a quantization mechanism can achieve under the quantization bin $[\theta_1, \theta_2)$, we have

$$\mathcal{D}(\theta_1, \theta_2) = \inf_{\theta \in \mathbb{R}^q} \sup_{\theta'' \in [\theta_1, \theta_2)} d\left(\omega_{X_{\theta''}} \| \omega_{X_\theta}\right),$$

where $d(\cdot \| \cdot)$ is defined in Eq. (3). We also denote $\mathcal{D}^*(\theta_1, \theta_2) = \arg\inf_{\theta \in [\theta_1, \theta_2)} \sup_{\theta'' \in [\theta_1, \theta_2)} d\left(\omega_{X_{\theta''}} \| \omega_{X_\theta}\right)$. If the prior over parameters is $f_\Theta$, we have the Bellman equation

$$pri(t^*) = \min_{\theta \in [\underline{\theta}, t^* - \kappa], \mathcal{D}(\theta, t^*) \leq T} \frac{\int_{\underline{\theta}}^{\theta} f_\Theta(t)\, dt}{\int_{\underline{\theta}}^{t^*} f_\Theta(t)\, dt} \cdot pri(\theta) + \frac{\int_{\theta}^{t^*} f_\Theta(t)\, dt}{\int_{\underline{\theta}}^{t^*} f_\Theta(t)\, dt} \cdot \mathcal{P}(\theta, t^*)$$

with the initial state $pri(\underline{\theta}) = 0$, where

$$\mathcal{P}(\theta, t^*) = \mathbb{P}\left(\hat{g}^*(\theta') \in [g(\theta_0) - \epsilon, g(\theta_0) + \epsilon] \mid \theta_0 \in [\theta, t^*], \theta'\right)$$

$$= \sup_{t_1, t_2: \ \sup_{t', t'' \in [t_1, t_2]} |g(t'') - g(t')| = 2\epsilon} \frac{\int_{\max\{t_1, \theta\}}^{\min\{t_2, t^*\}} f_\Theta(t)\, dt}{\int_{\theta}^{t^*} f_\Theta(t)\, dt}.$$

$\theta'$ is the released parameter when the private parameter $\theta_0 \in [\theta, t^*]$ and $\hat{g}^*$ is the optimal attack strategy. The full algorithm is listed in Alg. 1. The time complexity of this algorithm is $\mathcal{O}\left(\left(\overline{\theta} - \underline{\theta}/\kappa\right)^2 \cdot \mathcal{C}_D \cdot \mathcal{C}_P \cdot \mathcal{C}_I\right)$, where $\mathcal{C}_D$ is the time complexity for computing $\mathcal{D}$ and $\mathcal{D}^*$, $\mathcal{C}_P$ is the time complexity for computing $\mathcal{P}$, and $\mathcal{C}_I$ is the time complexity for computing the integrals in the Bellman equation. In our cases studies, $\mathcal{D}$ and $\mathcal{D}^*$ can be computed in $\mathcal{C}_D = \mathcal{O}\left(\overline{\theta} - \underline{\theta}/\kappa\right)$, and $\mathcal{P}$ and the integrals can be computed in closed forms within constant time, i.e., $\mathcal{C}_P = \mathcal{C}_I = \mathcal{O}(1)$.

**Algorithm 1:** Dynamic-programming-based data release mechanism for single-parameter distributions.

**Input:** Parameter range: $\left[\underline{\theta}, \overline{\theta}\right)$
        Prior over parameter: $f_\Theta$
        Distortion budget: $T$
        Step size: $\kappa$ (which divides $\overline{\theta} - \underline{\theta}$)

**1** $pri(\underline{\theta}) \leftarrow 0$

**2** $\mathcal{I}\left(\underline{\theta}\right) \leftarrow \emptyset$

**3** **for** $t^* \leftarrow \underline{\theta} + \kappa, \underline{\theta} + 2\kappa, \ldots, \overline{\theta}$ **do**

**4**     $pri(t^*) \leftarrow \infty$

**5**     $min\_t \leftarrow \text{NULL}$

**6**     **for** $\theta \leftarrow t^* - \kappa, \ldots, \underline{\theta}$ **do**

**7**         **if** $\mathcal{D}\left(\theta, t^*\right) > T$ **then**

**8**             break

**9**         $p \leftarrow \dfrac{\int_{\underline{\theta}}^{\theta} f_\Theta(t)\mathrm{d}t}{\int_{\underline{\theta}}^{t^*} f_\Theta(t)\mathrm{d}t} \cdot pri\left(\theta\right) + \dfrac{\int_{\theta}^{t^*} f_\Theta(t)\mathrm{d}t}{\int_{\underline{\theta}}^{t^*} f_\Theta(t)\mathrm{d}t} \cdot \mathcal{P}\left(\theta, t^*\right)$

**10**         **if** $p < pri(t^*)$ **then**

**11**             $pri(t^*) \leftarrow p$

**12**             $min\_t \leftarrow \theta$

**13**     **if** $min\_t$ *is not NULL* **then**

**14**         $\mathcal{S}_{t^*} \leftarrow [min\_t, \ t^*)$

**15**         $\theta'_{t^*} \leftarrow \mathcal{D}^*\left(min\_t, t^*\right)$

**16**         $\mathcal{I}\left(t^*\right) \leftarrow \mathcal{I}\left(min\_t\right) \cup \{t^*\}$

**17** **if** $pri(\overline{\theta}) = \infty$ **then**

**18**     ERROR: No answer

**19** **return** $pri(\overline{\theta}), \left\{\mathcal{S}_i : i \in \mathcal{I}\left(\overline{\theta}\right)\right\}, \left\{\theta'_i : i \in \mathcal{I}\left(\overline{\theta}\right)\right\}$

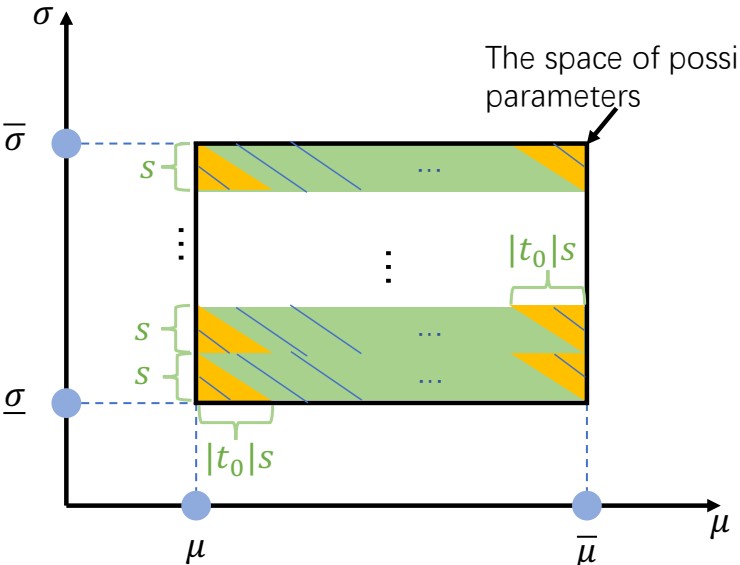

Figure 4: We separate the space of possible parameters into two regions (yellow and green) and bound the attacker's success rate on each region separately. The blue lines represent examples of $\mathcal{S}_{\mu,i}$.

When dynamic programming is not practical (e.g., in high-dimensional problems), we also provide a greedy algorithm in App. B as a baseline and show the empirical comparison between these two algorithms in the case studies (Apps. E, G and H).

### 5.3 Technique for Analyzing the Quantization Mechanism

We next provide an overview of techniques for analyzing the quantization mechanism, both for privacy and for distortion. We use these techniques for the analysis in our case studies, where we will make the expressions and claims more precise. For concreteness, we will recall the Gaussian example from §5.2, for which we have already derived a mechanism.

The mechanism presented in §5.2 can geometrically be interpreted as follows. Over the square of possible parameter values $\mu$ and $\sigma$ (Fig. 4), the mechanism selects intervals $\mathcal{S}_{\mu,i}$ that consist of short diagonal line segments (e.g., blue line segments in Fig. 4). When the true distribution parameters fall in one of these intervals, the mechanism releases the midpoint of the interval.

We find that many of our case studies naturally give rise to the same form of $t(\theta)$. As a result, all of the case studies we analyze theoretically (with multiple parameters) have mechanisms that instantiate intervals $\mathcal{S}_{\mu,i}$ as diagonal lines, as shown in Fig. 4. The sawtooth technique, which we present next, can be used to analyze the privacy of all such mechanism instantiations. More precisely, the following pattern of quantization mechanism admits diagonal line intervals, and can be analyzed with the sawtooth technique (§6 and Apps. E and G):

$$\mathcal{S}_{\mu,i} = \left\{ (\mu + t_0 \cdot t, \underline{\sigma} + (i + 0.5) \cdot s + t) \,|\, t \in \left[ -\frac{s}{2}, \frac{s}{2} \right) \right\} \quad,$$
$$\theta_{\mu,i}^* = (\mu, \underline{\sigma} + (i + 0.5) \cdot s) \quad,$$
$$\mathcal{I} = \left\{ (\mu, i) \,|\, i \in \mathbb{N}, \mu \in \mathbb{R} \right\},$$

where $s$ is a hyper-parameter of the mechanism that denotes quantization bin size and divides $(\overline{\sigma} - \underline{\sigma})$ and $t_0$ is a constant that can be determined by the mechanism design strategy described in §5.2.

**(1) Privacy analysis.** For ease of illustration, we assume that the support of parameters is $\mathrm{Supp}\,(\Theta) = \left\{ (a,b) \,|\, a \in \left[ \underline{\mu}, \overline{\mu} \right), b \in [\underline{\sigma}, \overline{\sigma}) \right\}$, but the analysis can be generalized to any case.

In Fig. 4, we separate the space of possible data parameters into two regions represented by yellow and green colors. The yellow regions $S_{yellow}$ constitute right triangles with height $s$ and width $|t_0|s$. The green

region $S_{green}$ is the rest of the parameter space. The high-level idea of our proof is as follows. Note that for any parameter $\theta \in S_{green}$, there exists a quantization bin $\mathcal{S}_{\mu,i}$ s.t. $\theta \in \mathcal{S}_{\mu,i}$ and $\mathcal{S}_{\mu,i} \subset S_{green}$. This occurs because the mechanism intervals (blue lines in Fig. 4) all have the same slope and a length of at most $s$ for $\sigma$. As such, each interval is either fully in the green region, or fully in the yellow region. Since we know the length of each bin, we can upper bound the attack success rate if $\theta \in S_{green}$. While the attacker can be more successful in the yellow region, the probability of $\theta \in S_{yellow}$ is small. Hence, we upper bound the overall attacker's success rate (i.e., $\Pi_{\epsilon,\omega_\Theta}$). More specifically, let the optimal attacker be $\hat{g}^*$. We have

$$\begin{aligned}
\Pi_{\epsilon,\omega_\Theta} &= \mathbb{P}\left(\hat{g}^*\left(\theta'\right) \in [g\left(\theta\right) - \epsilon, g\left(\theta\right) + \epsilon]\right) \\
&= \int_{\theta \in S_{green}} p(\theta)\mathbb{P}\left(\hat{g}^*\left(\theta'\right) \in [g\left(\theta\right) - \epsilon, g\left(\theta\right) + \epsilon]\right) d\theta \\
&\quad + \int_{\theta \in S_{yellow}} p(\theta)\mathbb{P}\left(\hat{g}^*\left(\theta'\right) \in [g\left(\theta\right) - \epsilon, g\left(\theta\right) + \epsilon]\right) d\theta \\
&< \sup_{\theta \in S_{green}} \mathbb{P}\left(\hat{g}^*\left(\theta'\right) \in [g\left(\theta\right) - \epsilon, g\left(\theta\right) + \epsilon]\right) + \int_{\theta \in S_{yellow}} p(\theta)d\theta
\end{aligned}$$

The first term can be bounded away from 1 due to the carefully chosen $t_0$. The second term is bounded away from 1 because the size of $S_{yellow}$ is relatively small. The formal justification is given in Prop. 2 and Apps. C.4.2, F.2 and G.4.

**(2) Distortion analysis.** For the distortion performance, it is straightforward to show that $\Delta = \sup_{\theta \in \mathrm{Supp}(\Theta)} d\left(\omega_{X_\theta} \| \omega_{X_{\theta^*_{I(\theta)}}}\right)$, where $\theta^*_{I(\theta)}$ is the released parameter when the original parameter is $\theta$. This quantity can often be derived directly from the mechanism and parameter support.

## 6   Case Studies

In this section, we instantiate the general results on concrete distributions and secrets (mean §6.1, quantile §6.2, and we defer standard deviation and discrete distribution fractions to Apps. G and H). See Table 1 for a summary of each setting we consider, and a pointer to any theoretical results. Our results in each setting generally include a privacy lower bound, a concrete instantiation of the quantization mechanism, and privacy-distortion analysis of the data release mechanisms. In §6.3, we will discuss how to extend the data release mechanisms to the cases when data holders only have data samples and do not know the parameters of the underlying distributions.

These data release mechanisms serve as the initial version of *summary statistic privacy toolbox* (Fig. 1).

Table 1: Summary of the case studies.

| Distribution / Secret | Continuous Distribution (order-optimal mechanism) | | | Ordinal Distribution (Alg. 1 and Alg. 3) | | | Categorical Distribution (order-optimal mechanism) |
|---|---|---|---|---|---|---|---|
| | Gaussian | Uniform | Exponential | Geometric | Binomial | Poisson | |
| Mean | §6.1 | | | App. E | | | Not applicable |
| Quantile | §6.2 and App. F | | | Not applicable | | | Not applicable |
| Standard Deviation | App. G.1 | | | App. G.2 | | | Not applicable |
| Fraction | Not applicable | | | App. H.1 | | | App. H.2 |

### 6.1   Secret = Mean

In this section, we discuss how to protect the mean of a distribution for general continuous distributions. We start with a lower bound.

**Corollary 1** (Privacy lower bound, secret = mean of a continuous distribution)**.** *Consider the secret function* $g\left(\theta\right) = \int_x x f_{X_\theta}\left(x\right) dx$. *For any* $T \in (0,1)$, *when* $\Pi_{\epsilon,\omega_\Theta} \leq T$, *we have* $\Delta > \left(\lceil \frac{1}{T} \rceil - 1\right) \cdot \epsilon$.

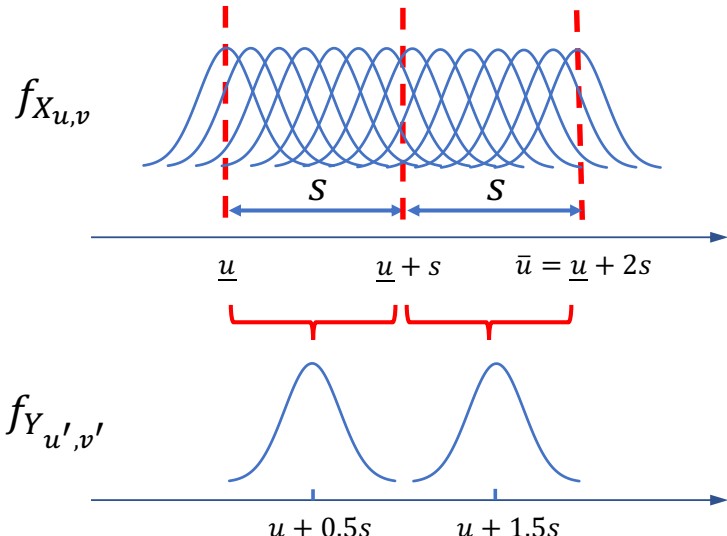

Figure 5: Illustration of the data release mechanism for continuous distributions when secret=mean.

The proof is in App. C.1. We next design a data release mechanism that achieves a tradeoff close to this bound.

**Data release mechanism.** We first consider continuous distributions that can be parameterized with a location parameter, where the prior distribution of the location parameter is uniform and independent of other factors. We relax this assumption to Lipschitz-continuous priors in App. D.1. For now, we assume the following:

**Assumption 1.** *The distribution parameter vector $\theta$ can be written as $(u, v)$, where $u \in \mathbb{R}$, $v \in \mathbb{R}^{q-1}$, and for any $u \neq u'$, $f_{X_{u,v}}(x) = f_{X_{u',v}}(x - u' + u)$. The prior over distribution parameters is $f_{U,V}(a, b) = f_U(a) \cdot f_V(b)$, where $f_U(a) = \frac{1}{\overline{u} - \underline{u}} \mathbb{I}(a \in [\underline{u}, \overline{u}])$.*

Examples include the Gaussian, Laplace, and uniform distributions, as well as shifted distributions (e.g., shifted exponential, shifted log-logistic). Using the strategy from §5.2, we derive the following quantization mechanism.

**Mechanism 1** (For secret = mean of a continuous distribution)**.** *The parameters of the data release mechanism are*

$$\mathcal{S}_{i,v} = \{(t, v) \,|\, t \in [\underline{u} + i \cdot s, \ \underline{u} + (i+1) \cdot s)\}, \tag{13}$$

$$\theta^*_{i,v} = (\underline{u} + (i + 0.5) \cdot s, v), \tag{14}$$

$$\mathcal{I} = \{(i, v) : i \in \{0, 1, \ldots, N - 1\}, v \in Supp(\omega_V)\}, \tag{15}$$

*where $s$ is a hyper-parameter of the mechanism that divides $(\overline{u} - \underline{u})$ and $N = \frac{\overline{u} - \underline{u}}{s} \in \mathbb{N}$.*

Fig. 5 shows an example when the original data distribution is Gaussian, i.e., $X_\theta \sim \mathcal{N}(u, v)$, and $u \in [\underline{\mu}, \overline{\mu}]$. Intuitively, our data release mechanism "quantizes" the range of possible mean values into segments of length $s$. It then shifts the mean of private distribution $f_{X_{u,v}}$ to the midpoint of its corresponding segment, and releases the resulting distribution. This simple deterministic mechanism is able to achieve order-optimal privacy-distortion tradeoff in some cases, as shown below.

**Proposition 1.** *Under Asm. 1, Mech. 1 has $\Pi_{\epsilon, \omega_\Theta} \leq \frac{2\epsilon}{s}$ and $\Delta = \frac{s}{2} < 2\Delta_{opt}$, where $\Delta_{opt}$ is the minimal distortion an optimal data release mechanism can achieve given the privacy Mech. 1 achieves.*

The proof is in App. C.2. The two takeaways from this proposition are that: (1) the data holder can use $s$ to control the trade-off between distortion and privacy, and (2) the mechanism is order-optimal with multiplicative factor 2.

## 6.2  Secret = Quantiles

S3 in §2.1 explains how quantiles of continuous distributions can reveal sensitive information. In this section, we show how to protect it for a typical continuous distribution: the (shifted) exponential distribution. We analyze the Gaussian and uniform distributions in App. F. We choose these distributions as a starting point of our analysis as many distributions in real-world data can be approximated by one of these distributions.

In our analysis, the parameters of (shifted) exponential distributions are denoted by:

- Exponential distribution: $\theta = \lambda$, where $\lambda$ is the scale parameter. In other words, $f_{X_\lambda}(x) = \frac{1}{\lambda} e^{-x/\lambda}$.
- Shifted exponential distribution generalizes the exponential distribution with an additional shift parameter $h$: $\theta = (\lambda, h)$. In other words, $f_{X_{\lambda,h}}(x) = \frac{1}{\lambda} e^{-(x-h)/\lambda}$.

As before, we first present a lower bound.

**Corollary 2** (Privacy lower bound, secret = $\alpha$-quantile of a continuous distribution). *Consider the secret function $g(\theta) = \alpha$-quantile of $f_{X_\theta}$. For any $T \in (0,1)$, when $\Pi_{\epsilon,\omega_\Theta} \leq T$, we have $\Delta > \left( \lceil \frac{1}{T} \rceil - 1 \right) \cdot 2\gamma\epsilon$, where $\gamma$ is defined as follows:*

- *Exponential:*

$$\gamma = -\frac{1}{2\ln(1-\alpha)}.$$

- *Shifted exponential:*

$$\gamma = \begin{cases} \frac{1}{2} \left| 1 + \frac{\ln(1-\alpha)+1}{W_{-1}\left(-\frac{\ln(1-\alpha)+1}{2(1-\alpha)e}\right)} \right| & \alpha \in [0, 1-e^{-1}) \\ \frac{1}{2} \left| 1 + \frac{\ln(1-\alpha)+1}{W_0\left(-\frac{\ln(1-\alpha)+1}{2(1-\alpha)e}\right)} \right| & \alpha \in [1-e^{-1}, 1) \end{cases},$$

*where $W_{-1}$ and $W_0$ are Lambert W functions.*

The proof is in App. C.3. Next, we provide data release mechanisms for each of the distributions that achieve trade-offs close to these bounds.

**Mechanism 2** (For secret = quantile of a continuous distribution). *We design mechanisms for each of the distributions. In both cases, $s > 0$ is the quantization bin size chosen by the operator to divide $(\overline{\lambda} - \underline{\lambda})$, where $\overline{\lambda}$ and $\underline{\lambda}$ are upper and lower bounds of $\lambda$.*

- *Exponential:*

$$\begin{aligned} \mathcal{S}_i &= [\underline{\lambda} + i \cdot s, \underline{\lambda} + (i+1) \cdot s) \quad, \\ \theta_i^* &= \underline{\lambda} + (i+0.5) \cdot s \quad, \\ \mathcal{I} &= \mathbb{N}. \end{aligned}$$

- *Shifted exponential:*

$$\begin{aligned} \mathcal{S}_{i,h} &= \left\{ (\underline{\lambda} + (i+0.5)s + t, h - t_0 \cdot t) \, | t \in \left[ -\frac{s}{2}, \frac{s}{2} \right) \right\} \quad, \\ \theta_{i,h}^* &= (\underline{\lambda} + (i+0.5)s, h) \quad, \\ \mathcal{I} &= \{ (i,h) | i \in \mathbb{N}, h \in \mathbb{R} \}, \end{aligned}$$

*where*

$$t_0 = \begin{cases} -1 - \ln(1-\alpha) - W_{-1}\left(-\frac{\ln(1-\alpha)+1}{2(1-\alpha)e}\right) & (\alpha \in [0, 1-e^{-1})) \\ -1 - \ln(1-\alpha) - W_0\left(-\frac{\ln(1-\alpha)+1}{2(1-\alpha)e}\right) & (\alpha \in [1-e^{-1}, 1)) \end{cases}.$$

For the privacy-distortion trade-off analysis of Mech. 2, we assume that the parameters of the original data are drawn from a uniform distribution with lower and upper bounds. Again, we relax this assumption to Lipschitz priors in App. D.2. Precisely,

**Assumption 2.** *The prior over distribution parameters is:*

- *Exponential: $\lambda$ follows the uniform distribution over $\left[\underline{\lambda}, \overline{\lambda}\right)$.*

- *Shifted exponential: $(\lambda, h)$ follows the uniform distribution over $\left\{(a,b) | a \in \left[\underline{\lambda}, \overline{\lambda}\right), b \in \left[\underline{h}, \overline{h}\right]\right\}$.*

We relax Asm. 2 and analyze the privacy-distortion trade-off of Mech. 2 in App. D.2.

**Proposition 2.** *Under Asm. 2, Mech. 2 has the following $\Pi_{\epsilon,\omega_\Theta}$ and $\Delta$ value/bound.*

- *Exponential:*

$$\Pi_{\epsilon,\omega_\Theta} = \frac{2\epsilon}{-\ln\left(1-\alpha\right)s}, \qquad \Delta = \frac{1}{2}s < 2\Delta_{opt}.$$

- *Shifted exponential:*

$$\Pi_{\epsilon,\omega_\Theta} < \frac{2\epsilon}{|\ln\left(1-\alpha\right) + t_0|s} + \frac{|t_0|s}{\overline{h} - \underline{h}},$$

$$\Delta = \frac{s}{2}\left(t_0 - 1\right) + se^{-t_0} < \left(2 + \frac{|t_0| \cdot |\ln\left(1-\alpha\right) + t_0|s^2}{\epsilon\left(\overline{h} - \underline{h}\right)}\right)\Delta_{opt}.$$

*Under the high-precision regime where $\frac{s^2}{\overline{h}-\underline{h}} \to 0$ as $s, (\overline{h} - \underline{h}) \to \infty$, when $\alpha \in [0.01, 0.25] \cup [0.75, 0.99]$, $\Delta$ satisfies*

$$\lim_{\frac{s^2}{\overline{h}-\underline{h}} \to 0} \sup \Delta < 3\Delta_{opt}.$$

$\Delta_{opt}$ *is the optimal achievable distortion given the privacy achieved by Mech. 2, and $t_0$ is a constant defined in Mech. 2.*

The proof is in App. C.4. Note that the quantization bin size $s$ cannot be too small, or the attacker can always successfully guess the secret within a tolerance $\epsilon$ (i.e., $\Pi_{\epsilon,\omega_\Theta} = 1$). Therefore, for the "high-precision" regime, we consider the asymptotic scaling as both $s$ and $\overline{h} - \underline{h}$ grow.

Prop. 2 shows that the quantization mechanism is order-optimal with multiplicative factor 2 for the exponential distribution. For shifted exponential distribution, order-optimality holds asymptotically in the high-precision regime.

### 6.3 Extending Data Release Mechanisms for Dataset Input/Output

The data release mechanisms discussed in previous sections assume that data holders know the *distribution parameter* of the original data. In practice, data holders often only have a dataset of samples from the data distribution and do not know the parameters of the underlying distributions. As mentioned in §3, our data release mechanisms can be easily adapted to handle dataset input/output.

The high-level idea is that the data holders can estimate the distribution parameters $\theta$ from the data samples and find the corresponding quantization bins $\mathcal{S}_i$ according to the estimated parameters, and then modify the original samples as if they are sampled according to the released parameter $\theta_i^*$. For brevity, we only present the concrete procedure for secret=mean on continuous distributions as an example. For a dataset of $\mathcal{X} = \{x_1, \ldots, x_n\}$, the procedure is:

1. Estimate the mean from the data samples: $\hat{\mu} = \frac{1}{n}\sum_{i\in[n]} x_i$.

2. According to Eq. (13), compute the index of the corresponding set $i = \lfloor\frac{\hat{\mu}-\underline{\mu}}{s}\rfloor$.

3. According to [Eq. (14)](#), change the mean of the data samples to $\mu_{target} = \underline{\mu} + (i + 0.5) \cdot s$. This can be done by sample-wise operation $x'_i = x_i - \hat{\mu} + \mu_{target}$.

4. The released dataset is $\mathcal{M}_g(\mathcal{X}, z) = \{x'_1, \ldots, x'_n\}$.

Note that this mechanism applies to samples. Therefore, it can be applied either to the original data, or as an add-on to existing data sharing tools (Esteban et al., 2017; Lin et al., 2020; Yin et al., 2022; Jordon et al., 2018; Yoon et al., 2019). For example, it can be used to modify synthetically-generated samples after they are generated, or to modify the training dataset for a generative model, or to directly modify the original data for releasing.

# 7 Experiments

In the previous sections, we theoretically demonstrated the privacy-distortion tradeoffs of our data release mechanisms in some special case studies. In this section, we focus on *orthogonal* questions through real-world experiments: (1) how well our data release mechanisms perform when the assumptions do not hold in practice, and (2) why existing privacy frameworks are not suitable for summary statistic privacy (which we explained qualitatively in §2.2).

**Datasets.** We use three real-world datasets to simulate each of the motivating scenarios in §2.1.

1. Wikipedia Web Traffic Dataset (WWT) (Google, 2018) contains the daily page views of 145,063 Wikipedia web pages in 2015-2016. To preprocess it for our experiments, we remove the web pages with empty page view record on any day (117,277 left), and compute the mean page views across all dates for each web page. Our goal is to release the page views (i.e., a 117,277-dimensional vector) while protecting the **mean of the distribution** (which reveals the business scales of the company §2.1).

2. Google Cluster Trace Dataset (GCT) (Reiss et al., 2011) contains usage logs (e.g., CPU/memory) of an internal Google cluster with 12.5k machines in 2011. We use "platform ID" field of the dataset, which represents "microarchitecture and chipset version of the machine" (Reiss et al., 2011). Our goal is to release another distribution of platform ID while protecting the **fraction of one specific platform ID** (which reveals business strategy §2.1).

3. Measuring Broadband America Dataset (MBA) (Commission, 2018) contains network statistics (including network traffic counters) collected by United States Federal Communications Commission from homes across United States. We select the average network traffic (GB/measurement) from AT&T clients as our data. Our goal is to release a copy of this data while hiding the **0.95-quantile** (which reveals the network capability §2.1).

**Baselines.** We compare our mechanisms discussed in §6 with three popular mechanisms proposed in prior work (§2.2): differentially-private density estimation (Wasserman & Zhou, 2010) (shortened to DP), attribute-private Gaussian mechanism (Zhang et al., 2022) (shortened to AP), and Wasserstein mechanism for distribution privacy (Chen & Ohrimenko, 2022) (shortened to DistP). Note that these mechanisms are not designed for our problem setting—in that sense, these experiments are not a fair comparison. Nontheless, we include them simply to illustrate that prior techniques are not sufficient *for our problem setting*.

For a dataset of samples $\mathcal{X} = \{x_1, ..., x_n\}$, DP works by: (1) Dividing the space into $m$ bins: $B_1, ..., B_m$.[4] (2) Computing the histogram $C_i = \sum_{j=1}^n \mathbb{I}(x_j \in B_i)$. (3) Adding noise to the histograms $D_i = \max\{0, C_i + \text{Laplace}(0, \epsilon^2)\}$, where $\text{Laplace}(0, \epsilon^2)$ means a random noise from Laplace distribution with mean 0 and variance $\epsilon^2$. (4) Normalizing the histogram $p_i = \frac{D_i}{\sum_{j=1}^m D_j}$. We can then draw $y_i$ according to the histogram and release $\mathcal{Y} = \{y_1, ..., y_n\}$ with differential privacy guarantees. AP works by releasing

---

[4]In Google Cluster Trace Dataset, the bin is already pre-specified (i.e., the platform IDs), so this step is skipped.

$\mathcal{Y} = \left\{x_i + \mathcal{N}\left(0, \epsilon^2\right)\right\}_{i=1}^{n}.$[5] DistP works by releasing $\mathcal{Y} = \left\{x_i + \text{Laplace}\left(0, \epsilon^2\right)\right\}_{i=1}^{n}.$[6] Note that for each of these mechanisms, normally their noise parameters would be set carefully to match the desired privacy guarantees (e.g., differential privacy). In our case, since our privacy metric is different, it is unclear how to set the noise parameters for a fair privacy comparison. For this reason, we evaluate different settings of the noise parameters, and measure the empirical tradeoffs.

**Metrics.** Our privacy and distortion metrics depend on the prior distribution of the original data $\theta \sim \omega_{\Theta}$ (though the mechanism does not). In practice (and also in these experiments), the data holder only has one dataset. Therefore, we cannot empirically evaluate the proposed privacy and distortion metrics, and resort to surrogate metrics to bound our true privacy and distortion.

*Surrogate privacy metric.* For an original dataset $\mathcal{X} = \{x_1, ..., x_n\}$ and the released dataset $\mathcal{Y} = \{y_1, ..., y_n\}$, we define the surrogate privacy metric $\tilde{\Pi}_{\epsilon}$ as the error of an attacker who guesses the secret of the released dataset as the true secret: $\tilde{\Pi}_{\epsilon, \omega_{\Theta}} \triangleq -|g(\mathcal{X}) - g(\mathcal{Y})|$, where $g(\mathcal{D}) = $ mean of $\mathcal{D}$, fraction of a specific platform ID in $\mathcal{D}$, and 0.95-quantile of $\mathcal{D}$ in WWT, GCT, and MBA datasets respectively. Note that in the definition of $\tilde{\Pi}_{\epsilon, \omega_{\Theta}}$, a minus sign is added so that a smaller value indicates stronger privacy, as in privacy metric Eq. (2). This simple attacker strategy is in fact a good proxy for evaluating the privacy $\Pi_{\epsilon, \omega_{\Theta}}$ due to the following facts. (1) For our data release mechanisms for these secrets Mechs. 1, 2 and 5, when the prior distribution is uniform, this strategy is actually optimal, so there is a direct mapping between $\tilde{\Pi}_{\epsilon}$ and $\Pi_{\epsilon, \omega_{\Theta}}$. (2) For AP applied on protecting mean of the data (i.e., Wikipedia Web Traffic Dataset experiments), this strategy gives an unbiased estimator of the secret. (3) For DP and AP on other cases, this mechanism may not be an unbiased estimator of the secret, but it gives an *upper bound* on the attacker's error.

*Surrogate distortion metric.* We define our surrogate distortion metric as the distance between the two datasets: $\tilde{\Delta} \triangleq d(p_{\mathcal{X}} \| p_{\mathcal{Y}})$ where $p_D$ denotes the empirical distribution of a dataset $D$, and $d$ is defined as in our formulation §3 (i.e., Wassersstein-1 distance for continuous distributions in WWT and MBA, and TV distance for discrete distributions in GCT). This metric evaluates how much the mechanism distorts the dataset.

In fact, we can deduce a theoretical lower bound for the surrogate privacy and distortion metrics for secret = mean/fractions (shown later in Fig. 6) using similar techniques as the proofs in the main paper (see App. C.5).

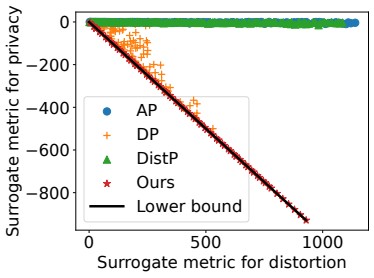

(a) Wikipedia Web Traffic Dataset. (secret=mean)

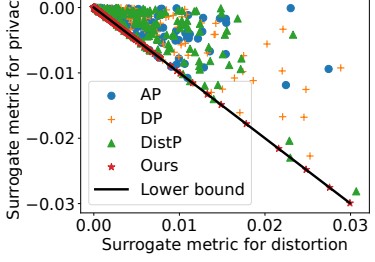

(b) Google Cluster Trace Dataset. (secret=categorical fraction)

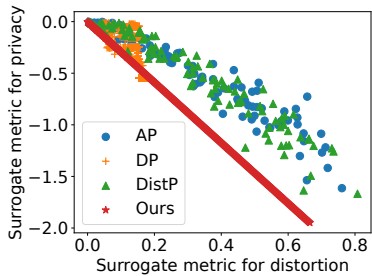

(c) Measuring Broadband America Dataset. (secret=quantile)

Figure 6: Privacy (lower is better) and distortion (lower is better) of AP, DP, DistP, and ours. Each point represents one instance of data release mechanism with one hyper-parameter. "Lower bound" is the theoretical lower bound of the achievable region. Our data release mechanisms achieve better privacy-distortion tradeoff than AP, DP, and DistP.

---

[5] In Google Cluster Trace Dataset, the Gaussian noise $\mathcal{N}\left(0, \epsilon^2\right)$ are added to the counts of different platform IDs. We then normalize the counts and sample released platform IDs from this categorical distribution.

[6] In Google Cluster Trace Dataset, the Laplace noise $\text{Laplace}\left(0, \epsilon^2\right)$ are added to the counts of different platform IDs. We then normalize the counts and sample released platform IDs from this categorical distribution.

## 7.1 Results

We enumerate the hyper-parameters of each method (bin size and $\epsilon$ for DP, $\epsilon$ for AP and DistP, and $s$ for ours). For each method and each hyper-parameter, we compute their surrogate privacy and distortion metrics. The results are shown in Fig. 6 (bottom left is best); each data point represents one realization of mechanism $\mathcal{M}_g$ under a distinct hyperparameter setting. Two takeaways are below.

*(1) Our data release mechanisms has good privacy-distortion trade-offs even when the assumptions do not hold.* We envision that data holders can choose the data release mechanisms in the toolbox (Fig. 1) that matches their need. However, in practical scenarios, the data distributions supported in the toolbox may not always match real data exactly. Our data release mechanisms for mean (i.e., Mech. 1 used in WWT) and fractions (i.e., Mech. 5 used in GCT) support general continuous distributions and categorical distributions, and therefore, there is no such a distribution gap. Indeed, even for these surrogate metrics, our Mech. 1 and Mech. 5 are also optimal (see App. C.5). This is visualized in Figs. 6a and 6b where we can see that our data release mechanisms match the theoretical lower bound of the trade-off. However, our data release mechanisms for quantiles (i.e., Mech. 2 used in Fig. 6c) are order-optimal only when the distributions are within certain classes (§6.2). Observing that network traffic in MBA follows a one-side fat-tailed distribution (not shown), we apply the data release mechanism for exponential distribution (Mech. 2) for this dataset. Despite the distribution mismatch, our data release mechanism still achieves a good privacy-distortion compared to DP, AP, and DistP (Fig. 6c). More discussions are below.

*(2) Our data release mechanisms achieve better privacy-distortion trade-off than DP, AP, and DistP.* AP and DistP directly add Gaussian/Laplace noise to each sample. This process does not change the mean of the distribution on expectation. Therefore, Figure 6 shows that AP and DistP have a bad privacy-distortion tradeoff. DP quantizes (bins) the samples before adding noise. Quantization has a better property in terms of protecting the mean of the distribution, and therefore we see that DP has a better privacy-distortion tradeoff than AP and DistP, but still worse than ours. Note that in Fig. 6c, a few of the DP instances have better privacy-distortion trade-offs than ours. This is *not* an indication that DP is fundamentally better. Instead, it is due to the randomness in DP (from the added Laplace noise), and some realizations of the specific noise in this experiment happened to lead to a better trade-off. Another instance of the DP algorithm could lead to a bad trade-off, and therefore, DP's achievable trade-off points are widespread.

**In summary, these results confirm our intuition in §2.2 that DP, AP, and DistP are not suitable for summary statistic privacy (which is expected—they are designed for a different objective). As such, the quantization mechanism (under the summary statistic privacy framework) gives better practical protections for summary statistic privacy.** Additional results on downstream tasks are in App. I.

# 8 Limitations

This work has several important limitations, some of which relate to the framework itself, others of which are specific to the mechanisms and results we prove. We outline several of these limitations.

## 8.1 Limitations of the Framework

**Prior knowledge of distribution.** The current privacy metric $\Pi_{\epsilon, \omega_\Theta}$ depends on the prior distribution of the parameters $\omega_\Theta$, which is typically unknown. The outcome is that if a mechanism is analyzed under a mismatched prior, it may lead a data holder to over- or under-estimate their privacy parameter.

**Composition guarantees.** Another limitation of the current privacy metric $\Pi_{\epsilon, \omega_\Theta}$ is that it does not provide composition guarantees; in other words, if one applies a summary statistic-private mechanism $\upsilon$ times, we cannot easily bound the privacy parameter of the $\upsilon$-fold composed mechanism. In contrast, composition is an important and desirable property exhibited by differential privacy (Dwork et al., 2006). The lack of composition can be problematic in situations where a data holder wants to release a dataset (or correlated datasets) multiple times.

### 8.2 Limitations of the Analysis and Mechanisms

Our analysis in this work considers the simplest set of cases, which are neither fully representative of how real data users release data, nor the secrets they wish to hide.

**Number of secrets.** In this work, we studied a case where the data holder only wishes to hide a single secret. In practice, data holders often want to hide multiple properties of their underlying data.

**The dimension and the type of data distributions.** Although our lower bounds in Section 4 apply to general prior distributions, we analyze the quantization mechanism under a limited set of one-dimensional distributions (Table 1) under which different parameters of the distribution are drawn independently of each other. An interesting direction for future work is to define mechanisms that have good tradeoffs under prior distributions with correlated parameters and priors.

## 9 Discussion and Future Work

We introduce *summary statistic privacy* for defining, analyzing, and protecting summary statistic privacy concerns in data sharing applications. This framework can be used to analyze the leakage of statistical information and the privacy-distortion trade-offs of data release mechanisms (§ 3 and 4). Our data release mechanisms can be used to protect statistical information (§ 5 and 6). However, as discussed in §8, this paper leaves many questions unanswered. Several of these pose interesting questions for future work.

**Approximation error.** We studied a number of data distributions and prior distributions in this work. However, an interesting question is to bound the error in privacy and distortion metrics as a function of approximation error when describing either the original data distribution or the prior.

**Extensions.** As described in §8, one limitation of the current privacy metric $\Pi_{\epsilon,\omega_\Theta}$ is that it depends on the prior distribution of the parameters $\omega_\Theta$, which is unknown in many applications. Motivated by maximal leakage (Issa et al., 2019) (§2.2), one possibility is to consider a *normalized* privacy metric:

$$\Pi'_{\epsilon,\omega_\Theta} \triangleq \sup_{\omega_\Theta} \ \log \frac{\Pi_{\epsilon,\omega_\Theta}}{\sup_{\hat{g}} \ \mathbb{P}\left(\hat{g}\left(\omega_\Theta\right) \in \left[g\left(\theta\right) - \epsilon, g\left(\theta\right) + \epsilon\right]\right)},$$

where $\hat{g}\left(\omega_\Theta\right)$ is an attacker that knows the prior distribution but does not see the released data, and the denominator is the probability that the strongest attacker guesses the secret within tolerance $\epsilon$. Similar to maximal leakage, we consider the worst-case leakage among all possible priors. This *normalized* $\Pi'_{\epsilon,\omega_\Theta}$ considers how much additional "information" that the released data provides to the attacker in the worst-case (see also inferential privacy (Ghosh & Kleinberg, 2016)). This privacy definition is strong so that we will not be able to achieve good privacy and reasonable distortion at the same time.

**Proposition 3.** *Let $\overline{\Delta} \triangleq \frac{1}{2}\sup_{\theta_1,\theta_2 \in Supp(\omega_\Theta)} d\left(\omega_{X_{\theta_1}}\|\omega_{X_{\theta_2}}\right)$. There exists no $\mathcal{M}_g$ such that $\Pi'_{\epsilon,\omega_\Theta} < \log 2$ and $\Delta < \overline{\Delta}$.*

The proof is in App. C.6. It would be interesting to further study the feasibility of such a formulation, for instance by changing the utility metric to an expected distortion, rather than a worst-case one.

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

## Appendix

## A   Analysis of the Alternative Formulation

In this section, we present the alternative formulation of minimizing privacy metric $\Pi_{\epsilon,\omega_\Theta}$ subject to a constraint on distortion $\Delta$:

$$\min_{\mathcal{M}_g} \ \Pi_{\epsilon,\omega_\Theta} \qquad \text{subject to} \ \ \Delta \leq T \tag{16}$$

**Theorem 2** (Lower bound of privacy-distortion tradeoff). *Let* $D\left(X_{\theta_1}, X_{\theta_2}\right) \triangleq \frac{1}{2} d\left(\omega_{X_{\theta_1}} \| \omega_{X_{\theta_2}}\right)$, *where* $d\left(\cdot \| \cdot\right)$ *is defined in Eq. (3). Further, let* $R\left(X_{\theta_1}, X_{\theta_2}\right) \triangleq |g(\theta_1) - g(\theta_2)|$, *and let* $\gamma \triangleq \inf_{\theta_1, \theta_2 \in Supp(\omega_\Theta)} \frac{D\left(X_{\theta_1}, X_{\theta_2}\right)}{R\left(X_{\theta_1}, X_{\theta_2}\right)}$. *For any* $T > 0$, *when* $\Delta \leq T$, *we have* $\Pi_{\epsilon,\omega_\Theta} \geq \lceil \frac{T}{2\gamma\epsilon} \rceil^{-1}$.

*Proof.* For any $\theta'$, we have

$$\begin{aligned}
T &\geq \Delta \\
&\geq \sup_{\theta \in \mathrm{Supp}(\omega_\Theta), z \in \mathrm{Supp}(\omega_Z):\mathcal{M}_g(\theta,z)=\theta'} d\left(\omega_{X_\theta} \| \omega_{X_{\theta'}}\right) \\
&\geq \sup_{\theta_i \in \mathrm{Supp}(\omega_\Theta), z_i:\mathcal{M}_g(\theta_i,z_i)=\theta'} D\left(X_{\theta_1}, X_{\theta_2}\right) \\
&\geq \gamma \cdot \sup_{\theta_i \in \mathrm{Supp}(\omega_\Theta), z_i:\mathcal{M}_g(\theta_i,z_i)=\theta'} R\left(X_{\theta_1}, X_{\theta_2}\right)
\end{aligned} \tag{17}$$

where Eq. (17) comes from triangle inequality.

Let

$$L_{\theta'} \triangleq \inf_{\theta \in \mathrm{Supp}(\omega_\Theta), z:\mathcal{M}_g(\theta,z)=\theta'} g\left(\theta\right) \ ,$$

$$R_{\theta'} \triangleq \sup_{\theta \in \mathrm{Supp}(\omega_\Theta), z:\mathcal{M}_g(\theta,z)=\theta'} g\left(\theta\right) \ .$$

From the above result, we know that $R_{\theta'} - L_{\theta'} \leq \frac{T}{\gamma}$. We can define a sequence of attackers such that $\hat{g}_i\left(\theta'\right) = L_{\theta'} + (i + 0.5) \cdot 2\epsilon$ for $i \in \left\{0, 1, \ldots, \lceil \frac{T}{2\gamma\epsilon} \rceil - 1\right\}$ (Fig. 7). We have

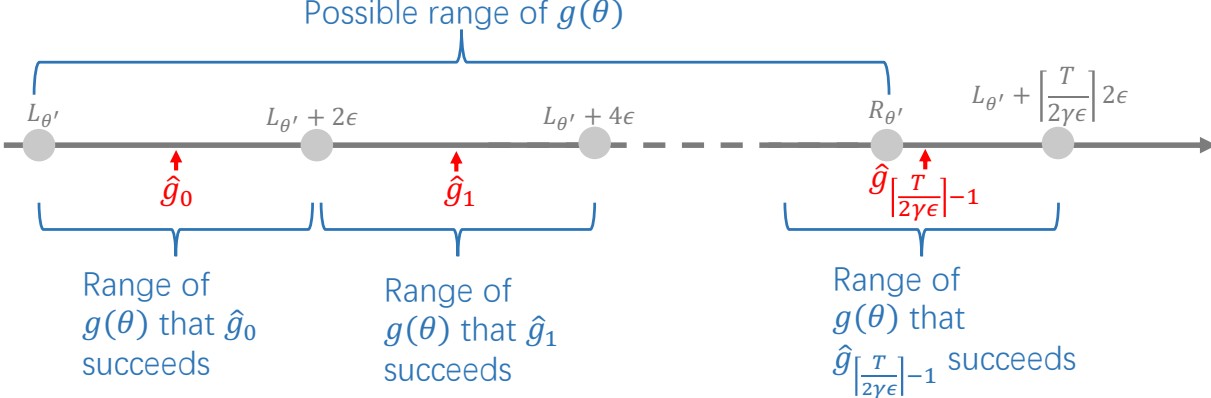

Figure 7: The construction of attackers for proof of Thm. 2. The $2\epsilon$ ranges of $\hat{g}_0, ..., \hat{g}_{\lceil \frac{T}{2\gamma\epsilon} \rceil - 1}$ jointly cover the entire range of possible secret $[L_{\theta'}, R_{\theta'}]$. Therefore, there exists one attacker whose probability of guessing the secret correctly within $\epsilon$ is $\geq \lceil \frac{T}{2\gamma\epsilon} \rceil^{-1}$ (Eq. (18)).

$$\sum_i \mathbb{P}\left(\hat{g}_i\left(\theta'\right) \in \left[g\left(\theta\right) - \epsilon, g\left(\theta\right) + \epsilon\right] \middle| \theta'\right) \geq 1,$$

and therefore,

$$\max_i \mathbb{P}\left(\hat{g}_i\left(\theta'\right) \in \left[g\left(\theta\right) - \epsilon, g\left(\theta\right) + \epsilon\right] \middle| \theta'\right) \geq \lceil\frac{T}{2\gamma\epsilon}\rceil^{-1}, \tag{18}$$

which implies that

$$\sup_{\hat{g}} \mathbb{P}\left(\hat{g}\left(\theta'\right) \in \left[g\left(\theta\right) - \epsilon, g\left(\theta\right) + \epsilon\right] \middle| \theta'\right) \geq \lceil\frac{T}{2\gamma\epsilon}\rceil^{-1}.$$

Therefore, we have

$$\Pi_{\epsilon,\omega_\Theta} = \sup_{\hat{g}} \mathbb{P}\left(\hat{g}\left(\theta'\right) \in \left[g\left(\theta\right) - \epsilon, g\left(\theta\right) + \epsilon\right]\right)$$

$$= \sup_{\hat{g}} \mathbb{E}\left(\mathbb{P}\left(\hat{g}\left(\theta'\right) \in \left[g\left(\theta\right) - \epsilon, g\left(\theta\right) + \epsilon\right] \middle| \theta'\right)\right)$$

$$= \mathbb{E}\left(\sup_{\hat{g}} \mathbb{P}\left(\hat{g}\left(\theta'\right) \in \left[g\left(\theta\right) - \epsilon, g\left(\theta\right) + \epsilon\right] \middle| \theta'\right)\right)$$

$$\geq \lceil\frac{T}{2\gamma\epsilon}\rceil^{-1}.$$

$\square$

## B Binary Search and Greedy Algorithms for Designing Quantization Mechanism

We use the binary search algorithm in Alg. 2 to search for the distortion budget that matches the privacy budget under the optimal data release mechanism.

---

**Algorithm 2:** Data release mechanism with privacy budget.

---

**Input:** Parameter range: $\left[\underline{\theta}, \overline{\theta}\right)$
        Privacy budget: $T$
        Distortion budget search range: $[\underline{B}, \overline{B}]$
        Step size: $s$ (which divides $\overline{\theta} - \underline{\theta}$)
        Precision: $\eta$

**1 while** $\overline{T} - \underline{T} \geq \eta$ **do**

**2**     $pri, \mathcal{S}, \theta' \leftarrow$ Algorithm-1 $\left(\left[\underline{\theta}, \overline{\theta}\right), \frac{\overline{T}+\underline{T}}{2}, \kappa\right)$

**3**     **if** $pri > T$ **then**

**4**        $\underline{B} \leftarrow \frac{\overline{T}+\underline{T}}{2}$

**5**     **else**

**6**        $\overline{B} \leftarrow \frac{\overline{T}+\underline{T}}{2}$

**7 return** *Data release mechanism parameters:* $\mathcal{S}, \theta'$

---

We provide the greedy algorithm in Alg. 3. In this algorithm, we greedily select the ranges of $\theta$ for each $\mathcal{S}_i$ in order. The left end point of the first range is the parameter lower bound (Line 2). We then scan across all possible right end point such that the distortion for this range will not exceed the budget $T$ (Line 8), and pick the one that gives the minimal attacker confidence (Line 10). After deciding the range of $\theta$, we will set of the released distribution for this range (Line 16), and then move on to the next range (Line 21). The time complexity of this algorithm is $\mathcal{O}\left(\left(\overline{\theta} - \underline{\theta}/\kappa\right)^2 \cdot \mathcal{C}_D \cdot \mathcal{C}_P\right)$, the same as the dynamic programming algorithm.

---

**Algorithm 3:** Greedy-based data release mechanism for single-parameter distributions.

---

**Input:** Parameter range: $\left(\underline{\theta}, \overline{\theta}\right]$
Prior over parameter: $f_{\Theta}$
Distortion budget: $T$
Step size: $\kappa$ (which divides $\overline{\theta} - \underline{\theta}$)

**1** $\mathcal{I} \leftarrow \emptyset$
**2** $L \leftarrow \underline{\theta}$
**3** $privacy \leftarrow 0$
**4** **while** $L < \overline{\theta}$ **do**
**5**    $min\_p \leftarrow \infty$
**6**    $min\_R \leftarrow$ NULL
**7**    $R \leftarrow L$
**8**    **while** $R \leq \overline{\theta}$ *and* $\mathcal{D}\left(L, R\right) \leq T$ **do**
**9**       $p \leftarrow \mathcal{P}\left(L, R\right)$
**10**       **if** $p \leq min\_p$ **then**
**11**          $min\_p \leftarrow p$
**12**          $min\_R \leftarrow R$
**13**       $R \leftarrow R + \kappa$
**14**    **if** *min\_R is not NULL* **then**
**15**       $\mathcal{S}_L \leftarrow \{X_\theta : \theta \in (L, \ min\_R]\}$
**16**       $\theta'_L \leftarrow \mathcal{D}\left(L, min\_R\right)$
**17**       $\mathcal{I} \leftarrow \mathcal{I} \cup \{L\}$
**18**       $privacy \leftarrow \dfrac{\int_{\underline{\theta}}^{L} f_{\Theta}(t)\mathrm{d}t}{\int_{\underline{\theta}}^{min\_R} f_{\Theta}(t)\mathrm{d}t} \cdot privacy + \dfrac{\int_{L}^{min\_R} f_{\Theta}(t)\mathrm{d}t}{\int_{\underline{\theta}}^{min\_R} f_{\Theta}(t)\mathrm{d}t} \cdot min\_p$
**19**    **else**
**20**       ERROR: No answer
**21**    $L \leftarrow min\_R$
**22** **return** $privacy, \{\mathcal{S}_i : i \in \mathcal{I}\}, \{\theta'_i : i \in \mathcal{I}\}$

---

## C Proofs

### C.1 Proof of Corollary 1

*Proof.* For any $X_{\theta_1}, X_{\theta_2}$, we have

$$
\begin{aligned}
D\left(X_{\theta_1}, X_{\theta_2}\right) &= \frac{1}{2} d_{\text{Wasserstein-1}}\left(\omega_{X_{\theta_1}} \| \omega_{X_{\theta_2}}\right) \\
&\geq \frac{1}{2}|g\left(\theta_1\right) - g\left(\theta_2\right)| \\
&= \frac{1}{2} R\left(X_{\theta_1}, X_{\theta_2}\right).
\end{aligned}
\tag{19}
$$

where Eq. (19) comes from Jensen inequality. Therefore, we have $\gamma = \inf_{\theta_1, \theta_2 \in \text{Supp}(\omega_\Theta)} \frac{D\left(X_{\theta_1}, X_{\theta_2}\right)}{R\left(X_{\theta_1}, X_{\theta_2}\right)} \geq \frac{1}{2}$. The result then follows from Thm. 1. □

### C.2 Proof of Prop. 1

*Proof.* For any released parameter $\theta' = (u', v')$, there exists $i \in \{0, ..., N-1\}$ such that $u' = \underline{u} + (i + 0.5) \cdot s$. We have

$$
\sup_{\hat{g}} \mathbb{P}\left(\hat{g}\left(\theta'\right) \in [g\left(\theta\right) - \epsilon, g\left(\theta\right) + \epsilon] \,\big|\, \theta'\right)
$$

$$
= \sup_{\hat{g}} \int_{\underline{u}+i\cdot s}^{\underline{u}+(i+1)\cdot s} f_{U|U'}\left(u|u'\right) \cdot \int_{u-\epsilon}^{u+\epsilon} f_{\hat{g}(u', v')}\left(h\right) \; dh \; du
$$

$$
= \sup_{\hat{g}} \int_{\underline{u}+i\cdot s-\epsilon}^{\underline{u}+(i+1)\cdot s+\epsilon} f_{\hat{g}(u', v')}(h) \cdot \int_{\hat{g}\left(f_{X_{u', v'}}\right)-\epsilon}^{\hat{g}\left(f_{X_{u', v'}}\right)+\epsilon} f_{U|U'}\left(u|u'\right) \; du \; dh
$$

$$
\leq \sup_{\hat{g}} \int_{\underline{u}+i\cdot s-\epsilon}^{\underline{u}+(i+1)\cdot s+\epsilon} \frac{2\epsilon}{s} \cdot f_{\hat{g}(u', v')}(h) \; dh
$$

$$
\leq \frac{2\epsilon}{s}.
$$

Therefore, we have

$$
\begin{aligned}
\Pi_{\epsilon, \omega_\Theta} &= \sup_{\hat{g}} \mathbb{P}\left(\hat{g}\left(\theta'\right) \in [g\left(\theta\right) - \epsilon, g\left(\theta\right) + \epsilon]\right) \\
&= \sup_{\hat{g}} \mathbb{E}\left(\mathbb{P}\left(\hat{g}\left(\theta'\right) \in [g\left(\theta\right) - \epsilon, g\left(\theta\right) + \epsilon] \,\Big|\, \theta'\right)\right) \\
&= \mathbb{E}\left(\sup_{\hat{g}} \mathbb{P}\left(\hat{g}\left(\theta'\right) \in [g\left(\theta\right) - \epsilon, g\left(\theta\right) + \epsilon] \,\Big|\, \theta'\right)\right) \\
&\leq \frac{2\epsilon}{s}.
\end{aligned}
$$

For the distortion, we can easily get that $\Delta = \frac{s}{2}$. According to Corollary 1, we have $\Delta_{\text{opt}} > \left(\lceil \frac{1}{\Pi_{\epsilon, \omega_\Theta}} \rceil - 1\right)\epsilon \geq \epsilon$. We can get that

$$
\begin{aligned}
\Delta &= \Delta_{\text{opt}} + \Delta - \Delta_{\text{opt}} \\
&< \Delta_{\text{opt}} + \Delta - \left(\lceil \frac{1}{\Pi_{\epsilon, \omega_\Theta}} \rceil - 1\right) \cdot \epsilon \\
&\leq \Delta_{\text{opt}} + \epsilon + \Delta - \frac{\epsilon}{\Pi_{\epsilon, \omega_\Theta}} \\
&\leq \Delta_{\text{opt}} + \epsilon \\
&\leq 2\Delta_{\text{opt}}.
\end{aligned}
$$

$\square$

### C.3 Proof of Corollary 2

#### C.3.1 Exponential Distribution

*Proof.* Let $X_{\lambda_1}, X_{\lambda_2}$ be two exponential random variables. We have

$$\frac{D\left(X_{\lambda_1}, X_{\lambda_2}\right)}{R\left(X_{\lambda_1}, X_{\lambda_2}\right)} = \frac{\frac{1}{2}\left(\lambda_1 - \lambda_2\right)}{-\ln\left(1-\alpha\right)\left(\lambda_1 - \lambda_2\right)} = -\frac{1}{2\ln\left(1-\alpha\right)}. \tag{20}$$

Therefore we can get that

$$\gamma = -\frac{1}{2\ln\left(1-\alpha\right)}.$$

$\square$

#### C.3.2 Shifted Exponential Distribution

*Proof.* Let $X_{\lambda_1,h_1}, X_{\lambda_2,h_2}$ be random variables from shifted exponential distributions. Let $\lambda_2 \leq \lambda_1$ without loss of generality. Let $a = \frac{\lambda_1}{\lambda_2}$ and $b = \left(h_1/\lambda_1 - h_2/\lambda_2\right)\lambda_2$. We can get that $f_{X_{\lambda_1,h_1}}\left(x\right) = af_{X_{\lambda_2,h_2}}\left(a\left(x+b\right)\right)$, and

$$\begin{aligned}
D\left(X_{\lambda_1,h_1}, X_{\lambda_2,h_2}\right) &= \frac{1}{2}d_{\text{Wasserstein-1}}\left(\omega_{X_{\lambda_1,h_1}} \| \omega_{X_{\lambda_2,h_2}}\right) \\
&= \frac{1}{2}\int_{h_1}^{+\infty}\left|x - \left(\frac{x}{a} - b\right)\right| f_{X_{\lambda_1,h_1}}\left(x\right)\mathrm{d}x \\
&= \frac{\lambda_2}{2\lambda_1}\int_{h_1}^{+\infty}\left|\left(1/\lambda_2 - 1/\lambda_1\right)x + h_1/\lambda_1 - h_2/\lambda_2\right| e^{-\frac{1}{\lambda_1}\left(x-h_1\right)}\mathrm{d}x \\
&= \begin{cases} \frac{1}{2}\left(h_2 - h_1 + \lambda_2 - \lambda_1\right) - e^{\frac{h_2-h_1}{\lambda_2-\lambda_1}}\left(\lambda_2 - \lambda_1\right) & \left(h_1 < h_2\right) \\ \frac{1}{2}\left(h_1 - h_2 + \lambda_1 - \lambda_2\right) & \left(h_1 \geq h_2\right) \end{cases},
\end{aligned} \tag{21}$$

$$R\left(X_{\lambda_1,h_1}, X_{\lambda_2,h_2}\right) = \left|\ln\left(1-\alpha\right)\left(\lambda_1 - \lambda_2\right) + h_2 - h_1\right|.$$

When $h_1 < h_2$, let $t = \frac{h_2-h_1}{\lambda_1-\lambda_2} \in (0, +\infty)$. We have

$$\begin{aligned}
&\frac{D\left(X_{\lambda_1,h_1}, X_{\lambda_2,h_2}\right)}{R\left(X_{\lambda_1,h_1}, X_{\lambda_2,h_2}\right)} \\
&= \frac{h_2 - h_1 + \lambda_2 - \lambda_1 - 2e^{\frac{h_2-h_1}{\lambda_2-\lambda_1}}\left(\lambda_2 - \lambda_1\right)}{2\left|\ln\left(1-\alpha\right)\left(\lambda_1 - \lambda_2\right) + h_2 - h_1\right|} \\
&= \frac{t + 2e^{-t} - 1}{2\left|\ln\left(1-\alpha\right) + t\right|} \\
&\geq \begin{cases} \frac{1}{2}\left|1 + \frac{\ln(1-\alpha)+1}{W_{-1}\left(-\frac{\ln(1-\alpha)+1}{2(1-\alpha)e}\right)}\right| & \alpha \in [0, 1-e^{-1}) \\ \frac{1}{2}\left|1 + \frac{\ln(1-\alpha)+1}{W_0\left(-\frac{\ln(1-\alpha)+1}{2(1-\alpha)e}\right)}\right| & \alpha \in [1-e^{-1}, 1) \end{cases},
\end{aligned}$$

where $W_{-1}$ and $W_0$ are Lambert $W$ functions. "$=$" achieves when

$$t = t_0 \triangleq \begin{cases} -1 - \ln\left(1-\alpha\right) - W_{-1}\left(-\frac{\ln(1-\alpha)+1}{2(1-\alpha)e}\right) & \left(\alpha \in [0, 1-e^{-1})\right) \\ -1 - \ln\left(1-\alpha\right) - W_0\left(-\frac{\ln(1-\alpha)+1}{2(1-\alpha)e}\right) & \left(\alpha \in [1-e^{-1}, 1)\right) \end{cases}.$$

When $h_1 \geq h_2$, let $t = \frac{h_1 - h_2}{\lambda_1 - \lambda_2} \in (0, +\infty)$. We have

$$
\begin{aligned}
\frac{D\left(X_{\lambda_1, h_1}, X_{\lambda_2, h_2}\right)}{R\left(X_{\lambda_1, h_1}, X_{\lambda_2, h_2}\right)} &= \frac{h_1 - h_2 + \lambda_1 - \lambda_2}{2\left|\ln\left(1 - \alpha\right)\left(\lambda_1 - \lambda_2\right) + h_2 - h_1\right|} \\
&= \frac{t + 1}{2\left|\ln\left(1 - \alpha\right) - t\right|} \\
&\geq \min\left\{\frac{1}{2}, -\frac{1}{2\ln\left(1 - \alpha\right)}\right\}.
\end{aligned}
$$

Therefore we can get that

$$
\gamma = \begin{cases}
\frac{1}{2}\left|1 + \frac{\ln(1-\alpha)+1}{W_{-1}\left(-\frac{\ln(1-\alpha)+1}{2(1-\alpha)e}\right)}\right| & \alpha \in [0, 1 - e^{-1}) \\
\frac{1}{2}\left|1 + \frac{\ln(1-\alpha)+1}{W_0\left(-\frac{\ln(1-\alpha)+1}{2(1-\alpha)e}\right)}\right| & \alpha \in [1 - e^{-1}, 1)
\end{cases}.
$$

$\square$

## C.4 Proof of Prop. 2

### C.4.1 Exponential Distribution

*Proof.* The proof of $\Delta$ and $\Pi_{\epsilon, \omega_\Theta}$ is the same as App. C.2, except that we use the $D\left(\cdot, \cdot\right)$ and $R\left(\cdot, \cdot\right)$ from Eq. (20).

For $\Delta_{\mathrm{opt}}$, we have $\Delta_{\mathrm{opt}} > \left(\lceil\frac{1}{\Pi_{\epsilon, \omega_\Theta}}\rceil - 1\right) \cdot 2\gamma\epsilon \geq 2\gamma\epsilon$, where $\gamma = -\frac{1}{2\ln(1-\alpha)}$. We can get that

$$
\begin{aligned}
\Delta &= \Delta_{\mathrm{opt}} + \Delta - \Delta_{\mathrm{opt}} \\
&< \Delta_{\mathrm{opt}} + \Delta - \left(\lceil\frac{1}{\Pi_{\epsilon, \omega_\Theta}}\rceil - 1\right) \cdot 2\gamma\epsilon \\
&\leq \Delta_{\mathrm{opt}} + 2\gamma\epsilon + \Delta - \frac{2\gamma\epsilon}{\Pi_{\epsilon, \omega_\Theta}} \\
&= \Delta_{\mathrm{opt}} + 2\gamma\epsilon \\
&\leq 2\Delta_{\mathrm{opt}}.
\end{aligned}
$$

$\square$

### C.4.2 Shifted Exponential Distribution

*Proof.* We first focus on the proof for $\Pi_{\epsilon, \omega_\Theta}$.

In Fig. 8, we separate the space of possible data parameters into two regions represented by yellow and green colors. The yellow regions $S_{yellow}$ constitute right triangles with height $s$ and width $|t_0|s$. The green region $S_{green}$ is the rest of the parameter space. The high-level idea of our proof is as follows. Note that for any parameter $\theta \in S_{green}$, there exists a $\mathcal{S}_{i,h}$ s.t. $\theta \in \mathcal{S}_{i,h}$ and $\mathcal{S}_{\mu,i} \subset S_{green}$. Therefore, we can bound the attack success rate if $\theta \in S_{green}$. At the same time, the probability of $\theta \in S_{yellow}$ is bounded. Therefore, we can bound the overall attacker's success rate (i.e., $\Pi_{\epsilon, \omega_\Theta}$). More specifically, let the optimal attacker be $\hat{g}^*$. We

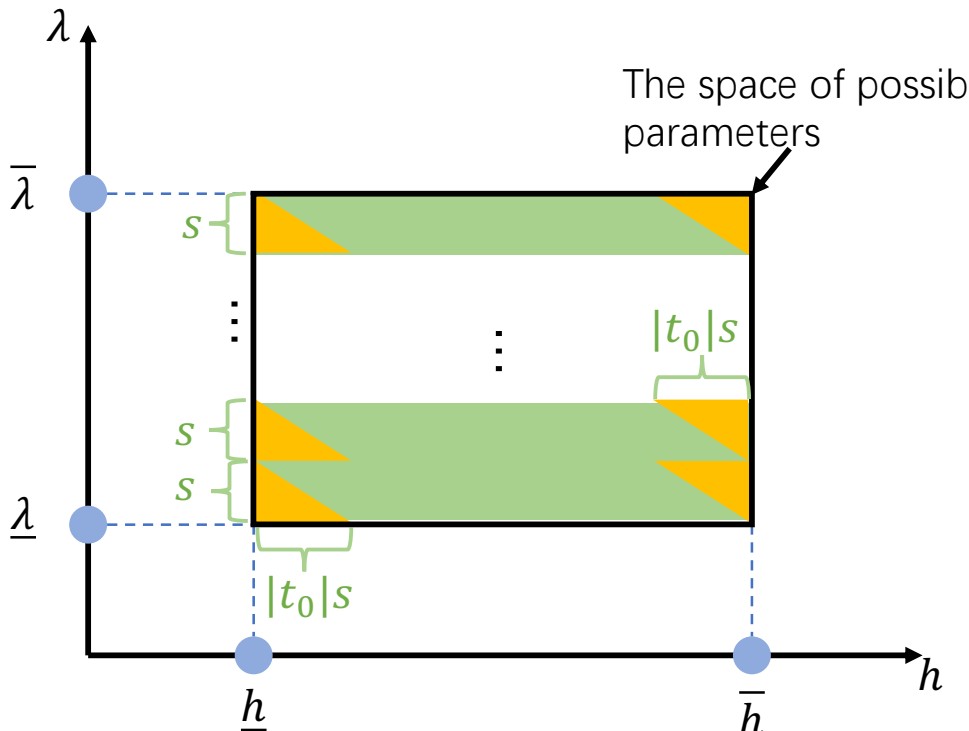

Figure 8: The construction for proof of Prop. 2 for shifted exponential distributions. We separate the space of possible parameters into two regions (yellow and green) and bound the attacker's success rate on each region separately.

have

$$\Pi_{\epsilon,\omega_\Theta} = \mathbb{P}\left(\hat{g}^*\left(\theta'\right) \in \left[g\left(\theta\right) - \epsilon, g\left(\theta\right) + \epsilon\right]\right)$$

$$= \int_{\theta \in S_{green}} p(\theta)\mathbb{P}\left(\hat{g}^*\left(\theta'\right) \in \left[g\left(\theta\right) - \epsilon, g\left(\theta\right) + \epsilon\right]\right) d\theta$$

$$+ \int_{\theta \in S_{yellow}} p(\theta)\mathbb{P}\left(\hat{g}^*\left(\theta'\right) \in \left[g\left(\theta\right) - \epsilon, g\left(\theta\right) + \epsilon\right]\right) d\theta$$

$$< \frac{2\epsilon}{|\ln\left(1 - \alpha\right) + t_0|s} + \frac{|t_0|s}{\overline{h} - \underline{h}}.$$

For the distortion, it is straightforward to get that $\Delta = \frac{s}{2}\left(t_0 - 1\right) + se^{-t_0}$ from Eq. (21), and $\Delta_{\text{opt}} > \left(\lceil\frac{1}{\Pi_{\epsilon,\omega_\Theta}}\rceil - 1\right) \cdot 2\gamma\epsilon \geq 2\gamma\epsilon$, where $\gamma$ is defined in Corollary 2. Denote $\zeta = \frac{2\epsilon}{|\ln(1-\alpha)+t_0|s} + \frac{|t_0|s}{\overline{h}-\underline{h}} - \Pi_{\epsilon,\omega_\Theta}$, we

can get that $\left(\Pi_{\epsilon,\omega_\Theta} + \zeta - \frac{|t_0|s}{\overline{h}-\underline{h}}\right) \cdot \Delta = 2\gamma\epsilon$ and

$$\Delta = \Delta_{\text{opt}} + \Delta - \Delta_{\text{opt}}$$

$$< \Delta_{\text{opt}} + \Delta - \left(\lceil \frac{1}{\Pi_{\epsilon,\omega_\Theta}} \rceil - 1\right) \cdot 2\gamma\epsilon$$

$$\le \Delta_{\text{opt}} + 2\gamma\epsilon + \Delta - \frac{2\gamma\epsilon}{\Pi_{\epsilon,\omega_\Theta}}$$

$$= \Delta_{\text{opt}} + 2\gamma\epsilon + \frac{\frac{|t_0|s}{\overline{h}-\underline{h}} - \zeta}{\frac{2\epsilon}{|\ln(1-\alpha)+t_0|s} + \frac{|t_0|s}{\overline{h}-\underline{h}} - \zeta} \cdot \Delta$$

$$< \Delta_{\text{opt}} + 2\gamma\epsilon + \frac{\frac{|t_0|s}{\overline{h}-\underline{h}}}{\frac{2\epsilon}{|\ln(1-\alpha)+t_0|s} + \frac{|t_0|s}{\overline{h}-\underline{h}}} \cdot \Delta.$$

Therefore,

$$\Delta < \left(1 + \frac{|t_0| \cdot |\ln(1-\alpha) + t_0|s^2}{2\epsilon(\overline{h}-\underline{h})}\right)(\Delta_{\text{opt}} + 2\gamma\epsilon)$$

$$\le \left(2 + \frac{|t_0| \cdot |\ln(1-\alpha) + t_0|s^2}{\epsilon(\overline{h}-\underline{h})}\right)\Delta_{\text{opt}}.$$

$t_0$ is bounded when $\alpha \in [0, c_1] \cup \left[1 - \frac{1}{e}, c_2\right]$, where $c_1 \in \left[0, 1 - \frac{1}{e}\right), c_2 \in \left[1 - \frac{1}{e}, 1\right)$. Therefore, when $\alpha \in [0.01, 0.25] \cup [0.75, 0.99]$, we can get that

$$\lim_{\frac{s^2}{\overline{h}-\underline{h}} \to 0} \sup \Delta < \lim_{\frac{s^2}{\overline{h}-\underline{h}} \to 0} \sup \left(2 + \frac{|t_0| \cdot |\ln(1-\alpha) + t_0|s^2}{\epsilon(\overline{h}-\underline{h})}\right)\Delta_{\text{opt}} < 3\Delta_{\text{opt}}.$$

$\square$

### C.5 Proofs for the Surrogate Metrics

#### C.5.1 Secret=Mean

For any $p_{\mathcal{Y}}$, we have

$$\tilde{\Delta} = d_{\text{Wasserstein-1}}(p_{\mathcal{X}} \| p_{\mathcal{Y}}) \ge \left| \frac{1}{n}\sum_{i=1}^{n} x_i - \frac{1}{n}\sum_{i=1}^{n} y_i \right| = -\Pi_{\epsilon,\omega_\Theta}^{\sim}.$$

For $p_{\mathcal{Y}}$ released from our mechanism (§6.3), we have $\tilde{\Delta} = d_{\text{Wasserstein-1}}(p_{\mathcal{X}} \| p_{\mathcal{Y}}) = \left| \frac{1}{n}\sum_{i=1}^{n} x_i - \frac{1}{n}\sum_{i=1}^{n} y_i \right| = -\Pi_{\epsilon,\omega_\Theta}^{\sim}$.

#### C.5.2 Secret=Fraction

Assume that we want to protect the fraction of class $j$, and $fraction(\mathcal{D}, j)$ means the fraction of sample $j$ in the dataset $\mathcal{D}$.

For any $p_{\mathcal{Y}}$, we have

$$\tilde{\Delta} = d_{\text{TV}}(p_{\mathcal{X}} \| p_{\mathcal{Y}}) \ge |fraction(\mathcal{X}, j) - fraction(\mathcal{Y}, j)| = -\Pi_{\epsilon,\omega_\Theta}^{\sim}.$$

For $p_{\mathcal{Y}}$ released from our mechanism (Mech. 5), we have $\tilde{\Delta} = d_{\text{TV}}(p_{\mathcal{X}} \| p_{\mathcal{Y}}) = |fraction(\mathcal{X}, j) - fraction(\mathcal{Y}, j)| = -\Pi_{\epsilon,\omega_\Theta}^{\sim}$.

### C.6 Proof of Prop. 3

*Proof.* We prove by contradiction. For any two parameters $\theta_1, \theta_2 \in \text{Supp}(\omega_\Theta)$, we can construct a prior distribution $\mathbb{P}(\theta = \theta_1) = \mathbb{P}(\theta = \theta_2) = \frac{1}{2}$. Because $\Pi'_{\epsilon, \omega_\Theta} < \log 2$, we have

$$\sup_{\hat{g}} \ \mathbb{P}(\hat{g}(\theta') \in [g(\theta) - \epsilon, g(\theta) + \epsilon]) < 1$$

under this prior distribution. Therefore, there exists $\theta'$ and $z_1, z_2 \in \text{Supp}(\omega_\Theta)$ s.t. $\mathcal{M}_g(\theta_1, z_1) = \mathcal{M}_g(\theta_2, z_2) = \theta'$. According to triangle inequality, we have $\max\left\{d\left(\omega_{X_{\theta_1}} \| \omega_{X_{\theta'}}\right), d\left(\omega_{X_{\theta_2}} \| \omega_{X_{\theta'}}\right)\right\} \geq \frac{1}{2} d\left(\omega_{X_{\theta_1}} \| \omega_{X_{\theta_2}}\right)$. Therefore, we have $\Delta \geq \overline{\Delta}$, which gives a contradiction. □

## D Privacy-Distortion Performance of Data Release Mechanism with Relaxed Assumption

### D.1 Privacy-Distortion Performance of Mech. 1 with Relaxed Assumption

We relax Asm. 1 as follows.

**Assumption 3.** *The distribution parameter vector $\theta$ can be written as $(u, v)$, where $u \in \mathbb{R}$, $v \in \mathbb{R}^{q-1}$, and for any $u \neq u'$, $f_{X_{u,v}}(x) = f_{X_{u',v}}(x - u' + u)$. The prior over distribution parameters is $f_{U,V}(a, b) = f_U(a) \cdot f_V(b)$, where $\text{Supp}(U) = [\underline{u}, \overline{u})$, and $f_U$ is $\mathcal{L}$-Lipschitz continuous and has lower bound $\underline{c}$.*

Based on Asm. 3, the Privacy-distortion performance of Mech. 1 is shown below.

**Proposition 4.** *Under Asm. 3, Mech. 1 has $\Delta = \frac{s}{2}$ and $\Pi_{\epsilon, \omega_\Theta} \leq \frac{2\epsilon[\underline{c} + \mathcal{L}(s - x^* - \epsilon)]}{\underline{c}s + \frac{\mathcal{L}}{2}(s - x^*)^2}$, where $x^* = s + \frac{\underline{c}}{\mathcal{L}} - \epsilon - \sqrt{\left(\frac{\underline{c}}{\mathcal{L}} - \epsilon\right)^2 + \frac{2\underline{c}s}{\mathcal{L}}}$.*

*Proof.* We first provide the following lemma.

**Lemma 1.** *For a $\mathcal{L}$-Lipschitz continuous function $f(x), x \in [\underline{x}, \overline{x}]$, $\inf_{x \in [\underline{x}, \overline{x}]} f(x) \geq \underline{c} \geq 0$, it satisfies*

$$\sup_{x' \in [\underline{x}, \overline{x} - \delta]} \frac{\int_{x'}^{x' + \delta} f(x) \mathrm{d}x}{\int_{\underline{x}}^{\overline{x}} f(x) \mathrm{d}x} \leq \frac{\delta\left[\underline{c} + \mathcal{L}\left(\overline{x} - x^* - \frac{\delta}{2}\right)\right]}{\underline{c}(\overline{x} - \underline{x}) + \frac{\mathcal{L}}{2}(\overline{x} - x^*)^2},$$

*where $x^* = \overline{x} + \frac{\underline{c}}{\mathcal{L}} - \frac{\delta}{2} - \sqrt{\left(\frac{\underline{c}}{\mathcal{L}} - \frac{\delta}{2}\right)^2 + \frac{2\underline{c}(\overline{x} - \underline{x})}{\mathcal{L}}}$.*

For any released parameter $\theta' = (u', v')$, there exists $i \in \{0, ..., N-1\}$ such that $u' = \underline{u} + (i + 0.5) \cdot s$. We have

$$\sup_{\hat{g}} \mathbb{P}\left(\hat{g}(\theta') \in [g(\theta) - \epsilon, g(\theta) + \epsilon] \,\middle|\, \theta'\right)$$

$$= \sup_{\hat{g}} \int_{\underline{u} + i \cdot s}^{\underline{u} + (i+1) \cdot s} f_{U|U'}(u|u') \cdot \int_{u - \epsilon}^{u + \epsilon} f_{\hat{g}(u', v')}(h) \ \mathrm{d}h \ \mathrm{d}u$$

$$= \sup_{\hat{g}} \int_{\underline{u} + i \cdot s - \epsilon}^{\underline{u} + (i+1) \cdot s + \epsilon} f_{\hat{g}(u', v')}(h) \cdot \int_{\hat{g}\left(f_{X_{u', v'}}\right) - \epsilon}^{\hat{g}\left(f_{X_{u', v'}}\right) + \epsilon} f_{U|U'}(u|u') \ \mathrm{d}u \ \mathrm{d}h.$$

For $\int_{\hat{g}\left(f_{X_{u', v'}}\right) - \epsilon}^{\hat{g}\left(f_{X_{u', v'}}\right) + \epsilon} f_{U|U'}(u|u') \ \mathrm{d}u$, denote

$$x_1 = \max\left(0, \hat{g}\left(f_{X_{u', v'}}\right) - \epsilon - \underline{u} - i \cdot s\right),$$

$$x_2 = \min\left(\hat{g}\left(f_{X_{u', v'}}\right) + \epsilon - \underline{u} - i \cdot s, s\right),$$

we have

$$\int_{\hat{g}\left(f_{X_{u',v'}}\right)-\epsilon}^{\hat{g}\left(f_{X_{u',v'}}\right)+\epsilon} f_{U|U'}\left(u|u'\right)\, \mathrm{d}u = \frac{\int_{x_1}^{x_2} f_U\left(\underline{u}+i\cdot s+x\right)\, \mathrm{d}x}{\int_0^s f_U\left(\underline{u}+i\cdot s+x\right)\, \mathrm{d}x}.$$

$f_U\left(\underline{u}+i\cdot s+x\right)$ is $\mathcal{L}$-Lipschitz and has lower bound $\underline{c}$. $x_2-x_1 \leq 2\epsilon$ and $x_1, x_2 \in [0,s]$. According to Lemma 1, we have

$$\begin{aligned}
\int_{\hat{g}\left(f_{X_{u',v'}}\right)-\epsilon}^{\hat{g}\left(f_{X_{u',v'}}\right)+\epsilon} f_{U|U'}\left(u|u'\right)\, \mathrm{d}u &= \frac{\int_{x_1}^{x_2} f_U\left(\underline{u}+i\cdot s+x\right)\, \mathrm{d}x}{\int_0^s f_U\left(\underline{u}+i\cdot s+x\right)\, \mathrm{d}x} \\
&\leq \frac{2\epsilon\left[\underline{c}+\mathcal{L}\left(s-x^*-\epsilon\right)\right]}{\underline{c}s+\frac{\mathcal{L}}{2}\left(s-x^*\right)^2},
\end{aligned}$$

where $x^* = s+\frac{\underline{c}}{\mathcal{L}}-\epsilon-\sqrt{\left(\frac{\underline{c}}{\mathcal{L}}-\epsilon\right)^2+\frac{2\underline{c}s}{\mathcal{L}}}$.

Therefore, we can get that

$$\begin{aligned}
\sup_{\hat{g}} \mathbb{P}&\left(\hat{g}\left(\theta'\right)\in\left[g\left(\theta\right)-\epsilon, g\left(\theta\right)+\epsilon\right]\big|\theta'\right) \\
&\leq \sup_{\hat{g}}\int_{\underline{u}+i\cdot s-\epsilon}^{\underline{u}+(i+1)\cdot s+\epsilon} \frac{2\epsilon\left[\underline{c}+\mathcal{L}\left(s-x^*-\epsilon\right)\right]}{\underline{c}s+\frac{\mathcal{L}}{2}\left(s-x^*\right)^2}\cdot f_{\hat{g}(u',v')}(h)\, \mathrm{d}h \\
&\leq \frac{2\epsilon\left[\underline{c}+\mathcal{L}\left(s-x^*-\epsilon\right)\right]}{\underline{c}s+\frac{\mathcal{L}}{2}\left(s-x^*\right)^2}.
\end{aligned}$$

Therefore, we have

$$\begin{aligned}
\Pi_{\epsilon,\omega_\Theta} &= \sup_{\hat{g}} \mathbb{P}\left(\hat{g}\left(\theta'\right)\in\left[g\left(\theta\right)-\epsilon, g\left(\theta\right)+\epsilon\right]\right) \\
&= \sup_{\hat{g}} \mathbb{E}\left(\mathbb{P}\left(\hat{g}\left(\theta'\right)\in\left[g\left(\theta\right)-\epsilon, g\left(\theta\right)+\epsilon\right]\Big|\theta'\right)\right) \\
&= \mathbb{E}\left(\sup_{\hat{g}}\mathbb{P}\left(\hat{g}\left(\theta'\right)\in\left[g\left(\theta\right)-\epsilon, g\left(\theta\right)+\epsilon\right]\Big|\theta'\right)\right) \\
&\leq \frac{2\epsilon\left[\underline{c}+\mathcal{L}\left(s-x^*-\epsilon\right)\right]}{\underline{c}s+\frac{\mathcal{L}}{2}\left(s-x^*\right)^2}.
\end{aligned}$$

For the distortion, we can easily get that $\Delta=\frac{s}{2}$. $\qquad\square$

### D.1.1 Proof of Lemma 1

Without loss of generality, we assume that $f(\overline{x})\geq f(\underline{x})$. Based on simple geometric analysis, we can get that when $\frac{\int_{x'}^{x'+\delta} f(x)\mathrm{d}x}{\int_{\underline{x}}^{\overline{x}} f(x)\mathrm{d}x}$ achieves supremum, as illustrated in Fig. 9, $f(\underline{x})=\underline{c}$, $x'=\overline{x}-\delta$, and $f(\overline{x})=\underline{x}+\mathcal{L}\left(\overline{x}-x''\right)$, where $x''\in[\underline{x}, x']$.

In this case, we can get that

$$\frac{\int_{\overline{x}-\delta}^{\overline{x}} f(x)\mathrm{d}x}{\int_{\underline{x}}^{\overline{x}} f(x)\mathrm{d}x} = \frac{\delta\left[\underline{c}+\mathcal{L}\left(\overline{x}-x''-\frac{\delta}{2}\right)\right]}{\underline{c}\left(\overline{x}-\underline{x}\right)+\frac{\mathcal{L}}{2}\left(\overline{x}-x''\right)^2} \triangleq h\left(x''\right),$$

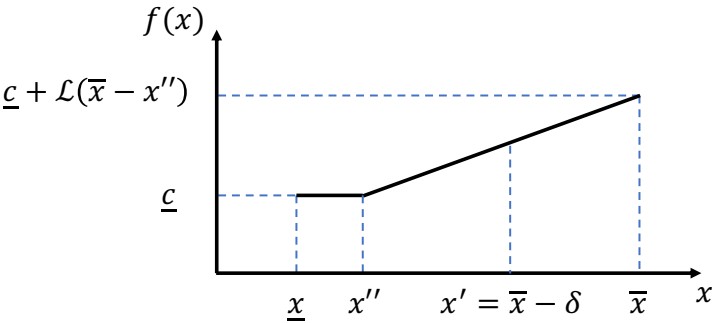

Figure 9: Illustration of $f(x)$ when $\dfrac{\int_{x'}^{x'+\delta} f(x)\mathrm{d}x}{\int_{\underline{x}}^{\overline{x}} f(x)\mathrm{d}x}$ achieves supremum.

where $x'' \in [\underline{x}, x']$. When $x'' = \overline{x} + \frac{\underline{c}}{\mathcal{L}} - \frac{\delta}{2} - \sqrt{\left(\frac{\underline{c}}{\mathcal{L}} - \frac{\delta}{2}\right)^2 + \frac{2\underline{c}(\overline{x}-\underline{x})}{\mathcal{L}}} \triangleq x^*$, $h(x'')$ achieves supremum. Therefore, we have

$$\sup_{x' \in [\underline{x}, \overline{x}-\delta]} \frac{\int_{x'}^{x'+\delta} f(x)\mathrm{d}x}{\int_{\underline{x}}^{\overline{x}} f(x)\mathrm{d}x} \leq \sup_{f} \sup_{x' \in [\underline{x}, \overline{x}-\delta]} \frac{\int_{x'}^{x'+\delta} f(x)\mathrm{d}x}{\int_{\underline{x}}^{\overline{x}} f(x)\mathrm{d}x}$$

$$= \frac{\delta \left[\underline{c} + \mathcal{L}\left(\overline{x} - x^* - \frac{\delta}{2}\right)\right]}{\underline{c}(\overline{x} - \underline{x}) + \frac{\mathcal{L}}{2}(\overline{x} - x^*)^2}.$$

## D.2 Privacy-Distortion Performance of Mech. 2 with Relaxed Assumption

We relax Asm. 2 as follows.

**Assumption 4.** *The prior over distribution parameters as specified below.*

- *Exponential: $Supp(\lambda) = [\underline{\lambda}, \overline{\lambda})$, and $f_\lambda$ is $\mathcal{L}$-Lipschitz continuous and has lower bound $\underline{c}$.*
- *Shifted exponential: $Supp(\lambda, h) = \left\{(a, b) | a \in [\underline{\lambda}, \overline{\lambda}), b \in [\underline{h}, \overline{h})\right\}$, $f_{\lambda,h}(a, b) = f_\lambda(a) \cdot f_h(b)$, and $f_\lambda$ (resp. $f_h$) is $\mathcal{L}_\lambda$-Lipschitz (resp. $\mathcal{L}_h$-Lipschitz) and has lower bound $\frac{k_\lambda}{\overline{\mu}-\underline{\mu}}$ with $k_\lambda \in (0, 1]$ (resp. $\frac{k_h}{\overline{\sigma}-\underline{\sigma}}$ with $k_h \in (0, 1]$).*

Based on Asm. 4, the Privacy-distortion performance of Mech. 2 is shown below.

**Proposition 5.** *Under Asm. 4, Mech. 2 has the following $\Delta$ and $\Pi_{\epsilon, \omega_\Theta}$ value/bound.*

- *Exponential:*

$$\Delta = \frac{1}{2}s,$$

$$\Pi_{\epsilon, \omega_\Theta} \leq \frac{\frac{2\epsilon}{-\ln(1-\alpha)} \cdot \left[\underline{c} + \mathcal{L}\left(s - x^* + \frac{\epsilon}{\ln(1-\alpha)}\right)\right]}{\underline{c}s + \frac{\mathcal{L}}{2}(s - x^*)^2},$$

*where $x^* = s + \frac{\underline{c}}{\mathcal{L}} + \frac{\epsilon}{\ln(1-\alpha)} - \sqrt{\left(\frac{\underline{c}}{\mathcal{L}} + \frac{\epsilon}{\ln(1-\alpha)}\right)^2 + \frac{2\underline{c}s}{\mathcal{L}}}.$*

- *Shifted exponential:*

$$\Delta = \frac{s}{2}\left(t_0 - 1\right) + se^{-t_0},$$

$$\Pi_{\epsilon,\omega_\Theta} < \frac{\frac{2\epsilon}{|\ln(1-\alpha)+t_0|} \cdot \left[\underline{c} + \mathcal{L}_{\lambda,h}\left(\frac{s}{2} - t^* - \frac{\epsilon}{|\ln(1-\alpha)+t_0|}\right)\right]}{\underline{c}s + \frac{\mathcal{L}_{\lambda,h}}{2}\left(\frac{s}{2} - t^*\right)^2} +$$

$$M\left(\overline{h} - \underline{h}, \frac{k_h}{\overline{h} - \underline{h}}, \mathcal{L}_h, 1\right) \cdot M\left(\overline{\lambda} - \underline{\lambda}, \frac{k_\lambda}{\overline{\lambda} - \underline{\lambda}}, \mathcal{L}_\lambda, 1\right) \cdot \left(\overline{\lambda} - \underline{\lambda}\right)|t_0|s,$$

*where* $\underline{c} = \frac{k_h k_\lambda}{\left(\overline{h} - \underline{h}\right) \cdot \left(\overline{\lambda} - \underline{\lambda}\right)}$, *function* $M$ *satisfies*

$$M\left(x, c, \mathcal{L}, \mathcal{A}\right) = \begin{cases} \frac{\mathcal{A}}{x} + \frac{\mathcal{L}x}{2}, & \text{if } c \leq \frac{\mathcal{A}}{x} - \frac{\mathcal{L}x}{2} \\ c + \sqrt{2\mathcal{L}\left(\mathcal{A} - cx\right)}, & \text{if } c > \frac{\mathcal{A}}{x} - \frac{\mathcal{L}x}{2} \end{cases},$$

$\mathcal{L}_{\lambda,h} = \mathcal{L}_\lambda M\left(\frac{\overline{h} - \underline{h}}{|t_0|}, \frac{k_h}{\overline{h} - \underline{h}}, |t_0|\mathcal{L}_h, \frac{1}{|t_0|}\right) + |t_0|\mathcal{L}_h M\left(\overline{\lambda} - \underline{\lambda}, \frac{k_\lambda}{\overline{\lambda} - \underline{\lambda}}, \mathcal{L}_\lambda, 1\right)$, *and*

$t^* = \frac{s}{2} + \frac{\underline{c}}{\mathcal{L}_{\lambda,h}} - \frac{\epsilon}{|\ln(1-\alpha)+t_0|} - \sqrt{\left(\frac{\underline{c}}{\mathcal{L}_{\lambda,h}} - \frac{\epsilon}{|\ln(1-\alpha)+t_0|}\right)^2 + \frac{2\underline{c}s}{\mathcal{L}_{\lambda,h}}}.$

*The* $t_0$ *parameter is defined in* [Mech. 2](#).

### D.2.1  Proof of [Prop. 5](#) for Exponential Distribution

It is straightforward to get the formula for $\Delta$ from [Eq. (20)](#). Here we focus on the proof for $\Pi_{\epsilon,\omega_\Theta}$.
Similar to the proof in [App. D.1](#), according to [Lemma 1](#), we have

$$\Pi_{\epsilon,\omega_\Theta} = \mathbb{E}\left(\sup_{\hat{g}} \mathbb{P}\left(\hat{g}\left(\theta'\right) \in \left[g\left(\theta\right) - \epsilon, g\left(\theta\right) + \epsilon\right]\Big|\theta'\right)\right)$$

$$\leq \sup_{i \in \mathbb{N}, t' \in \mathbb{R}} \frac{\int_{\max\{0,t'\}}^{\min\left\{s, t' - \frac{2\epsilon}{\ln(1-\alpha)}\right\}} f_\lambda\left(\underline{\lambda} + i \cdot s + t\right) dt}{\int_0^s f_\lambda\left(\underline{\lambda} + i \cdot s + t\right) dt}$$

$$\leq \frac{\frac{2\epsilon}{-\ln(1-\alpha)} \cdot \left[\underline{c} + \mathcal{L}\left(s - x^* + \frac{\epsilon}{\ln(1-\alpha)}\right)\right]}{\underline{c}s + \frac{\mathcal{L}}{2}\left(s - x^*\right)^2},$$

where $x^* = s + \frac{\underline{c}}{\mathcal{L}} + \frac{\epsilon}{\ln(1-\alpha)} - \sqrt{\left(\frac{\underline{c}}{\mathcal{L}} + \frac{\epsilon}{\ln(1-\alpha)}\right)^2 + \frac{2\underline{c}s}{\mathcal{L}}}.$

### D.2.2  Proof of [Prop. 5](#) for Shifted Exponential Distribution

It is straightforward to get the formula for $\Delta$ from [Eq. (21)](#). Here we focus on the proof for $\Pi_{\epsilon,\omega_\Theta}$.
According to [Eq. (13)](#), we can bound the attack success rate $\Pi_{\epsilon,\omega_\Theta}$ as

$$\Pi_{\epsilon,\omega_\Theta} < \sup_{\theta \in S_{green}} \mathbb{P}(\hat{g}^*\left(\theta'\right) \in \left[g\left(\theta\right) - \epsilon, g\left(\theta\right) + \epsilon\right]) + \int_{\theta \in S_{yellow}} p(\theta)d\theta.$$

As for the first term $\sup_{\theta \in S_{green}} \mathbb{P}\left(\hat{g}^*\left(\theta'\right) \in \left[g\left(\theta\right) - \epsilon, g\left(\theta\right) + \epsilon\right]\right)$, we can get that

$$\sup_{\theta \in S_{green}} \mathbb{P}(\hat{g}^*\left(\theta'\right) \in \left[g\left(\theta\right) - \epsilon, g\left(\theta\right) + \epsilon\right])$$

$$= \sup_{i \in \mathbb{N}, h, t' \in \mathbb{R}} \frac{\int_{\max\left\{-\frac{s}{2}, t'\right\}}^{\min\left\{\frac{s}{2}, t' + \frac{2\epsilon}{|\ln(1-\alpha)+t_0|}\right\}} f_{\lambda,h}\left(\underline{\lambda} + (i + 0.5) \cdot s + t, h - t_0 \cdot t\right) dt}{\int_{-\frac{s}{2}}^{\frac{s}{2}} f_{\lambda,h}\left(\underline{\lambda} + (i + 0.5) \cdot s + t, h - t_0 \cdot t\right) dt}.$$

To analyze the above term, we provide the following lemma.

**Lemma 2.** *For a $\mathcal{L}$-Lipschitz continuous function $f(x), x \in [\underline{x}, \overline{x}]$, if $\int_{\underline{x}}^{\overline{x}} f(x)\mathrm{d}x = \mathcal{A}$ and $\inf_{x \in [\underline{x}, \overline{x}]} f(x) \geq \underline{c}$, it satisfies*

$$\sup_{x \in [\underline{x}, \overline{x}]} f(x) \leq \begin{cases} \frac{\mathcal{A}}{\overline{x} - \underline{x}} + \frac{\mathcal{L}(\overline{x} - \underline{x})}{2}, & \text{if } \underline{c} \leq \frac{\mathcal{A}}{\overline{x} - \underline{x}} - \frac{\mathcal{L}(\overline{x} - \underline{x})}{2} \\ \underline{c} + \sqrt{2\mathcal{L}\left(\mathcal{A} - \underline{c}\left(\overline{x} - \underline{x}\right)\right)}, & \text{if } \underline{c} > \frac{\mathcal{A}}{\overline{x} - \underline{x}} - \frac{\mathcal{L}(\overline{x} - \underline{x})}{2} \end{cases}$$
$$\triangleq M\left(\overline{x} - \underline{x}, \underline{c}, \mathcal{L}, \mathcal{A}\right).$$

The proof is in App. D.2.3.

Since $f_{\lambda, h}\left(\underline{\lambda} + (i + 0.5) \cdot s + t, h - t_0 \cdot t\right) = f_\lambda\left(\underline{\lambda} + (i + 0.5) \cdot s + t\right) \cdot f_h\left(h - t_0 \cdot t\right)$, according to Lemma 2, we can get that $f_{\lambda, h}$ is $\mathcal{L}_{\lambda, h}$-Lipschitz continuous, where

$$\mathcal{L}_{\lambda, h} = \mathcal{L}_\lambda \cdot M\left(\frac{\overline{h} - \underline{h}}{|t_0|}, \frac{k_h}{\overline{h} - \underline{h}}, |t_0|\mathcal{L}_h, \frac{1}{|t_0|}\right) + |t_0|\mathcal{L}_h \cdot M\left(\overline{\lambda} - \underline{\lambda}, \frac{k_\lambda}{\overline{\lambda} - \underline{\lambda}}, \mathcal{L}_\lambda, 1\right).$$

We can also get that

$$\inf_{a \in [\underline{\lambda}, \overline{\lambda}), b \in [\underline{h}, \overline{h})} f_{\lambda, h}\left(a, b\right) \geq \frac{k_h k_\lambda}{\left(\overline{h} - \underline{h}\right) \cdot \left(\overline{\lambda} - \underline{\lambda}\right)} \triangleq \underline{c}.$$

Therefore, according to Lemma 1, we can get that

$$\sup_{\theta \in S_{green}} \mathbb{P}\left(\hat{g}^*\left(\theta'\right) \in [g\left(\theta\right) - \epsilon, g\left(\theta\right) + \epsilon]\right)$$

$$= \sup_{i \in \mathbb{N}, h, t' \in \mathbb{R}} \frac{\int_{\max\left\{-\frac{s}{2}, t'\right\}}^{\min\left\{\frac{s}{2}, t' + \frac{2\epsilon}{|\ln(1-\alpha) + t_0|}\right\}} f_{\lambda, h}\left(\underline{\lambda} + (i + 0.5) \cdot s + t, h - t_0 \cdot t\right)\mathrm{d}t}{\int_{-\frac{s}{2}}^{\frac{s}{2}} f_{\lambda, h}\left(\underline{\lambda} + (i + 0.5) \cdot s + t, h - t_0 \cdot t\right)\mathrm{d}t}$$

$$\leq \frac{\frac{2\epsilon}{|\ln(1-\alpha) + t_0|} \cdot \left[\underline{c} + \mathcal{L}_{\lambda, h}\left(\frac{s}{2} - t^* - \frac{\epsilon}{|\ln(1-\alpha) + t_0|}\right)\right]}{\underline{c}s + \frac{\mathcal{L}_{\lambda, h}}{2}\left(\frac{s}{2} - t^*\right)^2},$$

where $t^* = \frac{s}{2} + \frac{\underline{c}}{\mathcal{L}_{\lambda, h}} - \frac{\epsilon}{|\ln(1-\alpha) + t_0|} - \sqrt{\left(\frac{\underline{c}}{\mathcal{L}_{\lambda, h}} - \frac{\epsilon}{|\ln(1-\alpha) + t_0|}\right)^2 + \frac{2\underline{c}s}{\mathcal{L}_{\lambda, h}}}$, $\mathcal{L}_{\lambda, h} = \mathcal{L}_\lambda \cdot M\left(\frac{\overline{h} - \underline{h}}{|t_0|}, \frac{k_h}{\overline{h} - \underline{h}}, |t_0|\mathcal{L}_h, \frac{1}{|t_0|}\right) + |t_0|\mathcal{L}_h \cdot M\left(\overline{\lambda} - \underline{\lambda}, \frac{k_\lambda}{\overline{\lambda} - \underline{\lambda}}, \mathcal{L}_\lambda, 1\right)$, and $\underline{c} = \frac{k_h k_\lambda}{\left(\overline{h} - \underline{h}\right) \cdot \left(\overline{\lambda} - \underline{\lambda}\right)}$.

As for $\int_{\theta \in S_{yellow}} p(\theta)d\theta$, we have

$$\int_{\theta \in S_{yellow}} p(\theta)d\theta$$

$$\leq M\left(\overline{h} - \underline{h}, \frac{k_h}{\overline{h} - \underline{h}}, \mathcal{L}_h, 1\right) \cdot M\left(\overline{\lambda} - \underline{\lambda}, \frac{k_\lambda}{\overline{\lambda} - \underline{\lambda}}, \mathcal{L}_\lambda, 1\right) \cdot \int_{\theta \in S_{yellow}} d\theta$$

$$= M\left(\overline{h} - \underline{h}, \frac{k_h}{\overline{h} - \underline{h}}, \mathcal{L}_h, 1\right) \cdot M\left(\overline{\lambda} - \underline{\lambda}, \frac{k_\lambda}{\overline{\lambda} - \underline{\lambda}}, \mathcal{L}_\lambda, 1\right) \cdot \left(\overline{\lambda} - \underline{\lambda}\right)|t_0|s.$$

Above all, we can get that

$$\Pi_{\epsilon, \omega_\Theta} < \sup_{\theta \in S_{green}} \mathbb{P}\left(\hat{g}^*\left(\theta'\right) \in [g\left(\theta\right) - \epsilon, g\left(\theta\right) + \epsilon]\right) + \int_{\theta \in S_{yellow}} p(\theta)d\theta.$$

$$\leq \frac{\frac{2\epsilon}{|\ln(1-\alpha) + t_0|} \cdot \left[\underline{c} + \mathcal{L}_{\lambda, h}\left(\frac{s}{2} - t^* - \frac{\epsilon}{|\ln(1-\alpha) + t_0|}\right)\right]}{\underline{c}s + \frac{\mathcal{L}_{\lambda, h}}{2}\left(\frac{s}{2} - t^*\right)^2} +$$

$$M\left(\overline{h} - \underline{h}, \frac{k_h}{\overline{h} - \underline{h}}, \mathcal{L}_h, 1\right) \cdot M\left(\overline{\lambda} - \underline{\lambda}, \frac{k_\lambda}{\overline{\lambda} - \underline{\lambda}}, \mathcal{L}_\lambda, 1\right) \cdot \left(\overline{\lambda} - \underline{\lambda}\right)|t_0|s,$$

where $M\left(\cdot, \cdot, \cdot, \cdot\right), \underline{c}, \mathcal{L}_{\lambda, h}, t^*$ are defined as above.

### D.2.3 Proof of Lemma 2

Without loss of generality, we assume that $f(\overline{x}) \geq f(\underline{x})$. Based on simple geometric analysis, we can get that there are two patterns when $\sup_{x \in [\underline{x}, \overline{x}]} f(x)$ achieves supremum, which are shown in Fig. 10.

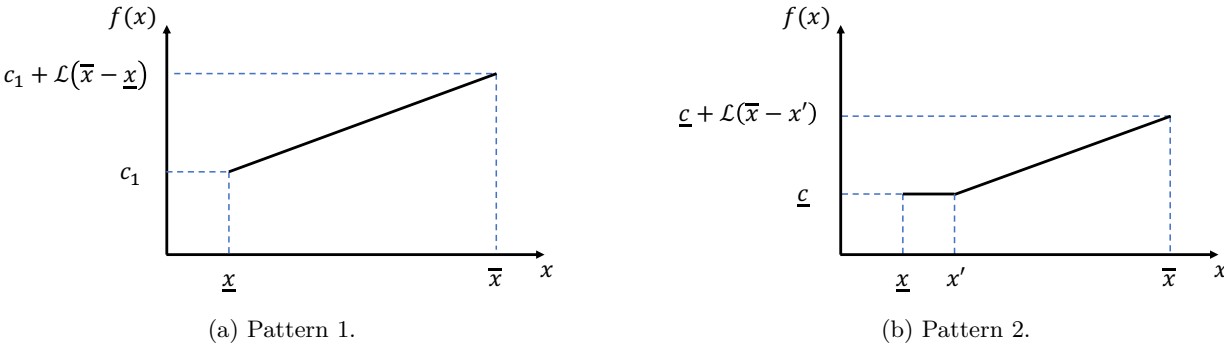

(a) Pattern 1.  (b) Pattern 2.

Figure 10: Two patterns when $\sup_{x \in [\underline{x}, \overline{x}]} f(x)$ achieves supremum.

For pattern 1, $f(\underline{x}) = c_1 \geq \underline{c}$, $f(\overline{x}) = c_1 + \mathcal{L}(\overline{x} - \underline{x})$, and $\int_{\underline{x}}^{\overline{x}} f(x)\mathrm{d}x = \left(c_1 + \frac{\mathcal{L}}{2}(\overline{x} - \underline{x})\right) \cdot (\overline{x} - \underline{x}) = \mathcal{A}$. Therefore, when $\underline{c} \leq \frac{\mathcal{A}}{\overline{x} - \underline{x}} - \frac{\mathcal{L}(\overline{x} - \underline{x})}{2}$, we have

$$\sup_{f} \sup_{x \in [\underline{x}, \overline{x}]} f(x) = c_1 + \mathcal{L}(\overline{x} - \underline{x}) = \frac{\mathcal{A}}{\overline{x} - \underline{x}} + \frac{\mathcal{L}(\overline{x} - \underline{x})}{2}.$$

For pattern 2, $f(\underline{x}) = \underline{c}$, $f(\overline{x}) = \underline{c} + \mathcal{L}(\overline{x} - x')$, where $x' \in (\underline{x}, \overline{x}]$, and $\int_{\underline{x}}^{\overline{x}} f(x)\mathrm{d}x = \underline{c}(\overline{x} - \underline{x}) + \frac{\mathcal{L}}{2}(\overline{x} - x')^2 = \mathcal{A}$. Therefore, when $\underline{c} > \frac{\mathcal{A}}{\overline{x} - \underline{x}} - \frac{\mathcal{L}(\overline{x} - \underline{x})}{2}$, we have

$$\sup_{f} \sup_{x \in [\underline{x}, \overline{x}]} f(x) = \underline{c} + \mathcal{L}(\overline{x} - x') = \underline{c} + \sqrt{2\mathcal{L}(\mathcal{A} - \underline{c}(\overline{x} - \underline{x}))}.$$

Above all, we can get that

$$\sup_{x \in [\underline{x}, \overline{x}]} f(x) \leq \sup_{f} \sup_{x \in [\underline{x}, \overline{x}]} f(x)$$

$$= \begin{cases} \frac{\mathcal{A}}{\overline{x} - \underline{x}} + \frac{\mathcal{L}(\overline{x} - \underline{x})}{2}, & \text{if } \underline{c} \leq \frac{\mathcal{A}}{\overline{x} - \underline{x}} - \frac{\mathcal{L}(\overline{x} - \underline{x})}{2} \\ \underline{c} + \sqrt{2\mathcal{L}(\mathcal{A} - \underline{c}(\overline{x} - \underline{x}))}, & \text{if } \underline{c} > \frac{\mathcal{A}}{\overline{x} - \underline{x}} - \frac{\mathcal{L}(\overline{x} - \underline{x})}{2} \end{cases}.$$

## E  Discrete Distribution with Secret = Mean

Here, we consider three typical examples of discrete distributions: geometric distributions, binomial distributions, and Poisson distributions with parameter $\theta$. More specifically, the original distribution is

$$\mathbb{P}(X_\theta = k) = \begin{cases} (1 - \theta)^k \theta & \text{(geometric distribution)} \\ \binom{n}{k}\theta^k(1 - \theta)^{n-k} & \text{(binomial distribution)} \\ \frac{\theta^k e^{-\theta}}{k!} & \text{(Poisson distribution)} \end{cases}$$

where $n$ standards for the number of trials in binomial distribution. The support of the parameter is $\mathrm{Supp}(\Theta) = \left\{X_\theta : \theta \in (\underline{\theta}, \overline{\theta}]\right\}$ where $(\underline{\theta}, \overline{\theta}] \subseteq (0, 1)$ for geometric distribution and binomial distribution, and $(\underline{\theta}, \overline{\theta}] \subseteq (0, \infty)$ for Poisson distribution.

We first analyze the lower bound.

**Corollary 3** (Privacy lower bound, secret = mean of a discrete distribution)**.** *Consider the secret function* $g(\theta) = \sum_x x f_{X_\theta}(x)$. *For any* $T \in (0,1)$, *when* $\Pi_{\epsilon, \omega_\Theta} \leq T$, *we have* $\Delta > \left(\lceil \frac{1}{T} \rceil - 1\right) \cdot 2\gamma\epsilon$, *where the value of* $\gamma$ *depends on the type of the distributions:*

- *Geometric:*

$$\gamma = \inf_{\underline{\theta} < \theta_1 < \theta_2 \leq \overline{\theta}} \frac{(1 - \theta_2)^{h(\theta_1, \theta_2)} - (1 - \theta_1)^{h(\theta_1, \theta_2)}}{2\left(\frac{1}{\theta_2} - \frac{1}{\theta_1}\right)},$$

*where* $h(\theta_1, \theta_2) = \lfloor \frac{\log(\theta_2) - \log(\theta_1)}{\log(1 - \theta_1) - \log(1 - \theta_2)} \rfloor + 1$.

- *Binomial:*

$$\gamma = \inf_{\underline{\theta} < \theta_1 < \theta_2 \leq \overline{\theta}}$$

$$\frac{I_{1 - \theta_2}(n - h(\theta_1, \theta_2), 1 + h(\theta_1, \theta_2)) - I_{1 - \theta_1}(n - h(\theta_1, \theta_2), 1 + h(\theta_1, \theta_2))}{2n(\theta_1 - \theta_2)},$$

*where* $h(\theta_1, \theta_2) = \lfloor k' \rfloor$, $k' = n \ln\left(\frac{1 - \theta_2}{1 - \theta_1}\right) \Big/ \ln\left(\frac{\theta_1(1 - \theta_2)}{\theta_2(1 - \theta_1)}\right)$, *and* $I$ *represents the regularized incomplete beta function.*

- *Poisson:*

$$\gamma = \inf_{\underline{\theta} < \theta_1 < \theta_2 \leq \overline{\theta}} \frac{Q(h(\theta_1, \theta_2), \theta_2) - Q(h(\theta_1, \theta_2), \theta_1)}{2(\theta_1 - \theta_2)},$$

*where* $h(\theta_1, \theta_2) = \lfloor \frac{\theta_1 - \theta_2}{\ln(\theta_1) - \ln(\theta_2)} \rfloor + 1$ *and* $Q$ *is the regularized gamma function.*

The proof is in App. E.1. The above lower bounds can be computed numerically.

Since these distributions only have one parameter, we can use Alg. 1 and Alg. 3 to derive a data release mechanism. The performance of greedy-based and dynamic-programming-based data release mechanisms for each distribution is shown in Fig. 11.

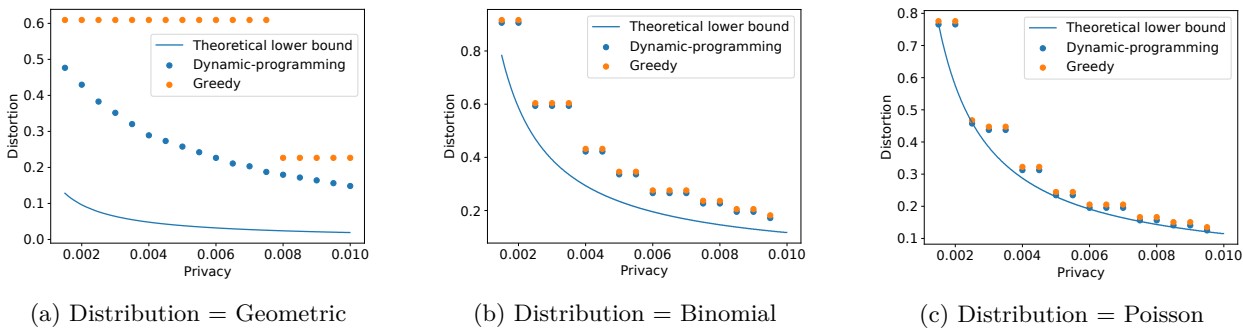

(a) Distribution = Geometric  (b) Distribution = Binomial  (c) Distribution = Poisson

Figure 11: Privacy-distortion performance of Alg. 1 and Alg. 3 for geometric, binomial and Poisson distribution when secret = mean.

As we can observe, the distortion that dynamic-programming-based data release mechanism achieves it is always smaller than or equal to that of the greedy-based data release mechanism.

### E.1 Proof of Corollary 3

### E.1.1 Geometric Distribution

*Proof.* Let $X_{\theta_1}$ and $X_{\theta_2}$ be two Geometric random variables with parameters $\theta_1$ and $\theta_2$ respectively. We assume that $\theta_1 > \theta_2$ without loss of generality. Let $k'$ satisfy $(1 - \theta_1)^{k'} \theta_1 = (1 - \theta_2)^{k'} \theta_2$ and $k_0 = \lfloor k' \rfloor + 1$.

Then we can get that

$$
\begin{aligned}
D\left(X_{\theta_1}, X_{\theta_2}\right) &= \frac{1}{2} d_{\mathrm{TV}}\left(\omega_{X_{\theta_1}} \| \omega_{X_{\theta_2}}\right) \\
&= \frac{1}{2}\left(1-\theta_2\right)^{k_0} - \frac{1}{2}\left(1-\theta_1\right)^{k_0}, \\
R\left(X_{\theta_1}, X_{\theta_2}\right) &= \frac{1}{\theta_2} - \frac{1}{\theta_1}.
\end{aligned}
$$

Therefore, we have

$$
\gamma = \inf_{\underline{\theta}<\theta_1<\theta_2\leq\bar{\theta}} \frac{\left(1-\theta_2\right)^{k_0} - \left(1-\theta_1\right)^{k_0}}{2\left(\frac{1}{\theta_2} - \frac{1}{\theta_1}\right)}.
$$

The rest follows from Thm. 1. $\qquad\square$

### E.1.2 Binomial Distribution

*Proof.* Let $X_{\theta_1}$ and $X_{\theta_2}$ be two binomial random variables with parameters $\theta_1$ and $\theta_2$ respectively with fixed number of trials $n$. We assume that $\theta_1 > \theta_2$ without loss of generality. Let $k'$ satisfy $\binom{n}{k'}\theta_1^{k'}\left(1-\theta_1\right)^{n-k'} = \binom{n}{k'}\theta_2^{k'}\left(1-\theta_1\right)^{n-k'}$ and $k_0 = \lfloor k' \rfloor$. We can get that

$$
\begin{aligned}
D\left(X_{\theta_1}, X_{\theta_2}\right) &= \frac{1}{2} d_{\mathrm{TV}}\left(\omega_{X_{\theta_1}} \| \omega_{X_{\theta_2}}\right) \\
&= \frac{1}{2} I_{1-\theta_2}\left(n-k_0, 1+k_0\right) - \frac{1}{2} I_{1-\theta_1}\left(n-k_0, 1+k_0\right), \\
R\left(X_{\theta_1}, X_{\theta_2}\right) &= n\left(\theta_1-\theta_2\right),
\end{aligned}
$$

where $I$ represents the regularized incomplete beta function.

Therefore, we have

$$
\gamma = \inf_{\underline{\theta}<\theta_1<\theta_2\leq\bar{\theta}} \frac{I_{1-\theta_2}\left(n-k_0, 1+k_0\right) - I_{1-\theta_1}\left(n-k_0, 1+k_0\right)}{2n\left(\theta_1-\theta_2\right)}.
$$

The rest follows from Thm. 1. $\qquad\square$

### E.1.3 Poisson Distribution

*Proof.* Let $X_{\theta_1}$ and $X_{\theta_2}$ be two Poisson random variables with parameters $\theta_1$ and $\theta_2$ respectively. We assume that $\theta_1 > \theta_2$ without loss of generality. Let $k'$ satisfy $\theta_1^{k'} e^{-\theta_1} = \theta_2^{k'} e^{-\theta_2}$ and $k_0 = \lfloor k' \rfloor + 1$. Then we can get that

$$
\begin{aligned}
D\left(X_{\theta_1}, X_{\theta_2}\right) &= \frac{1}{2} d_{\mathrm{TV}}\left(\omega_{X_{\theta_1}} \| \omega_{X_{\theta_2}}\right) \\
&= \frac{1}{2} Q\left(k_0, \theta_2\right) - \frac{1}{2} Q\left(k_0, \theta_1\right), \\
R\left(X_{\theta_1}, X_{\theta_2}\right) &= \theta_1-\theta_2,
\end{aligned}
$$

where $Q$ is the regularized gamma function.

Therefore, we have

$$
\gamma = \inf_{\underline{\theta}<\theta_1<\theta_2\leq\bar{\theta}} \frac{Q\left(k_0, \theta_2\right) - Q\left(k_0, \theta_1\right)}{2\left(\theta_1-\theta_2\right)}.
$$

The rest follows from Thm. 1. $\qquad\square$

# F More Distributions with Secret = Quantiles

In this section, we discuss how to protect the quantiles for typical examples of continuous distributions: Gaussian distributions and uniform distributions. In our analysis, their parameters are denoted by:

- Gaussian distributions: $\theta = (\mu, \sigma)$, where $\mu, \sigma$ are the mean and the standard deviation of the Gaussian distribution.
- Uniform distributions: $\theta = (m, n)$, where $m, n$ denote the lower and upper bound of the uniform distribution. In other words, $X_{m,n}$ is a random variable from uniform distribution $U([m, n])$.

As before, we first present the lower bound.

**Corollary 4** (Privacy lower bound, secret = $\alpha$-quantile of a continuous distribution). *Consider the secret function $g(\theta) = \alpha$-quantile of $f_{X_\theta}$. For any $T \in (0, 1)$, when $\Pi_{\epsilon, \omega_\Theta} \leq T$, we have $\Delta > \left(\lceil \frac{1}{T} \rceil - 1\right) \cdot 2\gamma\epsilon$, where the value of $\gamma$ depends on the type of the distributions:*

- *Gaussian:*

$$\gamma = \min_t \frac{\sqrt{\frac{1}{2\pi}} e^{-\frac{1}{2}t^2} - t\left(\frac{1}{2} - \Phi(t)\right)}{|t + Q_\alpha|},$$

*where $\Phi$ denotes the CDF of the standard Gaussian distribution and $Q_\alpha \triangleq \Phi^{-1}(\alpha)$.*
- *Uniform:*

$$\gamma = \begin{cases} \sqrt{\alpha^2 - \alpha + \frac{1}{2}} + \alpha - \frac{1}{2} & \alpha \leq 0.5 \\ \sqrt{\alpha^2 - \alpha + \frac{1}{2}} - \alpha + \frac{1}{2} & \alpha > 0.5 \end{cases}.$$

The proof is in [App. F.1](#). The bound for uniform is in closed form, while the bound for Gaussian can be computed numerically.

Next, we provide data release mechanisms for each of the distributions. Here, we assume that the parameters of the original data are drawn from a uniform distribution with lower and upper bounds. In more details, we make the following assumptions.

***Assumption 5.*** *The prior over distribution parameters as specified below.*

- *Gaussian: $(\mu, \sigma)$ follows the uniform distribution over $\left\{ (a, b) \mid a \in [\underline{\mu}, \overline{\mu}], b \in [\underline{\sigma}, \overline{\sigma}) \right\}$.*

- *Uniform: $(M, N)$ follows the uniform distribution over $\left\{ (a, b) \mid a \in [\underline{m}, \overline{m}), b \in [\underline{m}, \overline{m}), a < b \right\}$.*

***Mechanism 3*** (For secret = quantile of a continuous distribution). *We design mechanisms for each of the distributions.*

- *Gaussian:*

$$\mathcal{S}_{\mu, i} = \left\{ (\mu + t_0 \cdot t, \underline{\sigma} + (i + 0.5) \cdot s + t) \mid t \in \left[ -\frac{s}{2}, \frac{s}{2} \right) \right\},$$
$$\theta^*_{\mu, i} = (\mu, \underline{\sigma} + (i + 0.5) \cdot s),$$
$$\mathcal{I} = \{(\mu, i) : i \in \mathbb{N}, \mu \in \mathbb{R}\},$$

*where $s$ is a hyper-parameter of the mechanism that divides $(\overline{\sigma} - \underline{\sigma})$ and*

$$t_0 = \arg\min_t \frac{\sqrt{\frac{1}{2\pi}} e^{-\frac{1}{2}t^2} - t\left(\frac{1}{2} - \Phi(t)\right)}{|t + Q_\alpha|}.$$

.
- *Uniform:*

$$\mathcal{S}_{m, i} = \left\{ (m - t_0 \cdot t, m + (i + 0.5) \cdot s + t) \mid t \in \left( -\frac{s}{2(t_0 + 1)}, \frac{s}{2(t_0 + 1)} \right] \right\},$$
$$\theta^*_{m, i} = (m, m + (i + 0.5) \cdot s),$$
$$\mathcal{I} = \{(m, i) \mid i \in \mathbb{Z}_{>0}, m \in \mathbb{R}\},$$

*where* $t_0 = \frac{1}{\frac{1}{l}-1}$ *for*

$$l = \begin{cases} \alpha + \sqrt{\alpha^2 - \alpha + \frac{1}{2}} & \alpha \leq 0.5 \\ \alpha - \sqrt{\alpha^2 - \alpha + \frac{1}{2}} & \alpha > 0.5 \end{cases}.$$

*and $s > 0$ is a hyper-parameter of the mechanism that divides $(\overline{m} - \underline{m})$.*

These data release mechanisms achieve the following $\Delta$ and $\Pi_{\epsilon,\omega_\Theta}$.

**Proposition 6.** *Under [Asm. 5](), [Mech. 3]() has the following $\Delta$ and $\Pi_{\epsilon,\omega_\Theta}$ value/bound.*

- *Gaussian:*

$$\Pi_{\epsilon,\omega_\Theta} < \frac{2\epsilon}{|t_0 + Q_\alpha|s} + \frac{|t_0|s}{\overline{\mu} - \underline{\mu}},$$

$$\Delta = \frac{s}{2}\sqrt{\frac{2}{\pi}}e^{-\frac{1}{2}t_0^2} - \frac{t_0 s}{2}\left(1 - 2\Phi\left(t_0\right)\right) < \left(2 + \frac{|t_0| \cdot |t_0 + Q_\alpha|s^2}{\left(\overline{\mu} - \underline{\mu}\right)\epsilon}\right)\Delta_{opt}.$$

*Under the "high-precision" regime where $\frac{s^2}{\overline{\mu}-\underline{\mu}} \to 0$ as $s, (\overline{\mu} - \underline{\mu}) \to \infty$, $\Delta$ satisfies*

$$\lim_{\frac{s^2}{\overline{\mu}-\underline{\mu}} \to 0} \sup \Delta < 3\Delta_{opt}.$$

- *Uniform:*

$$\Pi_{\epsilon,\omega_\Theta} < \frac{2\epsilon\left(t_0 + 1\right)}{|\left(1 - \alpha\right)t_0 - \alpha|s} + \frac{2s \cdot t_0}{\left(t_0 + 1\right)\left(\overline{m} - \underline{m}\right)} + \frac{s^2}{2\left(\overline{m} - \underline{m}\right)^2},$$

$$\Delta = \frac{\left(t_0^2 + 1\right)s}{4(t_0 + 1)^2}$$

$$< \left(2 + \frac{|\left(1 - \alpha\right)t_0 - \alpha|s}{\epsilon\left(t_0 + 1\right)} \cdot \left(\frac{2s \cdot t_0}{\left(t_0 + 1\right)\left(\overline{m} - \underline{m}\right)} + \frac{s^2}{2\left(\overline{m} - \underline{m}\right)^2}\right)\right)\Delta_{opt}.$$

*Under the "high-precision" regime where $\frac{s^2}{\overline{m}-\underline{m}} \to 0$ as $s, (\overline{m} - \underline{m}) \to \infty$, $\Delta$ satisfies*

$$\lim_{\frac{s^2}{\overline{m}-\underline{m}} \to 0} \sup \Delta < 3\Delta_{opt}.$$

*The $t_0$ parameter is defined in [Mech. 3]() for each distribution.*

The proof is in [App. F.2](). For Gaussian distribution, we relax [Asm. 5]() and analyze the privacy-distortion performance of [Mech. 3]() in [App. F.3](). For both distributions, we consider the "high-precision" regime. The two takeaways are that: (1) data holder can use $s$ to control the trade-off between distortion and privacy, and (2) the mechanism is order-optimal with multiplicative factor 3.

### F.1  Proof of [Corollary 4]()

#### F.1.1  Gaussian Distribution

*Proof.* Let $X_{\mu_1,\sigma_2}, X_{\mu_2,\sigma_2}$ be two Gaussian random variables with means $\mu_1, \mu_2$ and sigmas $\sigma_1, \sigma_2$ respectively. Let $\Phi$ denotes the CDF of the standard Gaussian distribution and let $\Phi^{-1}(\alpha) \triangleq Q_\alpha$.

When $\sigma_1 = \sigma_2$, we have

$$\frac{D\left(X_{\mu_1,\sigma_1}, X_{\mu_2,\sigma_2}\right)}{R\left(X_{\mu_1,\sigma_1}, X_{\mu_2,\sigma_2}\right)} = \frac{\frac{1}{2}|\mu_1 - \mu_2|}{|\mu_1 + \sigma Q_\alpha - (\mu_2 + \sigma Q_\alpha)|} = \frac{1}{2}.$$

When $\sigma_1 \neq \sigma_2$, we assume $\sigma_2 > \sigma_1$ without loss of generality. Let $a = \frac{\sigma_1}{\sigma_2}$ and $b = \frac{\sigma_2}{\sigma_1}\mu_1 - \mu_2$. Let $a = \frac{\sigma_1}{\sigma_2}$ and $b = \frac{\sigma_2}{\sigma_1}\mu_1 - \mu_2$. We can get that $f_{X_{\mu_1,\sigma_1}}(x) = a f_{X_{\mu_2,\sigma_2}}(a(x+b))$, and

$$
\begin{aligned}
D\left(X_{\mu_1,\sigma_1}, X_{\mu_2,\sigma_2}\right) &= \frac{1}{2} d_{\text{Wasserstein-1}}\left(\omega_{X_{\mu_1,\sigma_1}} \| \omega_{X_{\mu_2,\sigma_2}}\right) \\
&= \frac{1}{2}\int_{-\infty}^{+\infty}\left|x - \left(\frac{x}{a} - b\right)\right| f_{X_{\mu_1,\sigma_1}}(x)\,\mathrm{d}x \\
&= (\mu_1 - \mu_2)\left(\Phi\left(\frac{\mu_1 - \mu_2}{\sigma_2 - \sigma_1}\right) - \frac{1}{2}\right) \\
&\quad + \sqrt{\frac{1}{2\pi}}\,(\sigma_2 - \sigma_1)\, e^{-\frac{1}{2}\left(\frac{\mu_1 - \mu_2}{\sigma_2 - \sigma_1}\right)^2}, \\
R\left(X_{\mu_1,\sigma_1}, X_{\mu_2,\sigma_2}\right) &= \left|\mu_1 + \sigma_1 Q_\alpha - (\mu_2 + \sigma_2 Q_\alpha)\right| \\
&= \left|(\mu_1 - \mu_2) + (\sigma_1 - \sigma_2) Q_\alpha\right|.
\end{aligned}
\tag{22}
$$

Let $\frac{\mu_1 - \mu_2}{\sigma_1 - \sigma_2} \triangleq t$, we can get that

$$
\frac{D\left(X_{\mu_1,\sigma_1}, X_{\mu_2,\sigma_2}\right)}{R\left(X_{\mu_1,\sigma_1}, X_{\mu_2,\sigma_2}\right)} = \frac{\sqrt{\frac{1}{2\pi}}e^{-\frac{1}{2}t^2} - t\left(\frac{1}{2} - \Phi(t)\right)}{|t + Q_\alpha|} \triangleq h(t).
$$

Since $\lim_{t\to\infty} = \frac{1}{2}$, we have $\min\left\{\min_t h(t), \frac{1}{2}\right\} = \min_t h(t)$, and therefore we can get that

$$
\gamma = \min_t h(t).
$$

$\square$

### F.1.2   Uniform Distribution

*Proof.* Let $X_{m_1,n_1}, X_{m_2,n_2}$ be two uniform random variables. Let $F_{X_{m_1,n_1}}, F_{X_{m_2,n_2}}$ be their CDFs, and let $m_2 \geq m_1$ without loss of generality. We can get that

$$
\begin{aligned}
D\left(X_{m_1,n_1}, X_{m_2,n_2}\right) &= \frac{1}{2} d_{\text{Wasserstein-1}}\left(\omega_{X_{m_1,n_1}} \| \omega_{X_{m_2,n_2}}\right) \\
&= \frac{1}{2}\int_{-\infty}^{+\infty}\left|F_{X_{m_1,n_1}}(x) - F_{X_{m_2,n_2}}(x)\right|\mathrm{d}x \\
&= \begin{cases} \frac{m_2 - m_1 + n_2 - n_1}{4} & n_2 \geq n_1 \\ \frac{(m_2 - m_1)^2 + (n_1 - n_2)^2}{4(m_2 - m_1 + (n_1 - n_2))} & n_2 < n_1 \end{cases}, \\
R\left(X_{m_1,n_1}, X_{m_2,n_2}\right) &= \left|m_2 + \alpha(n_2 - m_2) - [m_1 + \alpha(n_1 - m_1)]\right| \\
&= \left|(1 - \alpha)(m_2 - m_1) + \alpha(n_2 - n_1)\right|.
\end{aligned}
\tag{23}
$$

When $n_2 = n_1$, we have

$$
\frac{D\left(X_{m_1,n_1}, X_{m_2,n_2}\right)}{R\left(X_{m_1,n_1}, X_{m_2,n_2}\right)} = \frac{m_2 - m_1}{4(1-\alpha)(m_2 - m_1)} = \frac{1}{4(1-\alpha)}.
$$

When $n_2 > n_1$, let $t_1 = \frac{m_2-m_1}{n_2-n_1} \in [0,+\infty)$, we have

$$
\begin{aligned}
\frac{D\left(X_{m_1,n_1}, X_{m_2,n_2}\right)}{R\left(X_{m_1,n_1}, X_{m_2,n_2}\right)} &= \frac{1}{4}\frac{m_2-m_1+n_2-n_1}{(1-\alpha)(m_2-m_1)+\alpha(n_2-n_1)} \\
&= \frac{1}{4}\frac{t_1+1}{(1-\alpha)t_1+\alpha} \\
&= \frac{1}{4(1-\alpha)}\left(1+\frac{1-2\alpha}{1-\alpha}\cdot\frac{1}{t_1+\frac{\alpha}{1-\alpha}}\right) \\
&\geq \begin{cases} \frac{1}{4(1-\alpha)} & \alpha \leq 0.5 \\ \frac{1}{4\alpha} & \alpha > 0.5 \end{cases}.
\end{aligned}
$$

When $n_2 < n_1$, let $t_2 = \frac{m_2-m_1}{n_1-n_2} \in (0,+\infty)$, we have

$$
\begin{aligned}
\frac{D\left(X_{m_1,n_1}, X_{m_2,n_2}\right)}{R\left(X_{m_1,n_1}, X_{m_2,n_2}\right)} &= \frac{1}{4}\frac{(m_2-m_1)^2+(n_1-n_2)^2}{(m_2-m_1+(n_1-n_2))}\cdot \\
&\quad \frac{1}{|(1-\alpha)(m_2-m_1)-\alpha(n_1-n_2)|} \\
&= \frac{1}{4}\frac{t_2^2+1}{(t_2+1)|(1-\alpha)t_2-\alpha|} \\
&\geq \begin{cases} \sqrt{\alpha^2-\alpha+\frac{1}{2}}+\alpha-\frac{1}{2} & \alpha \leq 0.5 \\ \sqrt{\alpha^2-\alpha+\frac{1}{2}}-\alpha+\frac{1}{2} & \alpha > 0.5 \end{cases}.
\end{aligned}
$$

"=" achieves when $t_2 = \frac{1}{\frac{1}{l}-1} \triangleq t_0$, where

$$
l = \begin{cases} \alpha + \sqrt{\alpha^2-\alpha+\frac{1}{2}} & \alpha \leq 0.5 \\ \alpha - \sqrt{\alpha^2-\alpha+\frac{1}{2}} & \alpha > 0.5 \end{cases}.
$$

Therefore we can get that

$$
\gamma = \begin{cases} \sqrt{\alpha^2-\alpha+\frac{1}{2}}+\alpha-\frac{1}{2} & \alpha \leq 0.5 \\ \sqrt{\alpha^2-\alpha+\frac{1}{2}}-\alpha+\frac{1}{2} & \alpha > 0.5 \end{cases}.
$$

$\square$

### F.2   Proof of Prop. 6

#### F.2.1   Gaussian Distribution

*Proof.* We first focus on the proof for $\Pi_{\epsilon,\omega_\Theta}$.

In Fig. 12, we separate the space of possible data parameters into two regions represented by yellow and green colors. The yellow regions $S_{yellow}$ constitute right triangles with height $s$ and width $|t_0|s$. The green region $S_{green}$ is the rest of the parameter space. The high-level idea of our proof is as follows. Note that for any parameter $\theta \in S_{green}$, there exists a $\mathcal{S}_{\mu,i}$ s.t. $\theta \in \mathcal{S}_{\mu,i}$ and $\mathcal{S}_{\mu,i} \subset S_{green}$. Therefore, we can bound the attack success rate if $\theta \in S_{green}$. At the same time, the probability of $\theta \in S_{yellow}$ is bounded. Therefore, we can bound the overall attacker's success rate (i.e., $\Pi_{\epsilon,\omega_\Theta}$). More specifically, let the optimal attacker be $\hat{g}^*$.

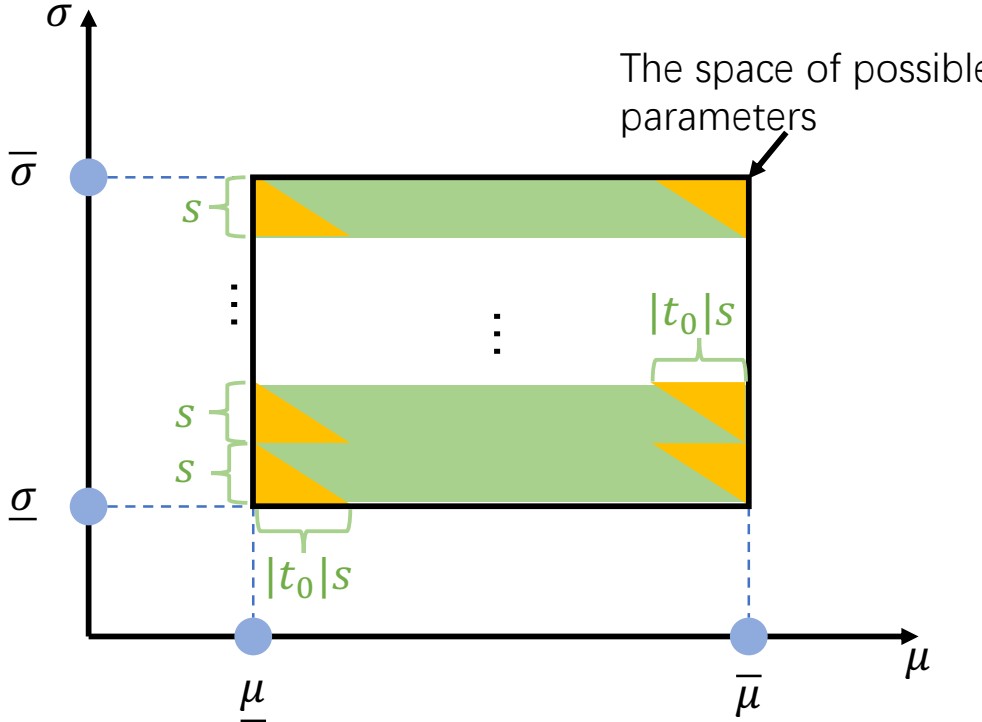

Figure 12: The construction for proof of Prop. 6 for Gaussian distributions. We separate the space of possible parameters into two regions (yellow and green) and bound the attacker's success rate on each region separately.

We have

$$
\begin{aligned}
\Pi_{\epsilon,\omega_\Theta} &= \mathbb{P}\left(\hat{g}^*\left(\theta'\right) \in \left[g\left(\theta\right) - \epsilon, g\left(\theta\right) + \epsilon\right]\right) \\
&= \int_{\theta \in S_{green}} p(\theta)\mathbb{P}\left(\hat{g}^*\left(\theta'\right) \in \left[g\left(\theta\right) - \epsilon, g\left(\theta\right) + \epsilon\right]\right) d\theta \\
&\quad + \int_{\theta \in S_{yellow}} p(\theta)\mathbb{P}\left(\hat{g}^*\left(\theta'\right) \in \left[g\left(\theta\right) - \epsilon, g\left(\theta\right) + \epsilon\right]\right) d\theta \\
&< \frac{2\epsilon}{|t_0 + Q_\alpha|s} + \frac{|t_0|s}{\overline{\mu} - \underline{\mu}}.
\end{aligned}
$$

For the distortion, it is straightforward to get that $\Delta = \frac{s}{2}\sqrt{\frac{2}{\pi}}e^{-\frac{1}{2}t_0^2} - \frac{t_0 s}{2}\left(1 - 2\Phi\left(t_0\right)\right)$ from Eq. (22), and $\Delta_{\mathrm{opt}} > \left(\lceil\frac{1}{\Pi_{\epsilon,\omega_\Theta}}\rceil - 1\right)\cdot 2\gamma\epsilon \geq 2\gamma\epsilon$, where $\gamma$ is defined in Corollary 4. We can get that $\left(\Pi_{\epsilon,\omega_\Theta} - \frac{|t_0|s}{\overline{\mu} - \underline{\mu}}\right)\cdot\Delta < 2\gamma\epsilon$

and

$$\Delta = \Delta_{\text{opt}} + \Delta - \Delta_{\text{opt}}$$

$$< \Delta_{\text{opt}} + \Delta - \left( \lceil \frac{1}{\Pi_{\epsilon,\omega_\Theta}} \rceil - 1 \right) \cdot 2\gamma\epsilon$$

$$\leq \Delta_{\text{opt}} + 2\gamma\epsilon + \Delta - \frac{2\gamma\epsilon}{\Pi_{\epsilon,\omega_\Theta}}$$

$$< \Delta_{\text{opt}} + 2\gamma\epsilon + \frac{\frac{|t_0|s}{\overline{\mu}-\underline{\mu}}}{\frac{2\epsilon}{|t_0+Q_\alpha|s} + \frac{|t_0|s}{\overline{\mu}-\underline{\mu}}} \cdot \Delta$$

$$= \left( 1 + \frac{|t_0| \cdot |t_0+Q_\alpha|s^2}{2\epsilon\left(\overline{\mu}-\underline{\mu}\right)} \right) (\Delta_{\text{opt}} + 2\gamma\epsilon)$$

$$\leq \left( 2 + \frac{|t_0| \cdot |t_0+Q_\alpha|s^2}{\epsilon\left(\overline{\mu}-\underline{\mu}\right)} \right) \Delta_{\text{opt}}.$$

$\square$

### F.2.2 Uniform Distribution

*Proof.* We first focus on the proof for $\Pi_{\epsilon,\omega_\Theta}$.

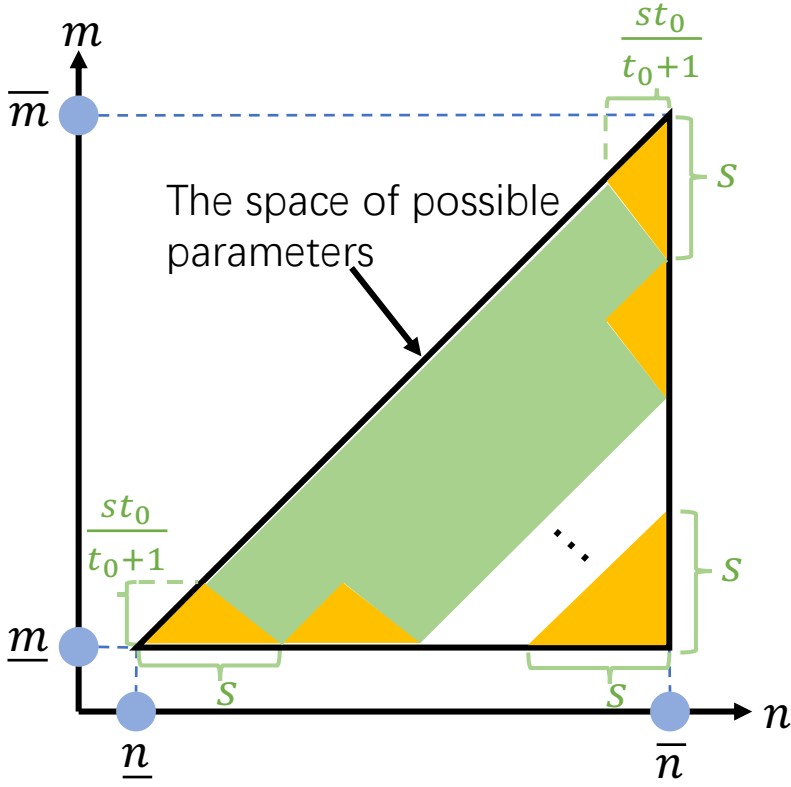

Figure 13: The construction for proof of Prop. 6 for uniform distributions. We separate the space of possible parameters into two regions (yellow and green) and bound the attacker's success rate on each region separately.

In Fig. 13, we separate the space of possible data parameters into two regions represented by yellow and green colors. The yellow regions $S_{yellow}$ constitute triangles with height $\frac{st_0}{t_0+1}$ and width $s$ (except for the

right-bottom triangle with height and width $s$). The green region $S_{green}$ is the rest of the parameter space. The high-level idea of our proof is as follows. Note that for any parameter $\theta \in S_{green}$, there exists a $\mathcal{S}_{\mu,i}$ s.t. $\theta \in \mathcal{S}_{\mu,i}$ and $\mathcal{S}_{\mu,i} \subset S_{green}$. Therefore, we can bound the attack success rate if $\theta \in S_{green}$. At the same time, the probability of $\theta \in S_{yellow}$ is bounded. Therefore, we can bound the overall attacker's success rate (i.e., $\Pi_{\epsilon,\omega_\Theta}$). More specifically, let the optimal attacker be $\hat{g}^*$. We have

$$
\begin{aligned}
\Pi_{\epsilon,\omega_\Theta} &= \mathbb{P}\left(\hat{g}^*\left(\theta'\right) \in [g\left(\theta\right) - \epsilon, g\left(\theta\right) + \epsilon]\right) \\
&= \int_{\theta \in S_{green}} p(\theta)\mathbb{P}\left(\hat{g}^*\left(\theta'\right) \in [g\left(\theta\right) - \epsilon, g\left(\theta\right) + \epsilon]\right) d\theta \\
&\quad + \int_{\theta \in S_{yellow}} p(\theta)\mathbb{P}\left(\hat{g}^*\left(\theta'\right) \in [g\left(\theta\right) - \epsilon, g\left(\theta\right) + \epsilon]\right) d\theta \\
&< \frac{2\epsilon\left(t_0 + 1\right)}{\left|\left(1 - \alpha\right) t_0 - \alpha\right|s} + \frac{2s \cdot t_0}{\left(t_0 + 1\right)\left(\overline{m} - \underline{m}\right)} + \frac{s^2}{2\left(\overline{m} - \underline{m}\right)^2}.
\end{aligned}
$$

The second term $\frac{2s \cdot t_0}{\left(t_0+1\right)\left(\overline{m}-\underline{m}\right)}$ bounds the probability of the yellow region except for the right-bottom triangle, and the last term $\frac{s^2}{2\left(\overline{m}-\underline{m}\right)^2}$ is the probability of the right-bottom triangle.

For the distortion, it is straightforward to get that $\Delta = \frac{\left(t_0^2+1\right)s}{4\left(t_0+1\right)^2}$ from Eq. (23), and $\Delta_{\text{opt}} > \left(\lceil \frac{1}{\Pi_{\epsilon,\omega_\Theta}} \rceil - 1\right) \cdot 2\gamma\epsilon \geq 2\gamma\epsilon$, where $\gamma$ is defined in Corollary 4. We can get that $\left(\Pi_{\epsilon,\omega_\Theta} - \frac{2s \cdot t_0}{\left(t_0+1\right)\left(\overline{m}-\underline{m}\right)} - \frac{s^2}{2\left(\overline{m}-\underline{m}\right)^2}\right) \cdot \Delta < 2\gamma\epsilon$ and

$$
\begin{aligned}
\Delta &= \Delta_{\text{opt}} + \Delta - \Delta_{\text{opt}} \\
&< \Delta_{\text{opt}} + \Delta - \left(\lceil \frac{1}{\Pi_{\epsilon,\omega_\Theta}} \rceil - 1\right) \cdot 2\gamma\epsilon \\
&\leq \Delta_{\text{opt}} + 2\gamma\epsilon + \Delta - \frac{2\gamma\epsilon}{\Pi_{\epsilon,\omega_\Theta}} \\
&< \Delta_{\text{opt}} + 2\gamma\epsilon + \frac{\frac{2s \cdot t_0}{\left(t_0+1\right)\left(\overline{m}-\underline{m}\right)} + \frac{s^2}{2\left(\overline{m}-\underline{m}\right)^2}}{\frac{2\epsilon\left(t_0+1\right)}{\left|\left(1-\alpha\right)t_0-\alpha\right|s} + \frac{2s \cdot t_0}{\left(t_0+1\right)\left(\overline{m}-\underline{m}\right)} + \frac{s^2}{2\left(\overline{m}-\underline{m}\right)^2}} \cdot \Delta \\
&= \left(1 + \frac{\left|\left(1-\alpha\right)t_0-\alpha\right|s}{2\epsilon\left(t_0+1\right)}\left(\frac{2s \cdot t_0}{\left(t_0+1\right)\left(\overline{m}-\underline{m}\right)} + \frac{s^2}{2\left(\overline{m}-\underline{m}\right)^2}\right)\right)\left(\Delta_{\text{opt}} + 2\gamma\epsilon\right) \\
&\leq \left(2 + \frac{\left|\left(1-\alpha\right)t_0-\alpha\right|s}{\epsilon\left(t_0+1\right)} \cdot \left(\frac{2s \cdot t_0}{\left(t_0+1\right)\left(\overline{m}-\underline{m}\right)} + \frac{s^2}{2\left(\overline{m}-\underline{m}\right)^2}\right)\right)\Delta_{\text{opt}}.
\end{aligned}
$$

When $\frac{s^2}{\overline{m}-\underline{m}} \to 0$ as $s, \left(\overline{m}-\underline{m}\right) \to \infty$, we can get that $\frac{s^3}{\left(\overline{m}-\underline{m}\right)^2} \to 0$. Therefore, in this case, $\limsup_{\frac{s^2}{\overline{m}-\underline{m}} \to 0} \Delta < 3\Delta_{\text{opt}}$.

$\square$

### F.3  Privacy-Distortion Performance of Mech. 3 with Relaxed Assumption

For Gaussian distribution, we relax Asm. 5 as follows.

**Assumption 6.** *The prior over Gaussian distribution parameters satisfies $Supp\left(\mu, \sigma\right) = \left\{(a,b) | a \in \left[\underline{\mu}, \overline{\mu}\right), b \in \left[\underline{\sigma}, \overline{\sigma}\right)\right\}$, $f_{\mu,\sigma}\left(a,b\right) = f_\mu\left(a\right) \cdot f_\sigma\left(b\right)$, and $f_\mu\left(a\right)$ (resp. $f_\sigma\left(b\right)$) is $\mathcal{L}_\mu$-Lipschitz (resp. $\mathcal{L}_\sigma$-Lipschitz) and has lower bound $\frac{k_\mu}{\overline{\mu}-\underline{\mu}}$ with $k_\mu \in (0,1]$ (resp. $\frac{k_\sigma}{\overline{\sigma}-\underline{\sigma}}$ with $k_\sigma \in (0,1]$).*

Based on Asm. 6, the Privacy-distortion performance of Mech. 3 is shown below.

**Proposition 7.** *Under [Asm. 6](), [Mech. 3]() has the following $\Delta$ and $\Pi_{\epsilon,\omega_\Theta}$ value/bound:*

$$\Delta = \frac{s}{2}\sqrt{\frac{2}{\pi}}e^{-\frac{1}{2}t_0^2} - \frac{t_0 s}{2}\left(1 - 2\Phi\left(t_0\right)\right),$$

$$\Pi_{\epsilon,\omega_\Theta} < \frac{\frac{2\epsilon}{|t_0+Q_\alpha|} \cdot \left[\underline{c} + \mathcal{L}_{\mu,\sigma}\left(\frac{s}{2} - t^* - \frac{\epsilon}{|t_0+Q_\alpha|}\right)\right]}{\underline{c}s + \frac{\mathcal{L}_{\mu,\sigma}}{2}\left(\frac{s}{2} - t^*\right)^2} +$$

$$M\left(\overline{\mu} - \underline{\mu}, \frac{k_\mu}{\overline{\mu} - \underline{\mu}}, \mathcal{L}_\mu, 1\right) \cdot M\left(\overline{\sigma} - \underline{\sigma}, \frac{k_\sigma}{\overline{\sigma} - \underline{\sigma}}, \mathcal{L}_\sigma, 1\right) \cdot \left(\overline{\sigma} - \underline{\sigma}\right)|t_0|s,$$

*where $\underline{c} = \frac{k_\mu k_\sigma}{(\overline{\mu}-\underline{\mu})\cdot(\overline{\sigma}-\underline{\sigma})}$, function $M$ satisfies*

$$M\left(x, c, \mathcal{L}, \mathcal{A}\right) = \begin{cases} \frac{\mathcal{A}}{x} + \frac{\mathcal{L}x}{2}, & \text{if } c \leq \frac{\mathcal{A}}{x} - \frac{\mathcal{L}x}{2} \\ c + \sqrt{2\mathcal{L}\left(\mathcal{A} - cx\right)}, & \text{if } c > \frac{\mathcal{A}}{x} - \frac{\mathcal{L}x}{2} \end{cases},$$

$$\mathcal{L}_{\mu,\sigma} = \mathcal{L}_\sigma \cdot M\left(\frac{\overline{\mu}-\underline{\mu}}{|t_0|}, \frac{k_\mu}{\overline{\mu}-\underline{\mu}}, |t_0|\mathcal{L}_\mu, \frac{1}{|t_0|}\right) + |t_0|\mathcal{L}_\mu \cdot M\left(\overline{\sigma} - \underline{\sigma}, \frac{k_\sigma}{\overline{\sigma}-\underline{\sigma}}, \mathcal{L}_\sigma, 1\right), \text{ and } t^* = \frac{s}{2} + \frac{\underline{c}}{\mathcal{L}_{\mu,\sigma}} - \frac{\epsilon}{|t_0+Q_\alpha|} - \sqrt{\left(\frac{\underline{c}}{\mathcal{L}_{\mu,\sigma}} - \frac{\epsilon}{|t_0+Q_\alpha|}\right)^2 + \frac{2\underline{c}s}{\mathcal{L}_{\mu,\sigma}}}.$$

*Proof.* It is straightforward to get the formula for $\Delta$ from [Eq. (22)](). Here we focus on the proof for $\Pi_{\epsilon,\omega_\Theta}$. Similar to [App. D.2.2](), based on [Lemma 1]() and [Lemma 2](), we can get that

$$\sup_{\theta \in S_{green}} \mathbb{P}\left(\hat{g}^*\left(\theta'\right) \in [g\left(\theta\right) - \epsilon, g\left(\theta\right) + \epsilon]\right)$$

$$= \sup_{i \in \mathbb{N}, \mu, t' \in \mathbb{R}} \frac{\int_{\max\left\{-\frac{s}{2}, t'\right\}}^{\min\left\{\frac{s}{2}, t' + \frac{2\epsilon}{|t_0+Q_\alpha|}\right\}} f_{\mu,\sigma}\left(\mu + t_0 \cdot t, \underline{\sigma} + (i + 0.5) \cdot s + t\right) \mathrm{d}t}{\int_{-\frac{s}{2}}^{\frac{s}{2}} f_{\mu,\sigma}\left(\mu + t_0 \cdot t, \underline{\sigma} + (i + 0.5) \cdot s + t\right) \mathrm{d}t}$$

$$\leq \frac{\frac{2\epsilon}{|t_0+Q_\alpha|} \cdot \left[\underline{c} + \mathcal{L}_{\mu,\sigma}\left(\frac{s}{2} - t^* - \frac{\epsilon}{|t_0+Q_\alpha|}\right)\right]}{\underline{c}s + \frac{\mathcal{L}_{\mu,\sigma}}{2}\left(\frac{s}{2} - t^*\right)^2},$$

where $t^* = \frac{s}{2} + \frac{\underline{c}}{\mathcal{L}_{\mu,\sigma}} - \frac{\epsilon}{|t_0+Q_\alpha|} - \sqrt{\left(\frac{\underline{c}}{\mathcal{L}_{\mu,\sigma}} - \frac{\epsilon}{|t_0+Q_\alpha|}\right)^2 + \frac{2\underline{c}s}{\mathcal{L}_{\mu,\sigma}}}$, $\mathcal{L}_{\mu,\sigma} = \mathcal{L}_\sigma \cdot M\left(\frac{\overline{\mu}-\underline{\mu}}{|t_0|}, \frac{k_\mu}{\overline{\mu}-\underline{\mu}}, |t_0|\mathcal{L}_\mu, \frac{1}{|t_0|}\right) + |t_0|\mathcal{L}_\mu \cdot M\left(\overline{\sigma} - \underline{\sigma}, \frac{k_\sigma}{\overline{\sigma}-\underline{\sigma}}, \mathcal{L}_\sigma, 1\right)$, and $\underline{c} = \frac{k_\mu k_\sigma}{(\overline{\mu}-\underline{\mu})\cdot(\overline{\sigma}-\underline{\sigma})}$.

As for $\int_{\theta \in S_{yellow}} p(\theta)d\theta$, we have

$$\int_{\theta \in S_{yellow}} p(\theta)d\theta$$

$$\leq M\left(\overline{\mu} - \underline{\mu}, \frac{k_\mu}{\overline{\mu} - \underline{\mu}}, \mathcal{L}_\mu, 1\right) \cdot M\left(\overline{\sigma} - \underline{\sigma}, \frac{k_\sigma}{\overline{\sigma} - \underline{\sigma}}, \mathcal{L}_\sigma, 1\right) \cdot \int_{\theta \in S_{yellow}} d\theta$$

$$= M\left(\overline{\mu} - \underline{\mu}, \frac{k_\mu}{\overline{\mu} - \underline{\mu}}, \mathcal{L}_\mu, 1\right) \cdot M\left(\overline{\sigma} - \underline{\sigma}, \frac{k_\sigma}{\overline{\sigma} - \underline{\sigma}}, \mathcal{L}_\sigma, 1\right) \cdot \left(\overline{\sigma} - \underline{\sigma}\right)|t_0|s.$$

Above all, we can get that

$$\Pi_{\epsilon,\omega_\Theta} < \sup_{\theta \in S_{green}} \mathbb{P}\left(\hat{g}^*\left(\theta'\right) \in \left[g\left(\theta\right) - \epsilon, g\left(\theta\right) + \epsilon\right]\right) + \int_{\theta \in S_{yellow}} p(\theta)d\theta.$$

$$\leq \frac{\frac{2\epsilon}{|t_0 + Q_\alpha|} \cdot \left[\underline{c} + \mathcal{L}_{\mu,\sigma}\left(\frac{s}{2} - t^* - \frac{\epsilon}{|t_0 + Q_\alpha|}\right)\right]}{\underline{c}s + \frac{\mathcal{L}_{\mu,\sigma}}{2}\left(\frac{s}{2} - t^*\right)^2} +$$

$$M\left(\overline{\mu} - \underline{\mu}, \frac{k_\mu}{\overline{\mu} - \underline{\mu}}, \mathcal{L}_\mu, 1\right) \cdot M\left(\overline{\sigma} - \underline{\sigma}, \frac{k_\sigma}{\overline{\sigma} - \underline{\sigma}}, \mathcal{L}_\sigma, 1\right) \cdot \left(\overline{\sigma} - \underline{\sigma}\right)|t_0|s,$$

where $M(\cdot, \cdot, \cdot, \cdot), \underline{c}, \mathcal{L}_{\mu,\sigma}, t^*$ are defined as above. $\qquad\square$

# G   Case Study with Secret = Standard Deviation

In this section, we discuss how to protect standard deviation for several continuous and discrete distributions.

## G.1   Continuous Distributions

We consider the same distributions discussed in §6.2 and App. F: Gaussian, uniform, and (shifted) exponential distributions.

**Corollary 5** (Privacy lower bound, secret = standard deviation of a continuous distribution)**.** *Consider the secret function $g(\theta) =$ standard deviation of $f_{X_\theta}$. For any $T \in (0, 1)$, when $\Pi_{\epsilon,\omega_\Theta} \leq T$, we have $\Delta > \left(\lceil \frac{1}{T} \rceil - 1\right) \cdot 2\gamma\epsilon$, where the value of $\gamma$ depends on the type of the distributions:*

- *Gaussian:*

$$\gamma = \min_t \sqrt{\frac{1}{2\pi}} e^{-\frac{1}{2}t^2} - t\left(\frac{1}{2} - \Phi\left(t\right)\right),$$

*where $\Phi$ denotes the CDF of the standard Gaussian distribution.*
- *Uniform: $\gamma = \frac{\sqrt{3}}{4}$.*
- *Exponential: $\gamma = \frac{1}{2}$.*
- *Shifted exponential: $\gamma = \frac{\ln 2}{2}$.*

The proof is in App. G.3. The bounds for Gaussian can be computed numerically, while the bounds for all other distributions are in closed form.

Next, we present the data release mechanism for these distributions and the secret under the same assumption as Asm. 2.

**Mechanism 4** (For secret = standard deviation of a continuous distribution)**.** *We design mechanisms for each of the distributions.*

- *Gaussian:*

$$\mathcal{S}_{\mu,i} = \left\{\left(\mu + t_0 \cdot t, \underline{\sigma} + (i + 0.5) \cdot s + t\right) | t \in \left[-\frac{s}{2}, \frac{s}{2}\right)\right\} \quad,$$
$$\theta^*_{\mu,i} = \left(\mu, \underline{\sigma} + (i + 0.5) \cdot s\right) \quad,$$
$$\mathcal{I} = \left\{(\mu, i) | i \in \mathbb{N}, \mu \in \mathbb{R}\right\},$$

*where $s$ is a hyper-parameter of the mechanism that divides $(\overline{\sigma} - \underline{\sigma})$ and*

$$t_0 = \arg\min_t \sqrt{\frac{1}{2\pi}} e^{-\frac{1}{2}t^2} - t\left(\frac{1}{2} - \Phi\left(t\right)\right).$$

.

- *Uniform:*

$$\mathcal{S}_{m,i} = \left\{ (m-t, m+(i+0.5) \cdot s + t) \, | \, t \in \left( -\frac{s}{4}, \frac{s}{4} \right] \right\} \quad,$$
$$\theta^*_{m,i} = (m, m+(i+0.5) \cdot s) \quad,$$
$$\mathcal{I} = \{ (m,i) \, | \, i \in \mathbb{Z}_{>0}, m \in \mathbb{R} \},$$

where $s > 0$ is a hyper-parameter of the mechanism that divides $(\overline{m} - \underline{m})$.

- *Exponential:*

$$\mathcal{S}_i = [\underline{\lambda} + i \cdot s, \underline{\lambda} + (i+1) \cdot s) \quad,$$
$$\theta^*_i = \underline{\lambda} + (i+0.5) \cdot s \quad,$$
$$\mathcal{I} = \mathbb{N},$$

where $s > 0$ is a hyper-parameter of the mechanism that divides $(\overline{\lambda} - \underline{\lambda})$.

- *Shifted exponential:*

$$\mathcal{S}_{i,h} = \left\{ (\underline{\lambda} + (i+0.5) s + t, h - \ln 2 \cdot t) \, | \, t \in \left[ -\frac{s}{2}, \frac{s}{2} \right) \right\} \quad,$$
$$\theta^*_{i,h} = (\underline{\lambda} + (i+0.5) s, h) \quad,$$
$$\mathcal{I} = \{ (i,h) | i \in \mathbb{N}, h \in \mathbb{R} \},$$

where $s > 0$ is a hyper-parameter of the mechanism that divides $(\overline{\lambda} - \underline{\lambda})$.

These data release mechanisms achieve the following $\Delta$ and $\Pi_{\epsilon, \omega_\Theta}$.

**Proposition 8.** *Under Asm. 2, Mech. 4 has the following $\Delta$ and $\Pi_{\epsilon, \omega_\Theta}$ value/bound.*

- *Gaussian:*

$$\Pi_{\epsilon, \omega_\Theta} < \frac{2\epsilon}{s} + \frac{|t_0| s}{\overline{\mu} - \underline{\mu}},$$

$$\Delta = \frac{s}{2} \sqrt{\frac{2}{\pi}} e^{-\frac{1}{2} t_0^2} - \frac{t_0 s}{2} (1 - 2\Phi(t_0)) < \left( 2 + \frac{|t_0| s^2}{(\overline{\mu} - \underline{\mu}) \epsilon} \right) \Delta_{opt},$$

where $t_0$ is defined in Mech. 4. Under the "high-precision" regime where $\frac{s^2}{\overline{\mu} - \underline{\mu}} \to 0$ as $s, (\overline{\mu} - \underline{\mu}) \to \infty$, $\Delta$ satisfies

$$\lim_{\frac{s^2}{\overline{\mu} - \underline{\mu}} \to 0} \sup \Delta < 3\Delta_{opt}.$$

- *Uniform:*

$$\Pi_{\epsilon, \omega_\Theta} < \frac{4\sqrt{3}\epsilon}{s} + \frac{s}{(\overline{m} - \underline{m})} + \frac{s^2}{2(\overline{m} - \underline{m})^2},$$

$$\Delta = \frac{s}{8} < \left( 2 + \frac{s}{2\sqrt{3}\epsilon} \cdot \left( \frac{s}{\overline{m} - \underline{m}} + \frac{s^2}{2(\overline{m} - \underline{m})^2} \right) \right) \Delta_{opt}.$$

*Under the "high-precision" regime where $\frac{s^2}{\overline{m} - \underline{m}} \to 0$ as $s, (\overline{m} - \underline{m}) \to \infty$, $\Delta$ satisfies*

$$\lim_{\frac{s^2}{\overline{m} - \underline{m}} \to 0} \sup \Delta < 3\Delta_{opt}.$$

- *Exponential:*

$$\Pi_{\epsilon, \omega_\Theta} = \frac{2\epsilon}{s},$$

$$\Delta = \frac{1}{2} s < 2\Delta_{opt}.$$

- *Shifted exponential:*

$$\Pi_{\epsilon,\omega_\Theta} < \frac{2\epsilon}{s} + \frac{s\ln 2}{\overline{h} - \underline{h}},$$

$$\Delta = \frac{s\ln 2}{2} < \left(2 + \frac{s^2\ln 2}{\epsilon\left(\overline{h} - \underline{h}\right)}\right)\Delta_{opt}.$$

*Under the "high-precision" regime where $\frac{s^2}{\overline{h}-\underline{h}} \to 0$ as $s, (\overline{h} - \underline{h}) \to \infty$, $\Delta$ satisfies*

$$\lim_{\substack{\sup \\ \frac{s^2}{\overline{h}-\underline{h}} \to 0}} \Delta < 3\Delta_{opt}.$$

The proof is in App. G.4. For Gaussian, exponential and shifted exponential distributions, we relax Asm. 2 and analyze the privacy-distortion performance of Mech. 4 in App. G.5. From these propositions, we have similar takeaways as the alpha-quantile case ( §6.2): (1) data holder can use $s$ to control the trade-off between distortion and privacy, and (2) the mechanism is order-optimal under the "high-precision" regime.

### G.2 Discrete Distributions

Here, we consider the same discrete distributions studied in App. E: Geometric distributions, binomial distributions, and Poisson distributions. We first analyze the lower bound.

**Corollary 6** (Privacy lower bound, secret = standard deviation of a discrete distribution)**.** *Consider the secret function $g(\theta) =$ standard deviation of $f_{X_\theta}$. For any $T \in (0, 1)$, when $\Pi_{\epsilon,\omega_\Theta} \leq T$, we have $\Delta > \left(\lceil\frac{1}{T}\rceil - 1\right) \cdot 2\gamma\epsilon$, where the value of $\gamma$ depends on the type of the distributions:*

- *Geometric:*

$$\gamma = \inf_{\underline{\theta} < \theta_1 < \theta_2 \leq \overline{\theta}} \frac{(1-\theta_2)^{h(\theta_1,\theta_2)} - (1-\theta_1)^{h(\theta_1,\theta_2)}}{2\left(\frac{\sqrt{1-\theta_2}}{\theta_2} - \frac{\sqrt{1-\theta_1}}{\theta_1}\right)},$$

*where $h(\theta_1, \theta_2) = \lfloor\frac{\log(\theta_2)-\log(\theta_1)}{\log(1-\theta_1)-\log(1-\theta_2)}\rfloor + 1$.*
- *Binomial:*

$$\gamma = \inf_{\underline{\theta} < \theta_1 < \theta_2 \leq \overline{\theta}}$$
$$\frac{I_{1-\theta_2}(n-h(\theta_1,\theta_2),1+h(\theta_1,\theta_2)) - I_{1-\theta_1}(n-h(\theta_1,\theta_2),1+h(\theta_1,\theta_2))}{2\left|\sqrt{n\theta_2(1-\theta_2)} - \sqrt{n\theta_1(1-\theta_1)}\right|},$$

*where $h(\theta_1, \theta_2) = \lfloor k'\rfloor$, $k' = n\ln\left(\frac{1-\theta_2}{1-\theta_1}\right)\bigg/\ln\left(\frac{\theta_1(1-\theta_2)}{\theta_2(1-\theta_1)}\right)$, and $I$ represents the regularized incomplete beta function.*
- *Poisson:*

$$\gamma = \inf_{\underline{\theta} < \theta_1 < \theta_2 \leq \overline{\theta}} \frac{Q(h(\theta_1,\theta_2),\theta_2) - Q(h(\theta_1,\theta_2),\theta_1)}{2(\sqrt{\theta_1} - \sqrt{\theta_2})},$$

*where $h(\theta_1, \theta_2) = \lfloor\frac{\theta_1-\theta_2}{\ln(\theta_1)-\ln(\theta_2)}\rfloor + 1$ and $Q$ is the regularized gamma function.*

The proof is in App. G.6. The above lower bounds can be computed numerically.

Since these distributions only have one parameter, we can use Alg. 1 and Alg. 3 to derive a data release mechanism. The performance of greedy-based and dynamic-programming-based data release mechanisms for each distribution is shown in Fig. 14.

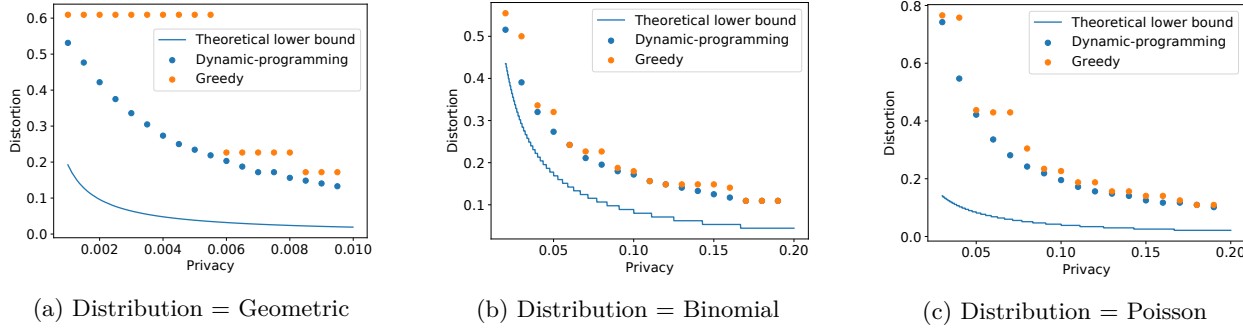

(a) Distribution = Geometric     (b) Distribution = Binomial     (c) Distribution = Poisson

Figure 14: Privacy-distortion performance of Alg. 1 and Alg. 3 for binomial and Poisson distribution when secret = standard deviation.

### G.3 Proof of Corollary 5

#### G.3.1 Gaussian Distribution

*Proof.* Let $X_{\mu_1,\sigma_2}, X_{\mu_2,\sigma_2}$ be two Gaussian random variables with means $\mu_1, \mu_2$ and sigmas $\sigma_1, \sigma_2$ respectively, where $\sigma_1 \neq \sigma_2$. Let $\Phi$ denotes the CDF of the standard Gaussian distribution. We can get that

$$D\left(X_{\mu_1,\sigma_1}, X_{\mu_2,\sigma_2}\right) = \left(\mu_1 - \mu_2\right)\left(\Phi\left(\frac{\mu_1 - \mu_2}{\sigma_2 - \sigma_1}\right) - \frac{1}{2}\right)$$
$$+ \sqrt{\frac{1}{2\pi}}\left(\sigma_2 - \sigma_1\right)e^{-\frac{1}{2}\left(\frac{\mu_1 - \mu_2}{\sigma_2 - \sigma_1}\right)^2},$$
$$R\left(X_{\mu_1,\sigma_1}, X_{\mu_2,\sigma_2}\right) = |\sigma_1 - \sigma_2|.$$

Let $\frac{\mu_1 - \mu_2}{\sigma_1 - \sigma_2} \triangleq t$, we can get that

$$\frac{D\left(X_{\mu_1,\sigma_1}, X_{\mu_2,\sigma_2}\right)}{R\left(X_{\mu_1,\sigma_1}, X_{\mu_2,\sigma_2}\right)} = \sqrt{\frac{1}{2\pi}}e^{-\frac{1}{2}t^2} - t\left(\frac{1}{2} - \Phi\left(t\right)\right) \triangleq h\left(t\right).$$

Therefore we can get that

$$\gamma = \min_t h\left(t\right).$$

$\square$

#### G.3.2 Uniform Distribution

*Proof.* Let $X_{m_1,n_1}, X_{m_2,n_2}$ be two uniform random variables. Let $F_{X_{m_1,n_1}}, F_{X_{m_2,n_2}}$ be their CDFs, and let $m_2 \geq m_1$ without loss of generality. We can get that

$$D\left(X_{m_1,n_1}, X_{m_2,n_2}\right) = \frac{1}{2}d_{\text{Wasserstein-1}}\left(\omega_{X_{m_1,n_1}} \| \omega_{X_{m_2,n_2}}\right)$$
$$= \frac{1}{2}\int_{-\infty}^{+\infty}|F_{X_{m_1,n_1}}\left(x\right) - F_{X_{m_2,n_2}}\left(x\right)|\mathrm{d}x$$
$$= \begin{cases} \frac{m_2 - m_1 + n_2 - n_1}{4} & n_2 \geq n_1 \\ \frac{(m_2 - m_1)^2 + (n_1 - n_2)^2}{4(m_2 - m_1 + (n_1 - n_2))} & n_2 < n_1 \end{cases},$$
$$R\left(X_{m_1,n_1}, X_{m_2,n_2}\right) = \left|\frac{1}{\sqrt{12}}\left(n_1 - m_1\right) - \frac{1}{\sqrt{12}}\left(n_2 - m_2\right)\right|$$
$$= \frac{1}{\sqrt{12}}|m_2 - m_1 - (n_2 - n_1)|.$$

Therefore, we can get that when $n_2 \geq n_1$, we have

$$\frac{D\left(X_{m_1,n_1}, X_{m_2,n_2}\right)}{R\left(X_{m_1,n_1}, X_{m_2,n_2}\right)} = \frac{\sqrt{3}}{2} \frac{m_2 - m_1 + n_2 - n_1}{|m_2 - m_1 - (n_2 - n_1)|}$$

$$\geq \frac{\sqrt{3}}{2}.$$

When $n_2 < n_1$, we have

$$\frac{D\left(X_{m_1,n_1}, X_{m_2,n_2}\right)}{R\left(X_{m_1,n_1}, X_{m_2,n_2}\right)} = \frac{\sqrt{3}}{2} \frac{(m_2 - m_1)^2 + (n_1 - n_2)^2}{(m_2 - m_1 + (n_1 - n_2))^2}$$

$$= \frac{\sqrt{3}}{2} \frac{(m_2 - m_1)^2 + (n_1 - n_2)^2}{(m_2 - m_1)^2 + (n_1 - n_2)^2 + 2(m_2 - m_1)(n_1 - n_2)}$$

$$\geq \frac{\sqrt{3}}{2} \cdot \frac{(m_2 - m_1)^2 + (n_1 - n_2)^2}{2\left[(m_2 - m_1)^2 + (n_1 - n_2)^2\right]}$$

$$= \frac{\sqrt{3}}{4}.$$

Therefore we can get that

$$\gamma = \frac{\sqrt{3}}{4}.$$

$\square$

### G.3.3 Exponential Distribution

*Proof.* Let $X_{\lambda_1}, X_{\lambda_2}$ be two exponential random variables. We have

$$\frac{D\left(X_{\lambda_1}, X_{\lambda_2}\right)}{R\left(X_{\lambda_1}, X_{\lambda_2}\right)} = \frac{\frac{1}{\lambda_1} - \frac{1}{\lambda_2}}{2\left(\frac{1}{\lambda_1} - \frac{1}{\lambda_2}\right)} = \frac{1}{2}.$$

Therefore we can get that

$$\gamma = \frac{1}{2}.$$

$\square$

### G.3.4 Shifted Exponential Distribution

*Proof.* Let $X_{\lambda_1,h_1}, X_{\lambda_2,h_2}$ be random variables from shifted exponential distributions. Let $\lambda_2 \leq \lambda_1$ without loss of generality. Let $a = \frac{\lambda_1}{\lambda_2}$ and $b = (h_1/\lambda_1 - h_2/\lambda_2)\lambda_2$. We can get that $f_{X_{\lambda_1,h_1}}(x) = af_{X_{\lambda_2,h_2}}(a(x+b))$, and

$$D\left(X_{\lambda_1,h_1}, X_{\lambda_2,h_2}\right) = \frac{1}{2}d_{\text{Wasserstein-1}}\left(\omega_{X_{\lambda_1,h_1}} \| \omega_{X_{\lambda_2,h_2}}\right)$$

$$= \frac{1}{2}\int_{h_1}^{+\infty} \left|x - \left(\frac{x}{a} - b\right)\right| f_{X_{\lambda_1,h_1}}(x)\,\mathrm{d}x$$

$$= \frac{\lambda_2}{2\lambda_1}\int_{h_1}^{+\infty} |(1/\lambda_2 - 1/\lambda_1)x + h_1/\lambda_1 - h_2/\lambda_2| e^{-\frac{1}{\lambda_1}(x-h_1)}\mathrm{d}x$$

$$= \begin{cases} \frac{1}{2}(h_2 - h_1 + \lambda_2 - \lambda_1) - e^{\frac{h_2-h_1}{\lambda_2-\lambda_1}}(\lambda_2 - \lambda_1) & (h_1 < h_2) \\ \frac{1}{2}(h_1 - h_2 + \lambda_1 - \lambda_2) & (h_1 \geq h_2) \end{cases},$$

$$R\left(X_{\lambda_1,h_1}, X_{\lambda_2,h_2}\right) = \lambda_1 - \lambda_2. \tag{24}$$

When $\lambda_1 = \lambda_2$ and $h_1 \neq h_2$, we have $\frac{D\left(X_{\lambda_1,h_1}, X_{\lambda_2,h_2}\right)}{R\left(X_{\lambda_1,h_1}, X_{\lambda_2,h_2}\right)} = \infty$.

When $\lambda_1 \neq \lambda_2$ and $h_1 < h_2$, let $t = \frac{h_2 - h_1}{\lambda_1 - \lambda_2} \in (0, +\infty)$. We have

$$
\frac{D\left(X_{\lambda_1,h_1}, X_{\lambda_2,h_2}\right)}{R\left(X_{\lambda_1,h_1}, X_{\lambda_2,h_2}\right)} = \frac{h_2 - h_1 + \lambda_2 - \lambda_1 - 2e^{\frac{h_2 - h_1}{\lambda_2 - \lambda_1}} (\lambda_2 - \lambda_1)}{2(\lambda_1 - \lambda_2)}
$$
$$
= \frac{t + 2e^{-t} - 1}{2}
$$
$$
\geq \frac{\ln 2}{2}.
$$

"=" achieves when $t = t_0 = \ln 2$.

When $\lambda_1 \neq \lambda_2$ and $h_1 \geq h_2$, we have

$$
\frac{D\left(X_{\lambda_1,h_1}, X_{\lambda_2,h_2}\right)}{R\left(X_{\lambda_1,h_1}, X_{\lambda_2,h_2}\right)} = \frac{h_1 - h_2 + \lambda_1 - \lambda_2}{2(\lambda_1 - \lambda_2)} \geq \frac{\lambda_1 - \lambda_2}{2(\lambda_1 - \lambda_2)} = \frac{1}{2}.
$$

Therefore we can get that

$$
\gamma = \frac{\ln 2}{2}.
$$

$\square$

## G.4 Proof of Prop. 8

The proof outline is almost the same as the ones in App. C.4 and App. F.2. We omit the details and point to the proof sections where we can adapt from.

### G.4.1 Gaussian Distribution

The proof is the same as App. F.2.1, except that we use the $D(\cdot, \cdot)$ and $R(\cdot, \cdot)$ from App. G.3.1.

### G.4.2 Uniform Distribution

The proof is the same as App. F.2.2, except that we use the $D(\cdot, \cdot)$ and $R(\cdot, \cdot)$ from App. G.3.2.

### G.4.3 Exponential Distribution

The proof is the same as App. C.4.1, except that we use the $D(\cdot, \cdot)$ and $R(\cdot, \cdot)$ from App. G.3.3.

### G.4.4 Shifted Exponential Distribution

The proof is the same as App. C.4.2, except that we use the $D(\cdot, \cdot)$ and $R(\cdot, \cdot)$ from App. G.3.4.

## G.5 Privacy-Distortion Performance of Mech. 4 with Relaxed Assumption

Based on Asm. 6 and Asm. 4, the Privacy-distortion performance of Mech. 4 is shown below.

**Proposition 9.** *Under Asm. 6 and Asm. 4, Mech. 4 has the following $\Delta$ and $\Pi_{\epsilon, \omega_\Theta}$ value/bound.*

- *Gaussian:*

$$
\Delta = \frac{s}{2}\sqrt{\frac{2}{\pi}} e^{-\frac{1}{2}t_0^2} - \frac{t_0 s}{2}(1 - 2\Phi(t_0)),
$$
$$
\Pi_{\epsilon, \omega_\Theta} < \frac{2\epsilon \cdot \left[\underline{c} + \mathcal{L}_{\mu,\sigma}\left(\frac{s}{2} - t^* - \epsilon\right)\right]}{\underline{c}s + \frac{\mathcal{L}_{\mu,\sigma}}{2}\left(\frac{s}{2} - t^*\right)^2} +
$$
$$
M\left(\overline{\mu} - \underline{\mu}, \frac{k_\mu}{\overline{\mu} - \underline{\mu}}, \mathcal{L}_\mu, 1\right) \cdot M\left(\overline{\sigma} - \underline{\sigma}, \frac{k_\sigma}{\overline{\sigma} - \underline{\sigma}}, \mathcal{L}_\sigma, 1\right) \cdot (\overline{\sigma} - \underline{\sigma})\, |t_0| s,
$$

where $t_0$ is defined in Mech. 4, $\underline{c} = \frac{k_\mu k_\sigma}{(\overline{\mu}-\underline{\mu})\cdot(\overline{\sigma}-\underline{\sigma})}$, function $M$ satisfies

$$M(x, c, \mathcal{L}, \mathcal{A}) = \begin{cases} \frac{\mathcal{A}}{x} + \frac{\mathcal{L}x}{2}, & \text{if } c \le \frac{\mathcal{A}}{x} - \frac{\mathcal{L}x}{2} \\ c + \sqrt{2\mathcal{L}(\mathcal{A} - cx)}, & \text{if } c > \frac{\mathcal{A}}{x} - \frac{\mathcal{L}x}{2} \end{cases},$$

$\mathcal{L}_{\mu,\sigma} = \mathcal{L}_\sigma M\left(\frac{\overline{\mu}-\underline{\mu}}{|t_0|}, \frac{k_\mu}{\overline{\mu}-\underline{\mu}}, |t_0|\mathcal{L}_\mu, \frac{1}{|t_0|}\right) + |t_0|\mathcal{L}_\mu M\left(\overline{\sigma} - \underline{\sigma}, \frac{k_\sigma}{\overline{\sigma}-\underline{\sigma}}, \mathcal{L}_\sigma, 1\right)$, and $t^* = \frac{s}{2} + \frac{c}{\mathcal{L}_{\mu,\sigma}} - \epsilon - \sqrt{\left(\frac{c}{\mathcal{L}_{\mu,\sigma}} - \epsilon\right)^2 + \frac{2cs}{\mathcal{L}_{\mu,\sigma}}}$.

- *Exponential:*

$$\Delta = \frac{1}{2}s,$$

$$\Pi_{\epsilon,\omega_\Theta} \le \frac{2\epsilon \cdot [\underline{c} + \mathcal{L}(s - x^* + \epsilon)]}{\underline{c}s + \frac{\mathcal{L}}{2}(s - x^*)^2},$$

where $x^* = s + \frac{c}{\mathcal{L}} + \epsilon - \sqrt{\left(\frac{c}{\mathcal{L}} + \epsilon\right)^2 + \frac{2cs}{\mathcal{L}}}$.

- *Shifted exponential:*

$$\Delta = \frac{s\ln 2}{2},$$

$$\Pi_{\epsilon,\omega_\Theta} < \frac{2\epsilon \cdot \left[\underline{c} + \mathcal{L}_{\lambda,h}\left(\frac{s}{2} - t^* - \epsilon\right)\right]}{\underline{c}s + \frac{\mathcal{L}_{\lambda,h}}{2}\left(\frac{s}{2} - t^*\right)^2} +$$

$$\ln 2 \cdot M\left(\overline{h} - \underline{h}, \frac{k_h}{\overline{h}-\underline{h}}, \mathcal{L}_h, 1\right) \cdot M\left(\overline{\lambda} - \underline{\lambda}, \frac{k_\lambda}{\overline{\lambda}-\underline{\lambda}}, \mathcal{L}_\lambda, 1\right) \cdot (\overline{\lambda} - \underline{\lambda})s,$$

where $\underline{c} = \frac{k_h k_\lambda}{(\overline{h}-\underline{h})\cdot(\overline{\lambda}-\underline{\lambda})}$, function $M$ satisfies

$$M(x, c, \mathcal{L}, \mathcal{A}) = \begin{cases} \frac{\mathcal{A}}{x} + \frac{\mathcal{L}x}{2}, & \text{if } c \le \frac{\mathcal{A}}{x} - \frac{\mathcal{L}x}{2} \\ c + \sqrt{2\mathcal{L}(\mathcal{A} - cx)}, & \text{if } c > \frac{\mathcal{A}}{x} - \frac{\mathcal{L}x}{2} \end{cases},$$

$\mathcal{L}_{\lambda,h} = \mathcal{L}_\lambda M\left(\frac{\overline{h}-\underline{h}}{\ln 2}, \frac{k_h}{\overline{h}-\underline{h}}, \ln 2 \cdot \mathcal{L}_h, \frac{1}{\ln 2}\right) + \ln 2 \cdot \mathcal{L}_h M\left(\overline{\lambda} - \underline{\lambda}, \frac{k_\lambda}{\overline{\lambda}-\underline{\lambda}}, \mathcal{L}_\lambda, 1\right)$, and $t^* = \frac{s}{2} + \frac{c}{\mathcal{L}_{\lambda,h}} - \epsilon - \sqrt{\left(\frac{c}{\mathcal{L}_{\lambda,h}} - \epsilon\right)^2 + \frac{2cs}{\mathcal{L}_{\lambda,h}}}$.

The proofs are the same as App. F.3, App. D.2.1 and App. D.2.2, except that we use the $D(\cdot, \cdot)$, and $R(\cdot, \cdot)$ from App. G.3.1, App. G.3.3, and App. G.3.4.

## G.6  Proof of Corollary 6

### G.6.1  Geometric Distribution

*Proof.* Let $X_{\theta_1}$ and $X_{\theta_2}$ be two Geometric random variables with parameters $\theta_1$ and $\theta_2$ respectively. We assume that $\theta_1 > \theta_2$ without loss of generality. Let $k'$ satisfy $(1 - \theta_1)^{k'}\theta_1 = (1 - \theta_2)^{k'}\theta_2$ and $k_0 = \lfloor k' \rfloor + 1$. Then we can get that

$$D(X_{\theta_1}, X_{\theta_2}) = \frac{1}{2}d_{\mathrm{TV}}\left(\omega_{X_{\theta_1}} \| \omega_{X_{\theta_2}}\right)$$

$$= \frac{1}{2}(1 - \theta_2)^{k_0} - \frac{1}{2}(1 - \theta_1)^{k_0},$$

$$R(X_{\theta_1}, X_{\theta_2}) = \frac{\sqrt{1 - \theta_2}}{\theta_2} - \frac{\sqrt{1 - \theta_1}}{\theta_1}.$$

Therefore, we can get that

$$\gamma = \inf_{\underline{\theta} < \theta_1 < \theta_2 \leq \overline{\theta}} \frac{(1 - \theta_2)^{k_0} - (1 - \theta_1)^{k_0}}{2 \left( \frac{\sqrt{1 - \theta_2}}{\theta_2} - \frac{\sqrt{1 - \theta_1}}{\theta_1} \right)}.$$

□

### G.6.2 Binomial Distribution

*Proof.* Let $X_{\theta_1}$ and $X_{\theta_2}$ be two binomial random variables with parameters $\theta_1$ and $\theta_2$ respectively with fixed number of trials $n$. We assume that $\theta_1 > \theta_2$ without loss of generality. Let $k'$ satisfy $\binom{n}{k'} \theta_1^{k'} (1 - \theta_1)^{n-k'} = \binom{n}{k'} \theta_2^{k'} (1 - \theta_1)^{n-k'}$ and $k_0 = \lfloor k' \rfloor$. We can get that

$$\begin{aligned}
D\left(X_{\theta_1}, X_{\theta_2}\right) &= \frac{1}{2} d_{\mathrm{TV}}\left(\omega_{X_{\theta_1}} \| \omega_{X_{\theta_2}}\right) \\
&= \frac{1}{2} I_{1-\theta_2}\left(n - k_0, 1 + k_0\right) - \frac{1}{2} I_{1-\theta_1}\left(n - k_0, 1 + k_0\right), \\
R\left(X_{\theta_1}, X_{\theta_2}\right) &= \left| \sqrt{n\theta_2\left(1 - \theta_2\right)} - \sqrt{n\theta_1\left(1 - \theta_1\right)} \right|,
\end{aligned}$$

where $I$ represents the regularized incomplete beta function.

Therefore, we can get that

$$\gamma = \inf_{\underline{\theta} < \theta_1 < \theta_2 \leq \overline{\theta}} \frac{I_{1-\theta_2}\left(n - k_0, 1 + k_0\right) - I_{1-\theta_1}\left(n - k_0, 1 + k_0\right)}{\left| \sqrt{n\theta_2\left(1 - \theta_2\right)} - \sqrt{n\theta_1\left(1 - \theta_1\right)} \right|}.$$

□

### G.6.3 Poisson Distribution

*Proof.* Let $X_{\theta_1}$ and $X_{\theta_2}$ be two Poisson random variables with parameters $\theta_1$ and $\theta_2$ respectively. We assume that $\theta_1 > \theta_2$ without loss of generality. Let $k'$ satisfy $\theta_1^{k'} e^{-\theta_1} = \theta_2^{k'} e^{-\theta_2}$ and $k_0 = \lfloor k' \rfloor + 1$. Then we can get that

$$\begin{aligned}
D\left(X_{\theta_1}, X_{\theta_2}\right) &= \frac{1}{2} d_{\mathrm{TV}}\left(\omega_{X_{\theta_1}} \| \omega_{X_{\theta_2}}\right) \\
&= \frac{1}{2} Q\left(k_0, \theta_2\right) - \frac{1}{2} Q\left(k_0, \theta_1\right), \\
R\left(X_{\theta_1}, X_{\theta_2}\right) &= \sqrt{\theta_1} - \sqrt{\theta_2},
\end{aligned}$$

where $Q$ is the regularized gamma function.

Therefore, we can get that

$$\gamma = \inf_{\underline{\theta} < \theta_1 < \theta_2 \leq \overline{\theta}} \frac{Q\left(k_0, \theta_2\right) - Q\left(k_0, \theta_1\right)}{2\left(\sqrt{\theta_1} - \sqrt{\theta_2}\right)}.$$

□

## H   Case Study with Secret = Fraction

As indicated in S1 in §2.1, the fraction of discrete distributions can reveal sensitive information. In this section, we first present the results for ordinal distributions, where there is a specific formula for the fractions at each bin (i.e., binomial, Poisson, geometric that we discussed in Apps. E and G.2). We then present the results for categorical distributions, where there is no constraint on the fractions of the bins so long as they are normalized.

### H.1 Ordinal Distribution

Here, we consider the same three discrete distributions studied in Apps. E and G.2: geometric distributions, binomial distributions, and Poisson distributions. We first analyze the lower bound. We assume that the secrete is the fraction of the $j$-th bin.

**Corollary 7** (Privacy lower bound, secret = fraction of an ordinal distribution)**.** *Consider the secret function* $g(\theta) = f_{X_\theta}(j)$. *For any* $T \in (0,1)$, *when* $\Pi_{\epsilon,\omega_\Theta} \le T$, *we have* $\Delta > \left(\lceil \frac{1}{T} \rceil - 1\right) \cdot 2\gamma\epsilon$, *where the value of* $\gamma$ *depends on the type of the distributions:*

- *Geometric:*

$$\gamma = \inf_{\underline{\theta} < \theta_1 < \theta_2 \le \overline{\theta}} \frac{(1-\theta_2)^{h(\theta_1,\theta_2)} - (1-\theta_1)^{h(\theta_1,\theta_2)}}{2\left|(1-\theta_2)^j \theta_2 - (1-\theta_1)^j \theta_1\right|} \quad ,$$

*where* $h(\theta_1, \theta_2) = \lfloor \frac{\log(\theta_2) - \log(\theta_1)}{\log(1-\theta_1) - \log(1-\theta_2)} \rfloor + 1.$

- *Binomial:*

$$\gamma = \inf_{\underline{\theta} < \theta_1 < \theta_2 \le \overline{\theta}}$$

$$\frac{I_{1-\theta_2}(n-h(\theta_1,\theta_2), 1+h(\theta_1,\theta_2)) - I_{1-\theta_1}(n-h(\theta_1,\theta_2), 1+h(\theta_1,\theta_2))}{2\left|\binom{n}{j}\theta_2^j(1-\theta_2)^{n-j} - \binom{n}{j}\theta_1^j(1-\theta_1)^{n-j}\right|},$$

*where* $h(\theta_1, \theta_2) = \lfloor k' \rfloor$, $k' = n \ln\left(\frac{1-\theta_2}{1-\theta_1}\right) / \ln\left(\frac{\theta_1(1-\theta_2)}{\theta_2(1-\theta_1)}\right)$, *and* $I$ *represents the regularized incomplete beta function.*

- *Poisson:*

$$\gamma = \inf_{\underline{\theta} < \theta_1 < \theta_2 \le \overline{\theta}} \frac{Q(h(\theta_1,\theta_2), \theta_2) - Q(h(\theta_1,\theta_2), \theta_1)}{2\left|\frac{\theta_1^j e^{-\theta_1}}{j!} - \frac{\theta_2^j e^{-\theta_2}}{j!}\right|},$$

*where* $h(\theta_1, \theta_2) = \lfloor \frac{\theta_1 - \theta_2}{\ln(\theta_1) - \ln(\theta_2)} \rfloor + 1$ *and* $Q$ *is the regularized gamma function.*

The proof is in App. H.3. The above lower bounds can be computed numerically.

Since these distributions only have one parameter, we can use Alg. 1 and Alg. 3 to derive a data release mechanism. The performance of greedy-based and dynamic-programming-based data release mechanisms for each distribution is shown in Fig. 15.

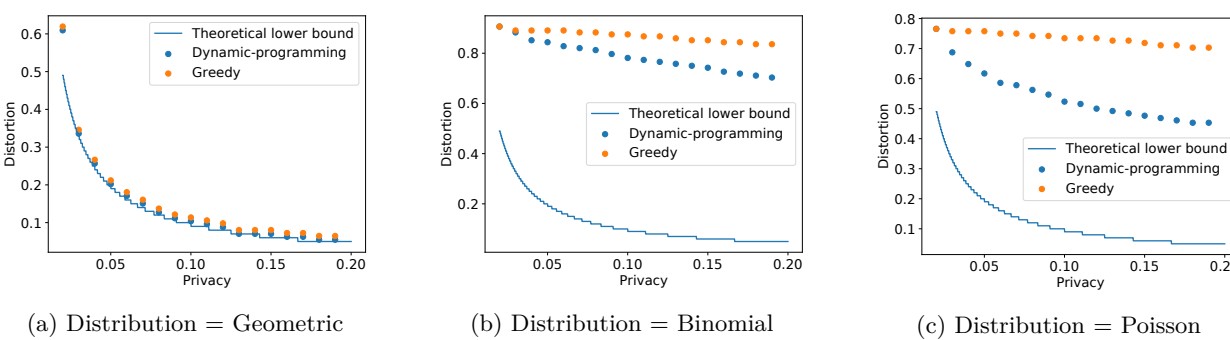

(a) Distribution = Geometric      (b) Distribution = Binomial      (c) Distribution = Poisson

Figure 15: Privacy-distortion performance of Alg. 1 and Alg. 3 for geometric, binomial and Poisson distribution when secret = fraction.

### H.2 Categorical Distribution

In this section, we consider categorical distributions where the fraction of each bin can be changed freely (as long as they are normalized). We assume that $\theta = (p_1, p_2, \ldots, p_C)$ s.t. $p_i \in [0,1]$ $\forall i \in [C]$ and $\sum_i p_i = 1$.

Note that this is completely different from the distributions discussed in App. H.1 where the parameter of the distribution is one-dimensional.

We first analyze the lower bound. Without loss of generality, we assume that we want to protect the fraction of the $j$-th bin, i.e. $p_j$.

**Corollary 8** (Privacy lower bound, secret = fraction of a general discrete distribution). *Consider the secret function $g(\theta) = p_1$. For any $T \in (0,1)$, when $\Pi_{\epsilon,\omega_\Theta} \le T$, we have $\Delta > \left(\lceil \frac{1}{T} \rceil - 1\right) \cdot \epsilon$.*

The proof is in App. H.4. Next, we present the data release mechanism under the following assumption.

**Assumption 7.** *The prior distribution of $(p_1, \ldots, p_C)$ is a uniform distribution over all the probability simplex $\{(p_1, \ldots, p_C) | p_i \in [0,1) \ \forall i \in [C] \text{ and } \sum_i p_i = 1\}$.*

**Mechanism 5** (For secret = fraction of a categorical distribution). *The parameters of the mechanism are as follows.*

$$
\mathcal{S}_{p_1,\ldots,p_C} = \left\{ \left( p_1 - \frac{t}{C-1}, \ldots, p_{j-1} - \frac{t}{C-1}, p_j + t, \right. \right.
$$
$$
\left. \left. p_{j+1} - \frac{t}{C-1}, \ldots, p_C - \frac{t}{C-1} \right) \middle| t \in \left[ -\frac{s}{2}, \frac{s}{2} \right) \right\} ,
$$
$$
\theta^*_{p_1,\ldots,p_C} = \left( p_1 - T, \ldots, p_{j-1} - T, p_j + (C-1)T, \right.
$$
$$
\left. p_{j+1} - T, \ldots, p_{C+1} - T \right) ,
$$

*where $T = \min \{p_1, \ldots, p_{j-1}, p_{j+1}, \ldots, p_C, 0\}$, and*

$$
\mathcal{I} = \left\{ (p_1, \ldots, p_C) \middle| \forall i \ p_i \in \left( -\frac{s}{2(C-1)}, 1 \right], \sum_i p_i = 1, \right.
$$
$$
\left. p_j = (k+0.5)s, \text{where } k \in \{0, 1, \ldots, C-1\} \right\}.
$$

*Here $s > 0$ is a hyper-parameter of the mechanism that divides 1.*

This data release mechanism achieves the following privacy-distortion trade-off.

**Proposition 10.** *Under Asm. 7, Mech. 5 has the following $\Pi_{\epsilon,\omega_\Theta}$ and $\Delta$ value/bound.*

$$
\Pi_{\epsilon,\omega_\Theta} < \frac{2\epsilon}{s} + 1 - \left( 1 - \frac{s}{C-1} \right)^{C-1} ,
$$
$$
\Delta = \frac{s}{2} < \left( 2 + \frac{s}{\epsilon} \right) \Delta_{opt}.
$$

*Under the regime $\sup(s) \to \mathcal{A}\epsilon$, where $\mathcal{A}$ is a constant larger than 2, $\Delta$ satisfies*

$$
\lim_{\sup(s) \to \mathcal{A}\epsilon} \Delta < (2 + \mathcal{A})\Delta_{opt}.
$$

$\Delta_{opt}$ *is the minimal distortion an optimal data release mechanism can achieve given the privacy Mech. 5 achieves.*

The proof is in App. H.5. To ensure that $\Pi_{\epsilon,\omega_\Theta} < 1$, $s$ should satisfy $s > 2\epsilon$. According to Prop. 10, the mechanism is order-optimal with multiplicative factor $2 + \mathcal{A}$ when $\sup(s) \to \mathcal{A}\epsilon$, where $\mathcal{A} > 2$.

### H.3 Proof of Corollary 7

### H.3.1 Geometric Distribution

*Proof.* Let $X_{\theta_1}$ and $X_{\theta_2}$ be two Geometric random variables with parameters $\theta_1$ and $\theta_2$ respectively. We assume that $\theta_1 > \theta_2$ without loss of generality. Let $k'$ satisfy $(1-\theta_1)^{k'}\theta_1 = (1-\theta_2)^{k'}\theta_2$ and $k_0 = \lfloor k' \rfloor + 1$.

Then we can get that

$$D\left(X_{\theta_1}, X_{\theta_2}\right) = \frac{1}{2} d_{\text{TV}}\left(\omega_{X_{\theta_1}} \| \omega_{X_{\theta_2}}\right)$$

$$= \frac{1}{2}\left(1 - \theta_2\right)^{k_0} - \frac{1}{2}\left(1 - \theta_1\right)^{k_0},$$

$$R\left(X_{\theta_1}, X_{\theta_2}\right) = \left|\left(1 - \theta_2\right)^j \theta_2 - \left(1 - \theta_1\right)^j \theta_1\right|.$$

Therefore, we can get that

$$\gamma = \inf_{\underline{\theta} < \theta_1 < \theta_2 \leq \bar{\theta}} \frac{\left(1 - \theta_2\right)^{k_0} - \left(1 - \theta_1\right)^{k_0}}{2\left|\left(1 - \theta_2\right)^j \theta_2 - \left(1 - \theta_1\right)^j \theta_1\right|}.$$

$\square$

### H.3.2   Binomial Distribution

*Proof.* Let $X_{\theta_1}$ and $X_{\theta_2}$ be two binomial random variables with parameters $\theta_1$ and $\theta_2$ respectively with fixed number of trials $n$. We assume that $\theta_1 > \theta_2$ without loss of generality. Let $k'$ satisfy $\binom{n}{k'}\theta_1^{k'}\left(1 - \theta_1\right)^{n-k'} = \binom{n}{k'}\theta_2^{k'}\left(1 - \theta_2\right)^{n-k'}$ and $k_0 = \lfloor k' \rfloor$. We can get that

$$D\left(X_{\theta_1}, X_{\theta_2}\right) = \frac{1}{2} d_{\text{TV}}\left(\omega_{X_{\theta_1}} \| \omega_{X_{\theta_2}}\right)$$

$$= \frac{1}{2} I_{1-\theta_2}\left(n - k_0, 1 + k_0\right) - \frac{1}{2} I_{1-\theta_1}\left(n - k_0, 1 + k_0\right),$$

$$R\left(X_{\theta_1}, X_{\theta_2}\right) = n\left(\theta_1 - \theta_2\right),$$

where $I$ represents the regularized incomplete beta function.

Therefore, we can get that

$$\gamma = \inf_{\underline{\theta} < \theta_1 < \theta_2 \leq \bar{\theta}} \frac{I_{1-\theta_2}\left(n - k_0, 1 + k_0\right) - I_{1-\theta_1}\left(n - k_0, 1 + k_0\right)}{2\left|\binom{n}{j}\theta_2^j\left(1 - \theta_2\right)^{n-j} - \binom{n}{j}\theta_1^j\left(1 - \theta_1\right)^{n-j}\right|}.$$

$\square$

### H.3.3   Poisson Distribution

*Proof.* Let $X_{\theta_1}$ and $X_{\theta_2}$ be two Poisson random variables with parameters $\theta_1$ and $\theta_2$ respectively. We assume that $\theta_1 > \theta_2$ without loss of generality. Let $k'$ satisfy $\theta_1^{k'} e^{-\theta_1} = \theta_2^{k'} e^{-\theta_2}$ and $k_0 = \lfloor k' \rfloor + 1$. Then we can get that

$$D\left(X_{\theta_1}, X_{\theta_2}\right) = \frac{1}{2} d_{\text{TV}}\left(\omega_{X_{\theta_1}} \| \omega_{X_{\theta_2}}\right)$$

$$= \frac{1}{2} Q\left(k_0, \theta_2\right) - \frac{1}{2} Q\left(k_0, \theta_1\right),$$

$$R\left(X_{\theta_1}, X_{\theta_2}\right) = \left|\frac{\theta_1^j e^{-\theta_1}}{j!} - \frac{\theta_2^j e^{-\theta_2}}{j!}\right|,$$

where $Q$ is the regularized gamma function.

Therefore, we can get that

$$\gamma = \inf_{\underline{\theta} < \theta_1 < \theta_2 \leq \bar{\theta}} \frac{Q\left(k_0, \theta_2\right) - Q\left(k_0, \theta_1\right)}{2\left|\frac{\theta_1^j e^{-\theta_1}}{j!} - \frac{\theta_2^j e^{-\theta_2}}{j!}\right|}.$$

$\square$

### H.4 Proof of Corollary 8

*Proof.* Let $X_{p_1^1, p_2^1, \ldots, p_C^1}$ and $X_{p_1^2, p_2^2, \ldots, p_C^2}$ be two categorical random variables. We have

$$
\begin{aligned}
&D\left(X_{p_1^1, p_2^1, \ldots, p_C^1}, X_{p_1^2, p_2^2, \ldots, p_C^2}\right) \\
&= \frac{1}{2} d_{\mathrm{TV}}\left(\omega_{X_{p_1^1, p_2^1, \ldots, p_C^1}} \| \omega_{X_{p_1^2, p_2^2, \ldots, p_C^2}}\right) \\
&\geq \frac{1}{2}\left|p_j^1 - p_j^2\right|, \\
&R\left(X_{p_1^1, p_2^1, \ldots, p_C^1}, X_{p_1^2, p_2^2, \ldots, p_C^2}\right) \\
&= \left|p_j^1 - p_j^2\right|.
\end{aligned}
\tag{25}
$$

Therefore, we can get that

$$
\gamma \geq \frac{1}{2}.
$$

$\square$

### H.5 Proof of Prop. 10

*Proof.* We first focus on the proof for $\Pi_{\epsilon, \omega_\Theta}$.

We separate the space of possible data parameters into two regions: $S_1 = \left\{(p_1, \ldots, p_C) | p_i \in \left[\frac{s}{2(C-1)}, 1 - \frac{s}{2(C-1)}\right] \ \forall i \in [C] \text{ and } \sum_i p_i = 1\right\}$ and $S_2 = \{(p_1, \ldots, p_C) | p_i \in [0, 1) \ \forall i \in [C] \text{ and } \sum_i p_i = 1\} \setminus S_1$. The high-level idea of our proof is as follows. Note that for any parameter $\theta \in S_1$, there exists a $\mathcal{S}_{p_1, \ldots, p_C}$ s.t. $\theta \in \mathcal{S}_{p_1, \ldots, p_C}$ and $\mathcal{S}_{p_1, \ldots, p_C} \subset S_1$. Therefore, we can bound the attack success rate if $\theta \in S_1$. At the same time, the probability of $\theta \in S_2$ is bounded. Therefore, we can bound the overall attacker's success rate (i.e., $\Pi_{\epsilon, \omega_\Theta}$). More specifically, let the optimal attacker be $\hat{g}^*$. We have

$$
\begin{aligned}
\Pi_{\epsilon, \omega_\Theta} &= \mathbb{P}\left(\hat{g}^*\left(\theta'\right) \in [g(\theta) - \epsilon, g(\theta) + \epsilon]\right) \\
&= \int_{\theta \in S_1} p(\theta) \mathbb{P}\left(\hat{g}^*\left(\theta'\right) \in [g(\theta) - \epsilon, g(\theta) + \epsilon]\right) d\theta \\
&\quad + \int_{\theta \in S_2} p(\theta) \mathbb{P}\left(\hat{g}^*\left(\theta'\right) \in [g(\theta) - \epsilon, g(\theta) + \epsilon]\right) d\theta \\
&< \frac{2\epsilon}{s} + \left(1 - \left(1 - \frac{s}{C-1}\right)^{C-1}\right).
\end{aligned}
$$

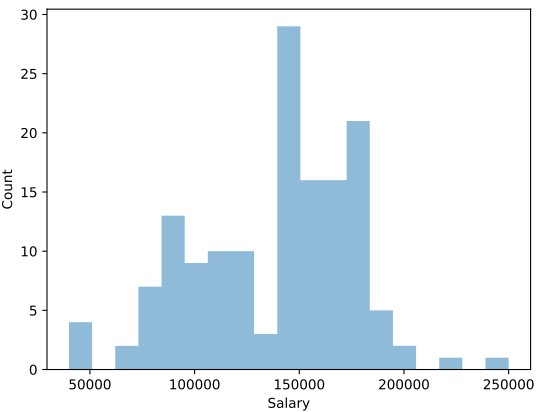

Figure 16: Histogram of salary dataset.

For the distortion, it is straightforward to get that $\Delta = \frac{s}{2}$ from Eq. (25), and $\Delta_{\text{opt}} > \left( \lceil \frac{1}{\Pi_{\epsilon,\omega_\Theta}} \rceil - 1 \right) \cdot \epsilon \geq \epsilon$ from Corollary 2. We can get that $\left( \Pi_{\epsilon,\omega_\Theta} - \left( 1 - \left( 1 - \frac{s}{C-1} \right)^{C-1} \right) \right) \cdot \Delta = \epsilon$ and

$$
\begin{aligned}
\Delta &= \Delta_{\text{opt}} + \Delta - \Delta_{\text{opt}} \\
&< \Delta_{\text{opt}} + \Delta - \left( \lceil \frac{1}{\Pi_{\epsilon,\omega_\Theta}} \rceil - 1 \right) \cdot \epsilon \\
&\leq \Delta_{\text{opt}} + \epsilon + \Delta - \frac{\epsilon}{\Pi_{\epsilon,\omega_\Theta}} \\
&= \Delta_{\text{opt}} + \epsilon + \frac{\left( 1 - \left( 1 - \frac{s}{C-1} \right)^{C-1} \right)}{\frac{2\epsilon}{s} + \left( 1 - \left( 1 - \frac{s}{C-1} \right)^{C-1} \right)} \cdot \Delta \\
&= \left( 1 + \frac{s}{2\epsilon} \left( 1 - \left( 1 - \frac{s}{C-1} \right)^{C-1} \right) \right) (\Delta_{\text{opt}} + 2\gamma\epsilon) \\
&\leq \left( 2 + \frac{s}{\epsilon} \left( 1 - \left( 1 - \frac{s}{C-1} \right)^{C-1} \right) \right) \Delta_{\text{opt}} \\
&< \left( 2 + \frac{s}{\epsilon} \right) \Delta_{\text{opt}}.
\end{aligned}
$$

$\square$

## I   Additional Results

In this section, we provide additional results on how released data from our mechanisms can support downstream applications.

We consider the salaries from people with Master's and PhD degrees in this Kaggle dataset `https://www.kaggle.com/datasets/rkiattisak/salaly-prediction-for-beginer`. We plot its histogram in Fig. 16. We can see that there are two peaks. They correspond to people with age<=40 and age>40 (see Fig. 17).

Assume the goal is to release this dataset and preserve the salary difference between people with age<=40 and age>40, while protecting the mean salaries. We can apply our mechanism for mean (§6.3) on this dataset. The histogram of the released data is shown in Fig. 18. Data receivers can obtain the salary

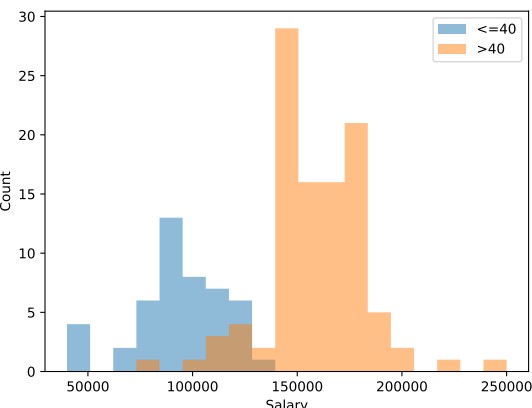

Figure 17: Histogram of salary dataset for people with age $<= 40$ and $> 40$.

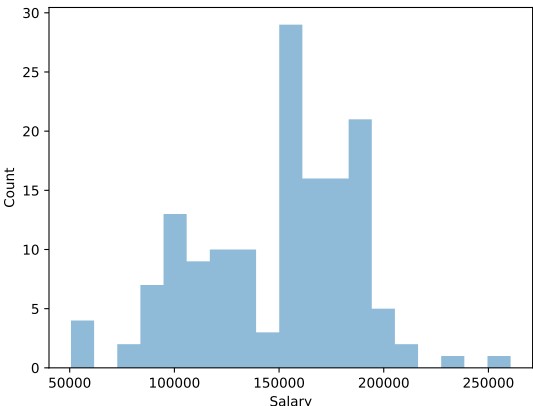

Figure 18: Histogram of salary dataset for applying the mechanism in §6.3.

difference between people with age<=40 and age>40 accurately by computing the difference between the two peaks, while the mean salaries are protected under our mechanism.

Here we use the salary difference between people with age<=40 and age>40 as an example. In general, any downstream tasks that depend only on the "shape" of the distribution will not be affected by our mechanism, since our mechanism shifts all samples by the same amount.

