# OpenReview forum: "Summary Statistic Privacy in Data Sharing"
_TMLR — Rejected by TMLR_

### Review · Reviewer_ctsd · 2023-04-04

**Summary Of Contributions:**

This paper aims to find a way to protect the privacy of statistical metrics when data is released to the public. The authors argue that the current methods used to protect privacy are not effective enough. To address this issue, the authors have developed a new framework that allows data holders to release data while maintaining privacy. They have also calculated the maximum amount of distortion that can occur. Based on this calculation, they have created mechanisms for data release that balance privacy protection and data accuracy. The authors have conducted experiments to show that their approach outperforms existing methods when measured by their specific definitions of privacy and distortion.

**Audience:**

Yes

**Broader Impact Concerns:**

no ethical issues

**Claims And Evidence:**

No

**Requested Changes:**

* Add convincing example why frameworks such as DP with correct sensitivity and mechanism cannot protect the statistics such as mean considering the surrogate metrics used in the work?

* Improve the proofs

* Explore the failures of this privacy framework.


**Strengths And Weaknesses:**



Improving the privacy of data releases, especially in the field of machine learning, is crucial given the current state of data privacy. One way to achieve this is by exploring alternative definitions of privacy that can enhance the effectiveness of data release mechanisms.

The paper argues that using differential privacy doesn’t protect the statistics of the data release because in expectation for example the mean of the data is still the same. Also the referred work does not exactly show this. Based on my understanding differential privacy does not work in expectation and you have to pay a privacy cost for every time you run the mechanism. Specially since the work also shows this in the experiment section that differential privacy has reasonable protection against the defined attacks. Moreover Census data uses differential privacy [R1]. It would be necessary for the work to come up with example where for example differential privacy does not protect the target statistics.

One of the main part of the paper is Theorem 1, however, I had a hard time following the proof. It would be beneficial to the work to maybe do a proof read. In Equation 7 I didn’t follow that the authors went to E[Sup P()] from Sup E[P()]. The argument in the next paragraph also does not read well to me. The authors argue RHS <= LHS, because “gˆ behaves according to θ′ ” but as far as I understand g^ can be any attack therefore I can fix g^ to a fix deterministic function and therefore it would be independent of the θ′. Is there anything specific that I am missing ?

Again in equation 9 the authors use  θ,  θ′,  θi,  θ1,  θ2 which is confusing.

In general the main downside of the work is the fact that this framework uses a specific property of the data as the privacy leakage. What happens if the data holder first only cares about the mean of the data, however, at the later date found out another property of the data is very important. The authors should discuss such scenarios more in the paper.






[R1]  Cohen, Aloni, et al. "Census TopDown: The impacts of differential privacy on redistricting." arXiv preprint arXiv:2203.05085 (2022).

---

> ### Author Response · Authors · 2023-04-11
> **Rebuttal (Q1)**
>
> **Q1: The paper argues that using differential privacy doesn’t protect the statistics of the data release because in expectation for example the mean of the data is still the same. Also the referred work does not exactly show this. Based on my understanding differential privacy does not work in expectation and you have to pay a privacy cost for every time you run the mechanism. Specially since the work also shows this in the experiment section that differential privacy has reasonable protection against the defined attacks. Moreover Census data uses differential privacy [R1]. It would be necessary for the work to come up with example where for example differential privacy does not protect the target statistics.**
>
> Thank you for the suggestion. Just to clarify, are you suggesting a mechanism where one could run a DP mechanism $k$ times, until the desired secret(s) are distorted by at least some amount from the true distribution (and incur a related privacy penalty via composition)? Such a mechanism could perhaps be used—however, the core problem remains that the only formal privacy guarantees we have about this mechanism are DP guarantees: that is, they bound the likelihood ratios of observing an output with respect to datasets that are neighboring (i.e. differ in at most one sample, or some number of samples in the case of group privacy). This does not inherently say anything about protecting the secret distribution statistics. So we could try to analyze this mechanism you propose, but the motivation underlying our work is that we currently lack an analytical framework for analyzing the privacy leakage over these statistics. Our contribution is to propose such a framework, which could (in principle) be used to analyze the mechanism you suggested.

---

> > ### Author Response · Authors · 2023-04-11
> > **Rebuttal (Q2)**
> >
> > **Q2: Add convincing example why frameworks such as DP with correct sensitivity and mechanism cannot protect the statistics such as mean considering the surrogate metrics used in the work?**
> >
> > We would like to clarify that the surrogate metrics are only for visualization purposes; we use them because the probabilistic metrics we analyze are difficult to compute empirically from realizations, without knowing the underlying distributions. The surrogate privacy metric alone does not provide any privacy guarantees. We apologize for the confusion, we will make this more explicit in the paper. One could, however, analyze a DP-based mechanism according to the privacy definition we have defined. We will attempt this analysis for the mechanism you proposed earlier, as we understood it.
> >
> > Suppose we have a dataset of n samples $\{x_1,\ldots, x_n\}$ drawn from a Gaussian distribution $N(\mu, \sigma)$, with empirical mean $\overline \mu$. We are trying to release a full dataset while hiding the mean. Our understanding of your proposed mechanism is that you set a threshold $T$, and add i.i.d. DP noise to each sample in the dataset. Here we can choose any sensitivity and epsilon, because again, we are not trying to protect the presence/absence of a single sample. Suppose we choose sensitivity $\Delta=1$ and $\epsilon=1$, so we add $u_i \sim Lap(1)$ noise to each sample $x_i$. Now, the protocol adds this noise to each sample in the dataset i.i.d., then checks if the mean is larger than or equal to $T$. If not, it repeats the procedure (incurring a penalty to $\epsilon$, which can be computed using composition theorems. However, again, we do not care about this penalty because our current goal is not to hide the presence/absence of a single sample. One could also use the sparse vector technique to reduce the penalty in $\epsilon$, but this is unnecessary since we are not trying to protect DP.
> >
> > Now, we can reason about the total noise added by looking at the normalized sum of the noise: $N=1/n * \sum_{i=1}^n u_i$, where $u_i$ denotes the noise added to sample $x_i$. Suppose the perturbed data has an empirical mean of $\overline \mu_U = \overline \mu + N$. The adversary’s goal is to estimate $\overline \mu$ from the observed data; we call this estimate $\hat \mu$. That is, we want $P(\hat \mu \in [\overline \mu - \epsilon_0, \overline \mu + \epsilon_0]|x_i + u_i~~\forall i\in [n], |N| \geq T)$ (we use $\epsilon_0$ to distinguish from the earlier $\epsilon$ used in the DP mechanism). Assuming a uniform prior over the true mean, we can use a max-likelihood estimate for $\hat \mu$. Hence $\hat \mu = \overline \mu_U - T$ w.p. 0.5, and $\hat \mu = \overline \mu_U + T$ w.p. 0.5. We can use concentration bounds to argue that $| |N|-T|$ is small (say within a threshold  with high probability. Hence, we can compute (with high probability) the probability that the adversary estimates the secret (in this case $\overline \mu$ within some threshold $\epsilon_0$; the form will be 0.5 plus a term that is exponentially vanishing in $\epsilon_0$.
> >
> > One could improve the privacy of this scheme by having the threshold be randomized and unknown to the adversary, but the more salient point is that we cannot analyze the privacy of the desired summary statistic (in this case, empirical mean) using the classical DP privacy framework because it is not designed to protect this type of data.

---

> > > ### Author Response · Authors · 2023-04-11
> > > **Rebuttal (Q3, Q4, Q5)**
> > >
> > > **Q3: \(Improve the proofs.\) One of the main part of the paper is Theorem 1, however, I had a hard time following the proof. It would be beneficial to the work to maybe do a proof read. In Equation 7 I didn’t follow that the authors went to E[Sup P()] from Sup E[P()]. The argument in the next paragraph also does not read well to me. The authors argue RHS <= LHS, because “gˆ behaves according to θ′ ” but as far as I understand g^ can be any attack therefore I can fix g^ to a fix deterministic function and therefore it would be independent of the θ′. Is there anything specific that I am missing ?**
> > >
> > > We apologize for the confusion over the explanation for Equation 7. For the RHS<=LHS argument, the point is that $\hat{g}$ can be affected (only) by $\theta’$. Note that $\hat g$ is the function being optimized over, so one could (as you observe) choose a deterministic $\hat g$ that is independent of $\theta’$, but this needs not be the optimizer (in fact, it generally won’t be). Since the LHS takes the supremum over $\hat{g}$, it is sufficient to find one attack strategy that depends only on $\theta’$ and map any $\arg \sup_{\hat{g}}$ in the RHS to LHS and obtain the same value (since the expectation is taken over $\theta’$). We will clarify this point in the proof, thank you for mentioning it.
> > >
> > >
> > > **Q4: Again in equation 9 the authors use θ, θ′, θi, θ1, θ2 which is confusing.**
> > >
> > > We are sorry that the notation caused confusion. $\theta$ is the private parameter vector that fully specifies the distribution, $\theta’$ is the released parameter vector through the data release mechanism. $\theta_1$ and $\theta_2$ are two different possible values of the private parameter vectors, and $\theta_i (i\in {1,2})$ represents $\theta_1$ and $\theta_2$ when $i=1$ and $i=2$ respectively. We felt it would be clearer to use $\theta_1$ and $\theta_2$ for this notation rather than introduce new letter variables, since they are possible values of $\theta$. We will add more explanation of the notation surrounding this equation to avoid confusion.
> > >
> > >
> > > **Q5: (Explore the failures of this privacy framework.) In general the main downside of the work is the fact that this framework uses a specific property of the data as the privacy leakage. What happens if the data holder first only cares about the mean of the data, however, at the later date found out another property of the data is very important. The authors should discuss such scenarios more in the paper.**
> > >
> > > It is true that we do not currently know how to defend against a threat model that is changing dynamically over time. However, the problem is nontrivial even in the static setting.  For this reason, we believe there is value in exploring the question in a simplified setting to begin, and building up to more complex and realistic scenarios.

---

### Review · Reviewer_Et4z · 2023-04-05

**Summary Of Contributions:**


The paper proposes a new approach to provide privacy and quantify privacy-accuracy (here, distortion) trade-offs, when the goal is to protect dataset-level properties or attributes (such as, for example, the mean value of some column of a dataset). More details on the contributions in "strengths"

**Audience:**

No

**Broader Impact Concerns:**

N/A. The point is to protect privacy so the goal of the paper is well aligned with broader impact considerations.

**Claims And Evidence:**

No

**Requested Changes:**

I do not think the paper can really be changed in a way that addresses my concerns without completely changing the results. I am sorry to say that I currently do not see a path to acceptance, unless I am seriously misunderstanding something about the contribution of the paper (in which case I would be happy to revise my review).

**Strengths And Weaknesses:**


The paper proposes a new approach to provide privacy and quantify privacy-accuracy (here, distortion) trade-offs, when the goal is to protect dataset-level properties or attributes (such as, for example, the mean value of some column of a dataset).


First, in terms of strengths:
- Within their framework, the authors provide a lower bound (Theorem 1) on the privacy-distortion trade-off. This lower bound comes with nice and intuitive properties in how it scales with the privacy budget, and a parameter $\gamma$ that measures how much of an effect a change in the value of the secret (the dataset-level property we are trying to protect) changes the data distribution. Intuitively, if a small change in the secret creates a large in the generated data, privacy is hard to protect information-theoretically
- Sections 6.1 and 6.2. are nice in that the authors show a couple of cases of practical interests (computing means and quantiles) where they develop mechanisms that essentially approximately achieve the lower bound on privacy-distortion trade-offs.

However, I have major concerns about the paper, its motivation, and how it inscribes itself in the literature. My concerns are the below:
1) I am a bit confused about the discussion of other privacy techniques. Some points here:
- My understanding is that the authors claim that differential privacy is innaproppriate here for several reasons. One of the reasons is that it protects the data of each individual data point, rather than some property of the entire dataset. But I want to point out here that group privacy can translate a notion of privacy at the level of each individual data point to the level of the whole dataset. I found the discussion there inaccurate and confusing and I think this may need to be made more careful.
- The authors acknowledge work on releasing dataset-level statistics while preserving privacy (e.g. Zhang et al., 2022, which I will focus on because I am more familiar with this paper than others). One critique the authors make is that they focus on "low-dimension statistical queries of the dataset instead of the entire datase". But this is also what the current paper is doing (aside from Section 6.3., but it only contains an overall framework with no real analysis and evidence that it does something reasonable), so I do not quite understand the novelty. I also think the authors missed the point of Zhang et al. 2022: the point of that paper is to deal with correlations across different columns of a dataset, and the problem of "how do I release information about the dataset while not revealing information about one secret in the dataset, and deal with the fact these two things might be correlated". The current paper takes a different point of view of "how to I reveal information about the current secret but not too much information" (e.g., in section 6.1., the protected secret is the mean, and the technique releases quantized estimates of the mean to reduce the information released, if I understood well). This comparison, is, I believe, apple-to-oranges. Also, on that note, Section 7 claims to compare their results to that of Zhang et al. 2022, but the considered benchmark (simply adding Gaussian noise to the secret) is a simplified and inacurrate version of what Zhang et al. 2022 (where the Gaussian noise has to be carefully computed to take correlation across the query we want to answer, F(X), and the dataset-level attributes we are aiming to protect). This leads me to points 2 and 3 below.
2) A major issue seems to be that the paper does not seem to deal with correlations. To give an example of this, I will go to the end of p7/beginning of p8 and the corresponding Gaussian example, as well as protecting the mean in Section 6.1. There, the goal is to hide the mean of a distribution (the secret). Say, that I focus on a Gaussian distribution like in p7-8, and I apply the mechanism proposed by the authors (deterministic quantization) on the mean. Now imagine that I know there is correlation between the mean and the variance of my Gaussian distribution (e.g. the prior is a joint distribution of $(\mu,\sigma)$ rather than two independent distributions); then, no matter what I do in terms of quantization to protect the mean, if I perfectly release the variance, I release information about the mean through the correlation between mean and variance. This, to the best of my knowledge, seems not to be taken into account in the paper (which focuses only on quantizing the secret itself without thinking about/modeling what information is released by other correlated attributes), which is a step back compared to previous work (e.g. Zhang et al 2022, etc.)  and potentially a major oversight from a privacy point of view.
3) Finally, I am not sure the problem asked by the paper makes practical sense. Effectively, what the paper asks for is "can I release a dataset-level property without releasing much information about said dataset-level property?". It's basically effectively asking for something that is impossible regardless of the frameworks/techniques used: reveal an accurate statistic, about, say, the mean of the population, without anyone learning information about the statistic of the population. You can only have one of those at a time.
The quantization mechanism for mean estimation, for example, to have good distortion/accuracy, must narrow down a small bin in which the true parameter belongs; doing this effectively reveals the mean/secret itself to high accuracy, and no reasonable privacy is then possible. Of course, you can still quantify how much distortion you can get versus privacy and try to understand this trade-off in a fine grained manner like the paper does, but I think this is missing a next step to be of practical interest here.
Something that could be useful for example is to say (this is a possible direction, but may not be the right one): "I have access to n agents, each agent has a secret and I do not want to learn the secret of that specific agent... but I am interested in learning the average secret over all agents" (note this is what DP already does but a comparison could be interesting in terms of the different metric of privacy proposed here). Also note that there are no composition guarantees here unlike in DP (for example); in fact, it is not hard to show that I could run two quantization mechanisms with relatively big values of the step sizes s and s' but that are very close to each (which, in the author's framework, would provide good privacy), yet for some values of the secret I could narrow down a very small bin to which it belong (e.g., the secret belongs in [u + s, u + s']) providing close to no privacy.
4) Some of the results feel incomplete. E.g., section 5, where the privacy analysis is written in terms of a hard-to-interpret integral, with a handwavy justification that some terms are bounded away from 1. SUch arguments need to be made formal and precise for the paper to be self-contained.

For the reason above, I am voting to reject the paper. There is an interesting high-level information-theoretic approach, but the motivation and the actual implementation (e.g. in the case studies) have, I believe, some serious issues.

---

> ### Author Response · Authors · 2023-04-11
> **Rebuttal (Q1)**
>
> We sincerely thank the reviewer for the detailed and insightful feedback on our paper! We believe that some critical concerns may be due to misunderstandings. We give explanations to each of the questions below, and we will update our paper to reduce potential confusion to future readers. We look forward to discussions with the reviewer about any further questions or concerns.
>
> **Q1: The authors acknowledge work on releasing dataset-level statistics while preserving privacy (e.g. Zhang et al., 2022, which I will focus on because I am more familiar with this paper than others). One critique the authors make is that they focus on "low-dimension statistical queries of the dataset instead of the entire dataset". But this is also what the current paper is doing (aside from Section 6.3., but it only contains an overall framework with no real analysis and evidence that it does something reasonable), so I do not quite understand the novelty.**
>
> We want to clarify that our goal is NOT to release “low-dimension statistical queries”, but to release “the entire dataset” (see the examples in Section 1, the motivating scenarios in Section 2, the formulation in Section 3, etc.). So while (Zhang et al., 2022) is a foundational piece of work in the area of protecting distributional secrets and provides significant insights in this direction, its goals are different from ours. Since the reviewer is familiar with Zhang et al., 2022 (https://arxiv.org/pdf/2009.04013.pdf), let us explain the difference using Example 1 in Zhang et al., 2022. “Consider a dataset that consists of students’ SAT scores X_s, heights X_h, weights X_w, gender X_g, and their family income X_i”. In Zhang et al., 2022, the goal can be “**releasing the number of students that are taller than 5’6’**, while protecting the distribution of family income of their students”. In our problem, the goal is to “**release another dataset that consists of students’ SAT scores, heights, weights, gender, and their family income \(the identity of these students need not have any relationship with the students in the original data\)**” while “protecting the distribution of family income” of the students in the original dataset.
>
> We guess that the reviewer’s confusion might come from the following sources:
>
> (1) Throughout the majority of the analysis, we assume that the data holder knows $\theta$, the parameter of the distribution from which the samples of the original dataset are drawn from, and the data release mechanism outputs another parameter $\theta’$ from which the released samples are drawn from (see Section 3). We want to clarify that: (a) In general, releasing $\theta’$ is strictly more informative than releasing any specific statistical queries of the dataset (as in Zhang et al., 2022), because $\theta’$ fully describes the distribution, releasing $\theta’$ allows one to compute **any \(i.e., infinite number of\)** statistical queries of the dataset. (b) We analyze based on the distribution parameter $\theta$ instead of the dataset itself for the purpose of cleaner mathematical expressions and analysis. Each value in $\theta$ does not need to have specific statistical meaning (e.g., mean, std); they can be defined implicitly from a dataset. For example, in our analysis of secret=mean (Section 6.1), we say $\theta=(u, v)$, where $v$ can be any vector that defines all distribution properties that are unrelated to the shift. In general, the dimension of this $v$ vector could be arbitrarily large. In Section 6.3, we give an example of how to relate the analysis and mechanisms on the distribution parameters to datasets.
>
> (2) We use Gaussian distributions in many working examples (e.g., Figure 5, Eq. 11). Therefore, it may appear that we want to release $\mu$ while keeping $\sigma$ secret (Eq. 11) or release $\sigma$ while keeping $\mu$ secret (Figure 5). However, we want to clarify that it is just a toy example to make it easier to understand. Some of our results are more general than that (e.g., Section 6.1 explained above).
>
> (3) In some of our analysis (e.g., Section 6.2), we restrict the distribution to certain classes. In those cases, releasing the distribution parameter is indeed equivalent to releasing certain statistical queries of the data. But those statistical queries could represent the entire distribution, and therefore it supports any other statistical queries on the released data. In addition, as explained above, our framework and some of our results are more general than that. The current results are already non-trivial, in our view. Extending the results to more general distributions is an important future work (Section 8).

---

> > ### Author Response · Authors · 2023-04-11
> > **Rebuttal (Q2, Q3)**
> >
> > **Q2: I also think the authors missed the point of Zhang et al. 2022: the point of that paper is to deal with correlations across different columns of a dataset, and the problem of "how do I release information about the dataset while not revealing information about one secret in the dataset, and deal with the fact these two things might be correlated". The current paper takes a different point of view of "how to I reveal information about the current secret but not too much information" \(e.g., in section 6.1., the protected secret is the mean, and the technique releases quantized estimates of the mean to reduce the information released, if I understood well\). This comparison, is, I believe, apple-to-oranges.**
> >
> > First, we would like to clarify that the problem we study is NOT to "how to reveal information about the current secret but not too much information", but to “release the entire dataset but not reveal much about the secret quantity”. The quantization mechanism is an artifact of the fidelity goal of minimizing the overall distribution distortion (Eq. 4), instead of trying to “reveal not too much information about the secret”. See the answer to Q1 for more detailed discussions of the goal of the paper.
> >
> > Secondly, we agree and understand that the goal of Zhang et al. 2022 is different from ours. In fact, our goal in Section 2 was to explain that existing privacy frameworks (including Zhang et al. 2022) are considering different problem settings and/or objectives to ours, which motivates the need for a new privacy framework. That being said, we agree that our experiments in Section 7 are an “apple-to-oranges” comparison. However, these experiments are NOT meant to say that existing privacy frameworks are bad **for their purposes**. Instead, the goal is to convey that they are not suitable **for our problem**, confirming our claims in Section 2. This point has been highlighted in the last paragraph of Section 7, but we will make it clearer to reduce confusion.
> >
> > We included these experiments because it may be natural for readers to ask how existing mechanisms from other privacy frameworks perform on our problem (even if they do not consider the same problem); hence, we felt these experiments were necessary.
> >
> > **Q3: Also, on that note, Section 7 claims to compare their results to that of Zhang et al. 2022, but the considered benchmark \(simply adding Gaussian noise to the secret\) is a simplified and inaccurate version of what Zhang et al. 2022 \(where the Gaussian noise has to be carefully computed to take correlation across the query we want to answer, F\(X\), and the dataset-level attributes we are aiming to protect\). This leads me to points 2 and 3 below.**
> >
> > We apologize for the misunderstanding–we do understand how the mechanisms in Zhang et al. 2022 work, e.g., “the Gaussian noise has to be carefully computed”. But that Gaussian std computation formula is for a given privacy parameter **in the definitions from  Zhang et al. 2022.** As mentioned in the answer to Q1 and in Section 2, **we consider different privacy metrics**, and our goal is to understand privacy-distortion trade-off under our metrics. Therefore, it is not possible to compute a single Gaussian std using the formula in Zhang et al. 2022 to obtain a one-to-one, fair correspondence to our privacy metric–the metrics are simply different. What we did is to apply mechanisms in Zhang et al. 2022 (adding Gaussian noise) with a wide range of Guassian std to visualize its achievable range of privacy-distortion metrics (each of which gives a different attribute privacy guarantee). We felt this would be more comprehensive and fair than representing Zhang et al. 2022 using a single Gaussian std. We will add more explanation to Section 7.1 to motivate why such a parameter sweep is necessary. If you have any suggestions on how to improve this comparison, we will be happy to implement it.

---

> > > ### Author Response · Authors · 2023-04-11
> > > **Rebuttal (Q4)**
> > >
> > > **Q4: A major issue seems to be that the paper does not seem to deal with correlations. To give an example of this, I will go to the end of p7/beginning of p8 and the corresponding Gaussian example, as well as protecting the mean in Section 6.1. There, the goal is to hide the mean of a distribution \(the secret\). Say, that I focus on a Gaussian distribution like in p7-8, and I apply the mechanism proposed by the authors \(deterministic quantization\) on the mean. Now imagine that I know there is correlation between the mean and the variance of my Gaussian distribution \(e.g. the prior is a joint distribution of $(\mu, \sigma)$ rather than two independent distributions\); then, no matter what I do in terms of quantization to protect the mean, if I perfectly release the variance, I release information about the mean through the correlation between mean and variance. This, to the best of my knowledge, seems not to be taken into account in the paper \(which focuses only on quantizing the secret itself without thinking about/modeling what information is released by other correlated attributes\), which is a step back compared to previous work \(e.g. Zhang et al 2022, etc.\) and potentially a major oversight from a privacy point of view.**
> > >
> > > We agree with you that our proposed mechanisms do not take this into account, and this is indeed a weakness. However, our general framework does cover this situation, with some caveats. More precisely, the correlation between different parameters can be captured by the prior distribution over the distribution parameters (see the 1st paragraph in Section 3). When mean and variance are correlated, we can model that using a prior distribution where mean and variance are not independent. Our framework and the general analysis (Section 3, Section 4) are applicable to any prior distribution. However, you are correct that in the case studies (Section 6), we assume independent prior distributions (i.e., we assume that such correlation does not exist). We agree that extending our mechanisms to handle such cases would be an interesting and important future work, potentially by combining the insights from Zhang et. al. 2022. We will clarify this point in the limitations section that we propose to add.

---

> > > > ### Author Response · Authors · 2023-04-11
> > > > **Rebuttal (Q5, Q6)**
> > > >
> > > > **Q5: Finally, I am not sure the problem asked by the paper makes practical sense. Effectively, what the paper asks for is "can I release a dataset-level property without releasing much information about said dataset-level property?". It's basically effectively asking for something that is impossible regardless of the frameworks/techniques used: reveal an accurate statistic, about, say, the mean of the population, without anyone learning information about the statistic of the population. You can only have one of those at a time. The quantization mechanism for mean estimation, for example, to have good distortion/accuracy, must narrow down a small bin in which the true parameter belongs; doing this effectively reveals the mean/secret itself to high accuracy, and no reasonable privacy is then possible. Of course, you can still quantify how much distortion you can get versus privacy and try to understand this trade-off in a fine grained manner like the paper does, but I think this is missing a next step to be of practical interest here. Something that could be useful for example is to say \(this is a possible direction, but may not be the right one\): "I have access to n agents, each agent has a secret and I do not want to learn the secret of that specific agent... but I am interested in learning the average secret over all agents" \(note this is what DP already does but a comparison could be interesting in terms of the different metric of privacy proposed here\).**
> > > >
> > > > We want to clarify (again) that the goal is NOT to “release a dataset-level property” or “reveal an accurate statistic”, but to release the entire dataset. Please refer to the answer to Q1 for detailed explanations.
> > > >
> > > > We believe this is a valid question to consider. In addition to “the dataset-level [secret] property” that the data holder wants to obfuscate, there are many other properties of data that we wish to also release. Our objective (Eq. 4) is to minimize the distribution distortion while maintaining bounded privacy leakage. First, it is a practically meaningful question. As explained in Section 1 and Section 2.1, there are many practical scenarios where data holders need to share a dataset (instead of only sharing statistical queries of the dataset). This happens already in practice, and is very common in the ML community (e.g., Kaggle).  At the same time, data holders often want to hide some specific dataset-level properties, as highlighted in Zhang et. al. 2022 and others;  anecdotally, we have spoken with multiple large enterprises that encounter exactly this problem, and do not have a satisfactory solution today. Second, this problem is technically challenging. For example, let’s consider the simple case studied in Section 6.2, where we want to hide the quantile of (samples from) a shifted exponential distribution, while minimizing the distribution distortion. It is non-trival to reason about how to change other parameters of the distributions in order to minimize the distribution distortion. Our analysis gives answers to these questions. More examples are given in the Appendix (summarized in Table 1).
> > > >
> > > > **Q6: Also note that there are no composition guarantees here unlike in DP \(for example\); in fact, it is not hard to show that I could run two quantization mechanisms with relatively big values of the step sizes s and s' but that are very close to each \(which, in the author's framework, would provide good privacy\), yet for some values of the secret I could narrow down a very small bin to which it belong \(e.g., the secret belongs in [u + s, u + s']\) providing close to no privacy.**
> > > >
> > > > It is true that our privacy definition does not provide composition guarantees, and this can be problematic in situations where a data holder wants to release a dataset (or correlated datasets) multiple times. We will add this limitation to the section on limitations that we plan to add.
> > > >
> > > > However, we would like to add that in many use cases, data release may be done in one shot, e.g. for providing data to the research community. For example, Google, Alibaba, and Microsoft have all released one-time datasets with cluster traces (Section 2.1). In such cases, composition guarantees may not be necessary.

---

> > > > > ### Author Response · Authors · 2023-04-11
> > > > > **Rebuttal (Q7, Q8)**
> > > > >
> > > > > **Q7: Some of the results feel incomplete. E.g., section 5, where the privacy analysis is written in terms of a hard-to-interpret integral, with a handwavy justification that some terms are bounded away from 1. SUch arguments need to be made formal and precise for the paper to be self-contained.**
> > > > >
> > > > > We apologize for the confusion. Section 5 is meant to be a high-level overview of the mechanisms and results that we use in Section 6. We wrote it this way to make it easier for readers to understand the high-level strategies before we go into details. Therefore, it contains expressions and claims that are proved only in Section 6. Indeed, all of these expressions and claims are made precise in Section 6 and the appendix (see Table 1 for the index of all the results). For example, the “justification that some terms are bounded away from 1” is given in Proposition 2, Appendix C.4.2, F.2, G.4. We will clarify this point early in Section 5, and polish the writing to reduce confusion.
> > > > >
> > > > > **Q8: My understanding is that the authors claim that differential privacy is inappropriate here for several reasons. One of the reasons is that it protects the data of each individual data point, rather than some property of the entire dataset. But I want to point out here that group privacy can translate a notion of privacy at the level of each individual data point to the level of the whole dataset. I found the discussion there inaccurate and confusing and I think this may need to be made more careful.**
> > > > >
> > > > > Thank you for pointing it out. We agree that group privacy can be used to protect the privacy of multiple correlated samples. However, if the number of samples that are correlated is high (e.g., in our case, it can be as large as $n$, the size of the database), then the resulting DP guarantees are too weak to be meaningful. We did not initially include this in our discussion for that reason, but we would be happy to add it.

---

### Review · Reviewer_oLdq · 2023-04-24

**Summary Of Contributions:**

This paper studies the problem of sharing data when some of the properties of the underlying distribution are sensitive and must be kept private. They define a notion of what “privacy” and “distortion” mean in this setting and design algorithms that achieve the optimal trade-off between the definition of these notions that they define.

**Audience:**

Yes

**Broader Impact Concerns:**

I have no ethical concerns.

**Claims And Evidence:**

Yes

**Requested Changes:**

- Evaluate performance on downstream tasks to provide evidence that produced synthetic data is useful.
- Discuss whether or not there exists a variant of DP that could be appropriate (e.g. by defining databases as neighboring if their "secret" summary statistics are close?) This would provide composability of the privacy guarantee.


**Strengths And Weaknesses:**

Comments:
This paper is well-written and easy to follow. I did not read all the proofs in fine detail, but I did not see any technical errors.

My main concern with this paper is the use of the metric defined in eqn (2) as a measure of privacy, and the ability of the quantization mechanism to provide meaningful privacy guarantees.
- The guarantee does not compose. That is, if I release the data (or summary statistics) using the quantization mechanism twice with two different quantizations, the combination of the two results may leak a lot more than intended. This makes this technique difficult to use in practice unless the original dataset is disgarded after a single use.
- My understanding is that the bound on the privacy metric in the examples given in section 6 only hold when a particular prior belief is held by the adversary. The authors discuss this towards the end of the paper. This seems like a major drawback since in practice, one would not know the adversaries prior.

I’m not convinced that there is not a variant of DP that is appropriate in this setting (although I agree that none of the versions that the authors discuss are appropriate). For example, in the example where the mean is the secret, one could protect the mean, up to a variation of size Delta, by adding noise Lap(Delta/eps) to the mean. This would likely have worse distortion than the quantization mechanism, but possibly not by too much, and would provide composable privacy guarantees. I suspect one could formalize the guarantee in this particular setting by defining two databases in the definition of DP to be “neighboring” if their means differ by at most Delta. Since the notion of summary statistic privacy presented in this paper seems much weaker than a definition like this, I think there should be convincing evidence that this weakening is necessary.

The motivation of protecting business secrets that are properties of the underlying distribution seems reasonable. However, I didn’t think the authors talked enough about what they hoped to maintain about the database, and whether or now the goals were fundamentally at odds (although I guess the lower bound on the distortion indicates that the goals are at odds). The experiments show that the quantization mechanism achieves the optimal privacy/distortion trade-off, but is the data released useful for downstream tasks? The authors don’t really address this in their discussion of examples. The example of wanting to release the salary difference between men and women without revealing the mean salaries seems like an achievable goal, and perhaps would have been a useful task to measure the success on.

The authors state several times that standard DP (and the variants they discuss) solve a different problem than the one the authors are attempting to address. However, they still use this as a comparison point in the experiments. This isn’t a fair comparison since these methods are basically designed to preserve summary statistics. Thus, the concentration of the DP-based methods in the low distortion, low “privacy” regime in Fig 6 is a feature, not a bug. Something like adding noise to the summary statistic (as described above for the mean) seems like a much more fair comparison.

Minor comments:
- The citation for DP should be “Calibrating Noise to Sensitivity in Private Data Analysis” by Dwork, McSherry, Nissim and Smith. It’s not true that DP has focused on low-dimensional statistical queries, there is a good amount of work on DP synthetic data (although I agree that it is usually in a different setting to this papers focus).
- It would have been helpful to have the formulation of summary statistic privacy before the discussion of indistinguishability approaches to provide a comparison point (since I was familiar with these approaches and so wanted a comparison point). In particular, I struggled to understand throughout this section what the authors hoped to retain about the data.
- Just before eq 5, it is stated that d is defined in eqn 3, but it is not.

---

> ### Author Response · Authors · 2023-05-03
> **Rebuttal (Q1, Q2, Q3)**
>
> **Q1: The guarantee does not compose. That is, if I release the data (or summary statistics) using the quantization mechanism twice with two different quantizations, the combination of the two results may leak a lot more than intended. This makes this technique difficult to use in practice unless the original dataset is disgarded after a single use.**
>
> It is true that our privacy definition does not provide composition guarantees, and this can be problematic in situations where a data holder wants to release a dataset (or correlated datasets) multiple times. We will add this limitation to the section on limitations that we plan to add.
>
> However, we would like to add that in many use cases, data release may be done in one shot, e.g. for providing data to the research community. For example, Google, Alibaba, and Microsoft have all released one-time datasets with cluster traces (Section 2.1). In such cases, composition guarantees may not be necessary.
>
> **Q2: My understanding is that the bound on the privacy metric in the examples given in section 6 only hold when a particular prior belief is held by the adversary. The authors discuss this towards the end of the paper. This seems like a major drawback since in practice, one would not know the adversaries prior.**
>
> You are correct that in the case studies (Section 6), the mechanisms only work for a particular prior distribution (discussed in Section 8). However, we want to point out that: (1) Our framework and the general analysis (Section 3, Section 4) are applicable to any prior distribution. (2) We agree that extending the mechanisms to handle any prior distribution is important. But we believe that understanding the mechanisms for simple prior distributions is a (nontrivial) first step towards this goal. Building on our results, we discussed one potential approach to extend our mechanisms and/or framework to handle any prior distributions in Section 8, which we will be studying as future work.
>
> **Q3: I’m not convinced that there is not a variant of DP that is appropriate in this setting (although I agree that none of the versions that the authors discuss are appropriate). For example, in the example where the mean is the secret, one could protect the mean, up to a variation of size Delta, by adding noise Lap(Delta/eps) to the mean. This would likely have worse distortion than the quantization mechanism, but possibly not by too much, and would provide composable privacy guarantees.**
>
> This is an interesting idea. It won’t achieve a good tradeoff under our current formulation, because our distortion metric (Eq. 3) is defined as the worst-case distance between the original and the release distributions. Therefore, the mechanism you proposed would give infinity, the worst possible distortion, as Laplacian noise is unbounded.
>
> However, we could perhaps consider a different formulation in which we trade off privacy for expected distortion (rather than worst-case), as is commonly done in the DP literature. In that case, we might be able to get composition for that particular mechanism. It would not be as trivial as using the composition guarantees for differential privacy, because our metric is different. We would need to analyze our privacy metric (roughly, the probability of guessing the true mean) under the k-fold application of this mechanism.

---

> > ### Author Response · Authors · 2023-05-03
> > **Rebuttal (Q4, Q5)**
> >
> > **Q4: (Discuss whether or not there exists a variant of DP that could be appropriate (e.g. by defining databases as neighboring if their "secret" summary statistics are close?) This would provide composability of the privacy guarantee.) I suspect one could formalize the guarantee in this particular setting by defining two databases in the definition of DP to be “neighboring” if their means differ by at most Delta. Since the notion of summary statistic privacy presented in this paper seems much weaker than a definition like this, I think there should be convincing evidence that this weakening is necessary.**
> >
> > Thank you for the suggestions. Indeed, we have studied these choices at the beginning of our exploration. One downside of using DP is that the amount of noise is too large due to its strong guarantee. For example, let’s consider “defining two databases in the definition of DP to be neighboring if their means differ by at most Delta” you proposed. If we do not restrict the set of distributions considered, these two distributions could differ a lot in other properties (e.g., considering $N(0, 1)$ and $N(0.001, 10^{10})$, and Delta>0.001). To make their outputs indistinguishable, the amount of noise needed to add is so large that the fidelity will be completely destroyed (e.g., one will not be able to recover the standard deviation of the Gaussian accurately). There are relaxations of DP that one can use for handling these issues (see discussions in Section 2.2.1). However, some assumptions on the prior distributions or neighboring definitions need to be made, which will raise similar concerns as your Q2. Therefore, instead of doing these ad-hoc fixes on DP, we considered starting from a clean slate and designing a new framework from scratch. (But we agree that building on top of DP is still a valuable direction to consider.)
> >
> > **Q5: The motivation of protecting business secrets that are properties of the underlying distribution seems reasonable. However, I didn’t think the authors talked enough about what they hoped to maintain about the database, and whether or now the goals were fundamentally at odds (although I guess the lower bound on the distortion indicates that the goals are at odds).**
> >
> > This is a good question. In general, we do not know what properties data holders and data users will require for their applications. Therefore, our approach in this paper is to try to preserve “as much as possible” about the original distribution, conditioned on hiding the relevant secret function. We considered various metrics for how to quantify what we preserve, and eventually settled on the Wasserstein-1 distance for a few reasons: (1) it is a general-purpose distributional distance, so it does not require prior knowledge of how the data will be used, and (2) it is analytically relatively easy to work with. One note is that Wasserstein-1 is sensitive to the exact numeric values of the distribution. Hence, it may not be the best metric if the data user is only interested in the “shape” of the data. However without knowing this a priori, we felt Wasserstein-1 distance is a reasonable guess.
> >
> > As you correctly observed, there is a tension between hiding summary statistics and reducing this distributional distance between released and original data, as evidenced by the lower bounds we demonstrated. However, we can still release useful information despite this tension, because distributions may have complex dependencies that can be preserved, even while hiding certain summary statistics. For instance, quantization is commonly used to release datasets in practice, such as by the U.S. census. (Their privacy goals when quantizing data are quite different from ours, but the main point is that in spite of quantizing their data, the released data is still useful for downstream applications.)

---

> > > ### Author Response · Authors · 2023-05-03
> > > **Rebuttal (Q6, Q7)**
> > >
> > > **Q6: (Evaluate performance on downstream tasks to provide evidence that produced synthetic data is useful.) The experiments show that the quantization mechanism achieves the optimal privacy/distortion trade-off, but is the data released useful for downstream tasks? The authors don’t really address this in their discussion of examples. The example of wanting to release the salary difference between men and women without revealing the mean salaries seems like an achievable goal, and perhaps would have been a useful task to measure the success on.**
> > >
> > > Thank you for the great suggestion! Yes, what you suggest is a compelling example. Instead, we conduct the following experiment to illustrate it.
> > >
> > > We consider the salaries from people with Master’s and PhD degrees in this dataset https://www.kaggle.com/datasets/rkiattisak/salaly-prediction-for-beginer from Kaggle. We plot its histogram in this anonymized link https://ibb.co/CzHnJ9z. We can see that there are two peaks. They correspond to people with age<=40 and age>40 (https://ibb.co/8bzbDyS).
> > >
> > > Assume the goal is to release this dataset and preserve the salary difference between people with age<=40 and age>40, while protecting the mean salaries. We can apply our mechanism for mean (Section 6.3) on this dataset. The histogram of the released data is at https://ibb.co/Bgn8KQW. Data receivers can obtain the salary difference between people with age<=40 and age>40 accurately by computing the difference between the two peaks, while the mean salaries are protected under our mechanism.
> > >
> > > Here we use the salary difference between people with age<=40 and age>40 as an example. In general, any downstream tasks that depend only on the “shape” of the distribution will not be affected by our mechanism, since our mechanism shifts all samples by the same amount.
> > >
> > > Note that we chose to use this example due to the data we were able to find and the limitations of our current mechanisms. The privacy analysis of our quantization mechanism applies only to 1D data distributions. Hence we needed a dataset in which the two categories were visibly separated even without knowing the labels (e.g., male vs female, in your suggestion). Because the differences in salary based on gender were relatively small in the datasets we found (I.e., they were not large enough to see a bimodal distribution in the unlabeled data), we had to demonstrate the trend on labels (in this case, age) for which the target values (salary) were more clearly separated.
> > >
> > > **Q7: The authors state several times that standard DP (and the variants they discuss) solve a different problem than the one the authors are attempting to address. However, they still use this as a comparison point in the experiments. This isn’t a fair comparison since these methods are basically designed to preserve summary statistics. Thus, the concentration of the DP-based methods in the low distortion, low “privacy” regime in Fig 6 is a feature, not a bug. Something like adding noise to the summary statistic (as described above for the mean) seems like a much more fair comparison.**
> > >
> > > We agree and understand that the goal of these DP mechanisms is different from ours. In fact, our goal in Section 2 was to explain that existing privacy frameworks (including DP and others) are considering different problem settings and/or objectives to ours, which motivates the need for a new privacy framework. That being said, we agree that our experiments in Section 7 are an “apple-to-oranges” comparison. However, these experiments are NOT meant to say that existing privacy frameworks are bad for their purposes. Instead, the goal is to convey that they are not suitable for our problem, confirming our claims in Section 2. This point has been highlighted in the last paragraph of Section 7, but we will make it clearer to reduce confusion. Also, we have received several comments from reviewers asking why various DP mechanisms are not suitable for our problem, so we believe that these experiments may be necessary to answer such questions.
> > >
> > > Regarding the mechanism of adding noise to the summary static, it will induce infinite distortion under our definition. See the answer to Q3.

---

> > > > ### Author Response · Authors · 2023-05-03
> > > > **Rebuttal (Q8, Q9, Q10, Q11)**
> > > >
> > > > **Q8: The citation for DP should be “Calibrating Noise to Sensitivity in Private Data Analysis” by Dwork, McSherry, Nissim and Smith.**
> > > >
> > > > Thanks for pointing it out. We will fix the citation.
> > > >
> > > > **Q9: It’s not true that DP has focused on low-dimensional statistical queries, there is a good amount of work on DP synthetic data (although I agree that it is usually in a different setting to this papers focus).**
> > > >
> > > > We agree that DP has used for releasing high-dimensional data, such as synthetic data generation (mentioned in Section 1). We did not mean that DP is not applicable due to low-dimensional statistical queries, but that it is not applicable due to the fact that it is originally designed for protecting individual samples instead of global statistics. For example, in Section 2.2, we said “Most existing privacy frameworks or mechanisms are not suitable for summary statistic privacy because they either focus on protecting individual records in the data (e.g., differential privacy [13], anonymization [38], sub-sampling [33]) (Fig. 1), or are designed for algorithms that release low-dimension statistical queries of the dataset instead of the entire dataset (e.g., attribute privacy [42], maximal leakage [18], privacy funnel [30]).”
> > > >
> > > > If there are inaccurate statements we missed, please let us know during the discussion phase and we will fix that.
> > > >
> > > > **Q10: It would have been helpful to have the formulation of summary statistic privacy before the discussion of indistinguishability approaches to provide a comparison point (since I was familiar with these approaches and so wanted a comparison point). In particular, I struggled to understand throughout this section what the authors hoped to retain about the data.**
> > > >
> > > > Thank you for the suggestion! We will take it into account when revising the paper.
> > > >
> > > > **Q11: Just before eq 5, it is stated that d is defined in eqn 3, but it is not.**
> > > >
> > > > Apologies, the definition is in the line after eqn 3. We will fix this.

---

### Author Response · Authors · 2023-04-11
**Thank you!**

First, we would like to thank the reviewers for taking the time to read our manuscript and to provide helpful suggestions and feedback. We really appreciate your input and time, and are grateful for the feedback.

We believe that many of the concerns stem from misunderstandings. We will attempt to address your concerns in our response below, and in a revision of the paper. By our reading of the reviews, we do not believe the paper is factually incorrect (though we are of course happy to discuss this further).

Moreover, based on the feedback we have received, we plan to add a section to our paper explicitly listing the limitations of our framework. Some of these are already discussed in the paper (e.g., Section 3.2.1, Section 8), whereas others have been pointed out by the reviewers.

---

### Author Response · Authors · 2023-05-05
**Thank you!**

Dear reviewers,

Thank you again for your valuable feedback and time. In addition to our responses to your comments, we have uploaded a revision in which we attempted to incorporate your comments in red text. We look forward to hearing from you if you have any additional questions or concerns.

---

### Decision · Action_Editors · 2023-06-01

**Recommendation:** Reject

**Comment:**

All three reviewers felt that the work does not yet adequately meet the TMLR acceptance requirement "Are the claims made in the submission supported by accurate, convincing and clear evidence?" Since the other claims in the paper would need to be re-evaluated in light of such a revision, acceptance with minor revision was felt to be inappropriate. The authors are encouraged to revise based on the stated critiques and to consider resubmitting. Indeed, beyond the definition itself and within the framework proposed, the reviewers were more positive. One reviewer also said that the motivation is interesting and is positive on a solution to this problem.

**Audience:**

All reviewers agree that the paper has significant interest to the TMLR audience, as data privacy for statistics is a topic which many in the community are interested in.

**Claims And Evidence:**

The authors (implicitly or explicitly) make the claims that existing privacy frameworks are insufficient to protect against the problem studied, and that the proposed framework is sufficient. All reviewers were unconvinced by these claims. One reviewer felt that the authors did not do a sufficient job showing that this serious weakening of privacy guarantees is necessary (i.e., the response Q3 to Reviewer oLdq). Another reviewer felt that the proposed framework does not protect privacy in a meaningful way. In particular, if one wants to protect the secret and doesn't care about correlations across different statistics, one could just not release the statistic. But the authors release a deterministic version of the secret and did not deal with the fact that it may be encoded through correlations with non-secret columns, which was a major problem to the reviewer privacy-wise. More thought could be put into what the framework protects and doesn't protect, as well as a clear discussion of this point. Finally, the last reviewer said that the work does not sufficiently argue that the current privacy frameworks do not protect statistical information about the data, and that the work referred to is insufficient -- the authors need to give more evidence for this claim.

**Resubmission Of Major Revision:**

The authors may consider submitting a major revision at a later time.